# Phylogenetic comparative analysis of the cerebello-cerebral system in 34 species highlights primate-general expansion of cerebellar crura I-II

Neville Magielse [1,2,3✉], Roberto Toro [4], Vanessa Steigauf[5], Mahta Abbaspour[6,7], Simon B. Eickhoff[1,3], Katja Heuer [4,8,9,10] & Sofie L. Valk [1,2,3,9,10✉]

The reciprocal connections between the cerebellum and the cerebrum have been suggested to simultaneously play a role in brain size increase and to support a broad array of brain functions in primates. The cerebello-cerebral system has undergone marked functionally relevant reorganization. In particular, the lateral cerebellar lobules crura I-II (the ansiform) have been suggested to be expanded in hominoids. Here, we manually segmented 63 cerebella (34 primate species; 9 infraorders) and 30 ansiforms (13 species; 8 infraorders) to understand how their volumes have evolved over the primate lineage. Together, our analyses support proportional cerebellar-cerebral scaling, whereas ansiforms have expanded faster than the cerebellum and cerebrum. We did not find different scaling between strepsirrhines and haplorhines, nor between apes and non-apes. In sum, our study shows primate-general structural reorganization of the ansiform, relative to the cerebello-cerebral system, which is relevant for specialized brain functions in an evolutionary context.

[1] Institute of Neuroscience and Medicine, Brain and Behaviour (INM-7), Research Center Jülich, Wilhelm-Johnen-Straße, 52428 Jülich, Germany. [2] Otto Hahn Cognitive Neurogenetics Group, Max Planck Institute for Human Cognitive and Brain Sciences, Stephanstraße 1A, 04103 Leipzig, Germany. [3] Institute of Systems Neuroscience, Medical Faculty, Heinrich Heine University Düsseldorf, Moorenstraße 5, 40225 Düsseldorf, Germany. [4] Institut Pasteur, Unité de Neuroanatomie Appliquée et Théorique, Université Paris Cité, 25 rue du Dr. Roux, 75724 Paris, France. [5] Department of Biology, Northern Michigan University, 1401 Presque Isle AveMI 49855 Marquette, USA. [6] Berlin School of Mind and Brain, Humboldt-Universität zu Berlin, Luisenstraße 56, Haus 1, 10117 Berlin, Germany. [7] Department of Neurology, Charité – Universitätsmedizin Berlin, Bonhoefferweg 3, 10117 Berlin, Germany. [8] Department of Neuropsychology, Max Planck Institute for Human Cognitive and Brain Sciences, Stephanstraße 1A, 04103 Leipzig, Germany. [9] These authors contributed equally: Katja Heuer, Sofie L. Valk. [10] These authors jointly supervised this work: Katja Heuer, Sofie L. Valk. ✉email: n.magielse@fz-juelich.de; s.valk@fz-juelich.de

The human cerebellum plays a role in various brain functions including executive, social, and emotional processing[1–4], spanning across motor and cognitive systems. Behavioral deficits across these wide-ranging behavioral domains have also been shown in lesion studies[5,6], suggesting the cerebellum does more than its canonical motor functions[7]. Functional magnetic resonance imaging (fMRI) studies reveal a complex cerebellar functional map with distinct activations[8–10] and resting-state connectivity profiles[11–14], distributed across the cortex. The cerebellum may encode errors, with the highly repetitive and evolutionarily conserved[15–18] cerebellar circuit supporting the same computational process across behavioral domains (the universal cerebellar transform theory)[19–23]. This has prompted the idea that the location of cerebellar functions may be chiefly related to its inputs and outputs[23]. Nonetheless, functional diversity may exist at the algorithmic level[22], with cerebellar areas integrating information from distributed brain areas[24–26], instead of simply mapping one-to-one to inputs. In either scenario, cerebellar function depends heavily on connectivity with the rest of the brain. However, how cerebellar volume has changed over the course of primate evolution relative to one of its primary connections, the cerebrum, remains incompletely understood.

The question of how cerebellar structure and its (evolutionary) reorganization may reveal its function[27–29] was already posed in 1906 by Louis Bolk and still motivates evolutionary studies today. So far, comparative studies of the primate cerebello-cerebral system (CCS) have indicated that CCS hyperscaling likely drives primate brain size[30]. Moreover, volumes[31,32] and neuron numbers[33,34] of primate CCS components evolve predictably. Furthermore, the rate of cerebellar volumetric[35] and surface area[36] evolution may have increased relative to the neocortex, specifically towards humans. Last, contemporary humans and apes have larger cerebella relative to recent human ancestors[37]. A reciprocal CCS connection, supportive of cerebellar learning in motor and non-motor function[38–41], was first established in non-human primates (NHPs). Within the highly connected CCS, motor and prefrontal cortex (PFC) streams were largely separated into distinct projection zones[23,42–46]. Such separated information streams can be described as distinct functional modules, supporting relatively distinct brain functions. Note that this macro-scale description of functional modules contrasts descriptions of the smallest operational unit that performs the cerebellar computation[47].

The cerebellum is not only connected to many areas of the cerebral cortex but also to the subcortex and other areas of the body. Its macroscale evolution could thus reflect selection on a combination of behaviors including sensorimotor, vestibular, and socio-cognitive abilities, as well as changes in social niche, diet, and mode of locomotion. However, macroscale evolutionary analyses of specific functional modules[48–50] may reflect somewhat more specific selective forces relevant to primate evolution. Although it is impossible to summarize a brain area's function in a single term—especially in an evolutionary context – human crura I-II appear to primarily be involved in coordinating more distributed processes associated with social cognition, language, and emotion[11–14,24,51]. This idea is furthered by functional imaging, which reveals a strong involvement in the multi-demand and socio-linguistic networks[52], together suggesting the area's general involvement in complex integrative computation. For example, crura I-II shows fMRI activations in tasks involving language and theory of mind, but also executive and emotional processing[9,10]. Resting-state profiles and meta-analytic connectivity studies show that these functions may mainly come from cerebral connectivity, as the area is connected with cerebral areas also involved in such transmodal, integrative

functions[12–14,53]. A cross-species homolog of human crura I-II has been referred to as the ansiform area[54,55]. This term is accredited to Louis Bolk's work[27] on the mammal cerebellum, where he first defined cerebellar lobular terminology[29]. We adopt the term 'ansiform' here to refer to this area. It is not guaranteed that ansiform functions established in humans will be organized in the same way in other primates. However, strong conservation of CCS structural connectivity across primates, alongside high correspondence between structural and functional connectivity, makes it reasonable to assume globally similar functions are supported by this area across primates.

The ansiform is a remarkable cerebellar area. It has a uniquely prolonged developmental trajectory[56]. Moreover, its disruption is often connected to brain disorders with cognitive alterations[57–59] while developmental disruption strongly alters normal socio-cognitive development[60]. In the capuchin (*Cebus apella*), it is interconnected with frontal rather than motor cortical areas. As in all cerebello-cerebral functional modules, a reciprocal closed-loop system connects the ansiform with transmodal cerebral areas through distinct pontine, thalamic, and dentate nucleus zones[23,42–45]. Recent tractography work also illustrates high similarity between the human and chimpanzee CCS[26]. Most generally, primate ansiforms connect reciprocally to transmodal cerebral cortical areas[11–14,26,51,61,62]. Accordingly, evolutionary changes to the ansiform may reflect the selection of abilities supported by an ansiform-transmodal cerebral cortical module.

Reorganization[63–66] of the CCS and its associated connectivity have thus been hypothesized to be important for global functional divisions within the cerebellum[23,46,50]. Indeed, reorganization within primate brains, rather than brain size alone, may best characterize anthropoids[50,63]. We focus our current study on both the whole cerebellar volume and, more specifically, the volume of the ansiform. Relatively large lateral cerebellar hemispheres (mostly consisting of the ansiform) can be noted in four vertebrate clades including primates[67] and particularly hominoids[68]. Specifically, ansiform volume fractions of the cerebellum are larger in humans than in macaques[69], and appear larger in primates than in mice and rats[54]. These gross anatomical expansions in primates, alongside cerebral transmodal expansions[50,51,70–72], could reflect selection of large-scale CCS networks[31,67,69,73,74] that are generally expected to support complex functional processes in cognitive, socio-linguistic, multi-demand networks[52]. Indeed, expansion of the ansiform from macaques to chimps to humans[69] occurs alongside expansion of the PFC[72], as well as the termination zone that connects them in the cerebral peduncles[46]. Currently, it remains unclear how the ansiform scales relative to the CCS, when taking allometry (scaling relationships) and phylogeny (evolutionary relationships) among species and traits into account.

Here we assessed primate CCS evolution through phylogenetic comparative analysis of cerebellar volumes. We used a large (34 species, 63 specimens) open MRI dataset alongside modern phylogenetic comparative methods to consolidate previous evidence of isometric (volume of region y and region x expand at the same rate) scaling between cerebellum and cerebrum[31,32,75]. We evaluated potential differences in slopes or intercepts of cerebellar scaling in apes and non-apes[35,37], as well as strepsirrhines and haplorhines. Additionally, we aimed to assess ansiform hyper-allometry[54,68,69] (hyperscaling; volume of y expands faster than x) in an allometric, phylogenetic context. Where possible, we explored intraspecific variability, substantial among brain traits[68,76–80] and an inherent challenge for comparative primatology[81]. Our analyses confirmed chiefly isometric scaling between cerebellum and cerebrum. Re-analysis of the Stephan collection[82] data indicated that this relation may be hypo-allometric (hyposcaling; volume of y expands slower than x), a

finding supported by several robustness analyses of our data. Conversely, accounting for allometry and phylogeny, we find that primate-general ansiform hyper-allometry directly explains previous observations of this area's large relative size in large-brained primates[68,69,83].

## Results

**Neuroanatomical traits.** First, we report volumetric variations across the 34 primate species included in our study (Fig. 1; Table 1). Intraspecific data spread for species with enough (N ≥ 4) observations (asterisks in Fig. 1 mark species with multiple observations) is given as median absolute deviations (MADs) (Table 1). Absolute volumes were largest in apes, consistent with body size scaling.

*Intraspecific variability is substantial.* For several species (N ≥ 4) we analyzed intraspecific variability. Variability, quantified as

MADs as percentage of the median, ranged from 4.8 to 15.8% for cerebellar volume, and from 6.2 to 20.7% for cerebral volume. Intraspecific ansiform observations were only available for humans (median: 43.73 ± 5.00 cm³; MAD = 11.4%) and chimpanzees (median: 15.54 ± 1.93 cm³; 12.4%) (Table 1, Supplementary Figs. 1, 2). Human volumes were systematically higher in males. The chimpanzee data were generally similar between sexes (Supplementary Fig. 3).

**Evolutionary model testing.** Next, we evaluated fit of evolutionary models, with the best-supported model used in subsequent analyses. Akaike Information Criterion (AIC) testing[84] revealed the following support for evolutionary models: Brownian Motion (BM) (λ = 1.0; AIC = 6209.62) > Early Burst (AIC = 6216.25) » BM (λ = 0.0 (star-phylogeny); AIC = 6267.87) » Ornstein-Uhlenbeck (OU) (single α; AIC = 6609.29) » OU (α per trait; AIC = 7143.13) » OU (full multivariate α; AIC = 8063.37).

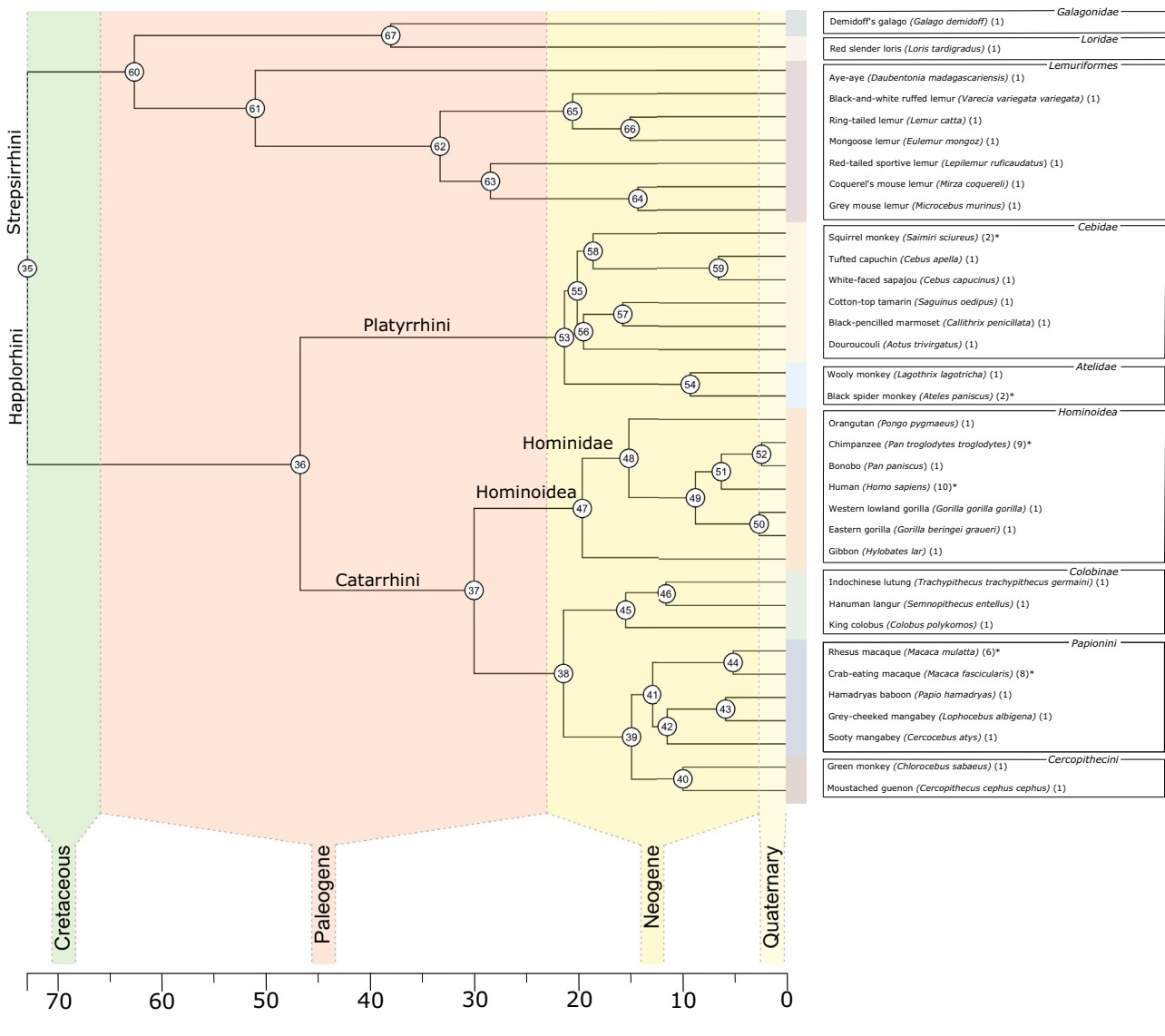

**Fig. 1 Consensus phylogenetic tree for the 34 primate species in this study.** We obtained the consensus tree for the 34 species in our dataset from 10kTrees Arnold et al.[142]. It represents the best-supported evolutionary relationships between the primates in our sample. Archeological epochs are superimposed on the tree to provide a temporal perspective of predicted species bifurcations. Internal node numbers are plotted and can be used to identify ancestral characters (Supplementary Table 1). Extant species included in the current study are provided on the right and are colored by clade membership. Additionally, the sample size per species is given in parentheses. Species with multiple specimens are marked with asterisks.

**Table 1 Neuroanatomical traits.**

| Species | Cerebellar volume | Cerebral volume | Ansiform area volume | Cerebellum-cerebrum ratio | Ansiform area-cerebellum ratio |
|---|---|---|---|---|---|
| Demidoff's galago (1) | 320 | 1,925 | | 16.65 | |
| Red slender loris (1) | 447 | 3,071 | | 14.54 | |
| Aye-aye (1) | 3,032 | 17,151 | 374 | 17.68 | 12.33 |
| Black-and-white ruffed lemur (1) | 3,377 | 16,032 | | 21.06 | |
| Ring-tailed lemur (1) | 2,035 | 12,658 | | 16.08 | |
| Mongoose lemur (1) | 3,224 | 14,495 | | 22.24 | |
| Red-tailed sportive lemur (1) | 882 | 4,087 | | 21.59 | |
| Coquerel's mouse lemur (1) | 884 | 3,885 | | 22.75 | |
| Gray mouse lemur (1) | 170 | 809 | | 20.99 | |
| Squirrel monkey (2) | 920 **± 55** | 8,540 **± 1,263** | 103 | 10.93 **± 2.26** | 11.20 |
| Tufted capuchin (1) | 5,191 | 36,285 | 1,012 | 14.31 | 19.50 |
| White-faced sapajou (1) | 4,274 | 4,005 | | 10.67 | |
| Cotton-top tamarin (1) | 750 | 6,254 | | 11.99 | |
| Black-pencilled marmoset (1) | 485 | 3,938 | | 12.31 | |
| Douroucouli (1) | 1,627 | 11,746 | 217 | 13.85 | 13.31 |
| Wooly monkey (1) | 5,595 | 49,385 | 874 | 11.33 | 15.62 |
| Black spider monkey (2) | 8,903 **± 1,411** | 77,381 **± 10,948** | | 11.49 **± 0.20** | |
| Orangutan (1) | 25,757 | 184,458 | 6,081 | 13.96 | 23.61 |
| Chimpanzee (9) | 57,047 **± 6,915** | 284,482 **± 26,063** | 15,544 **± 1,932** | 18.36 **± 3.56** | 27.25 **± 3.69** |
| Bonobo (1) | 48,408 | 243,036 | 6,627 | 19.92 | 13.69 |
| Human (10) | 142,171 **± 18,710** | 1,005,711 **± 179,368** | 43,726 **± 5,003** | 14.26 **± 1.75** | 30.76 **±1.91** |
| Western lowland gorilla (1) | 58,402 | 237,690 | 13,107 | 24.57 | 22.44 |
| Eastern gorilla (1) | 46,163 | 272,908 | | 16.92 | |
| Gibbon (1) | 8,681 | 60,685 | | 14.31 | |
| Indochinese lutung (1) | 3,756 | 37,988 | | 9.89 | |
| Hanuman langur (1) | 6,749 | 49,916 | | 13.52 | |
| King colobus (1) | 5,787 | 40,468 | 842 | 14.30 | 14.55 |
| Rhesus macaque (6)* | 10,456* **± 500** | 75,928* **± 4,689** | | 13.54 **± 0.64** | |
| Crab-eating macaque (8)* | 6,184* **± 955** | 46,286* **± 9,597** | | 12.89 **± 0.98** | |
| Hamadryas baboon (1) | 9,478 | 89,861 | 1,841 | 10.55 | 19.42 |
| Gray-cheeked mangabey (1) | 6,320 | 49,185 | | 12.85 | |
| Sooty mangabey (1) | 5,362 | 51,689 | | 10.37 | |
| Green monkey (1) | 4,169 | 53,627 | 505 | 7.77 | 12.11 |
| Moustached guenon (1) | 3,697 | 41,538 | | 8.90 | |

Species median measurements are provided for cerebellar, cerebral, and ansiform volumes (in mm$^3$). Species ratios between median cerebellar and cerebral volumes, and ansiform and cerebellar volumes are also given (in percentages). Importantly, these ratios are reported merely to illustrate how allometric scaling of cerebello-cerebral system components may lead to wide-ranging ratios reported in the literature. Species are ordered by the phylogenetic tree. For six species (Ateles paniscus, Homo sapiens, Macaca fascicularis, Macaca mulatta, Pan troglodytes, and Saimiri sciureus) more than four specimens were available. For these species, median absolute deviations are also reported (indicated in bold). Some specimens were recorded as outliers and removed before subsequent analyses. These include a crab-eating and rhesus macaque, which were outliers in their cerebellar and cerebral volumes, as marked by the asterisks.

BM was best supported, significantly more so than Early Burst. Chi-squared testing revealed further support for the $\lambda = 1.0$ versus the $\lambda = 0.0$ model ($\chi^2 = 53.68$). Thus, we adopted BM ($\lambda = 1.0$) for subsequent analyses.

**Ancestral character estimations**. Next, we aimed to estimate ancestral traits based on BM evolution. We provide ancestral volumes for cerebellar, cerebral, and ansiform volumes in Supplementary Table 1 and full ancestral character estimations (ACEs) can be found on GitHub.

*Ratios are confounded by allometry.* To visualize the evolutionary dynamics of the CCS we mapped ancestral character estimations (ACEs) to the phylogenetic tree. We plotted 95% confidence intervals for ACEs (Fig. 2), with uncertainty intuitively increasing with time to present (Supplementary Fig. 4). Cerebellar and cerebral volumes showed virtually identical evolutionary dynamics (Fig. 2a, b), with the largest volumes in apes. Cerebellar volume at the ancestral node of the 34-species tree was estimated

at 1856 mm$^3$, resembling the ring-tailed lemur in our data. Ansiforms (Fig. 2d) were also largest in apes. We observed that ratios of cerebellum-to-cerebrum (Fig. 2c) and ansiform-to-cerebellum (Fig. 2e) may vary greatly across species (see also Table 1). However, these ratios result directly from the allometries described in (Fig. 3a, c; next section), and should thus not be interpreted functionally. We highlight how traits developed over evolutionary time more directly in phenograms (Supplementary Fig. 5).

**Allometric scaling relationships**
*Accelerated ansiform scaling in primates.* Next, we aimed to formally quantify scaling relationships within the CCS. More specifically, we assessed scaling between cerebellum and cerebrum, and ansiform and cerebellum or cerebrum. We employed phylogenetic generalized least squares (PGLS) regression in the context of BM evolution (Fig. 3). Median cerebellar volumes were regressed on median cerebral volume for the full 34-species data (Fig. 3a) and for the 13 species with complete data (Fig. 3b). The

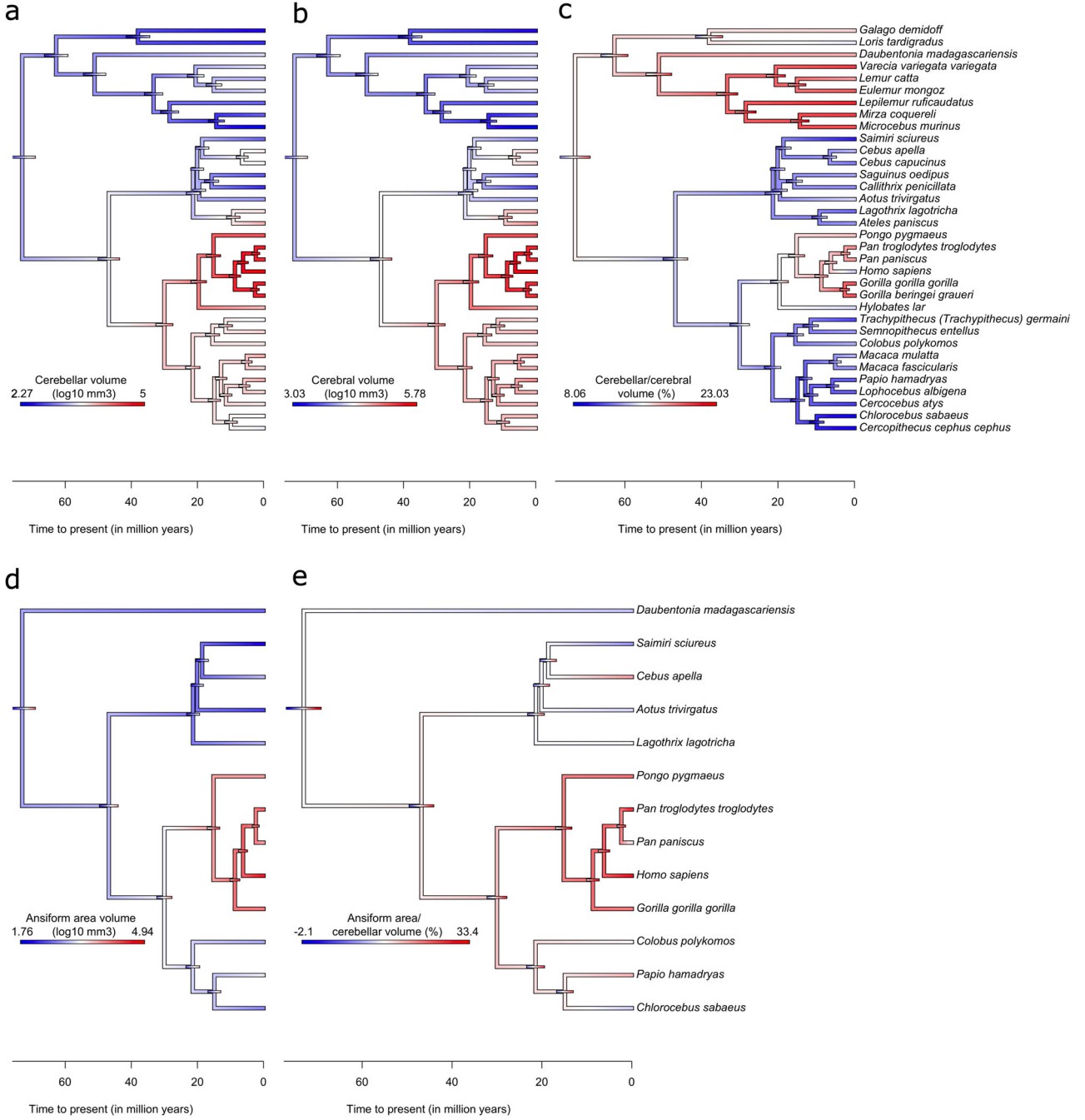

**Fig. 2 Ancestral character estimations for neuroanatomical traits.** Ancestral character estimations (ACEs) based on the Brownian Motion (BM) model of trait evolution are provided for absolute (**a**, **b**, **d**) and relative (**c**, **e**) volumes, alongside node-wise 95% confidence intervals (colored bars on tree nodes). ACEs for cerebellar (**a**), cerebral (**b**), and relative cerebellar-to-cerebral (**c**) volumes were calculated from, and mapped to, the full 34-species tree, whereas ansiform (**d**) and relative ansiform-to-cerebellar volumes (**e**) were calculated from, and mapped to, the 13-species tree. Note that the ratios in (**c**, **e**) serve merely to illustrate how ratios may result from the allometries described in Fig. 3a, c. Gradient colors on the tree represent $\log_{10}$-transformed brain area volumes (in mm$^3$) (**a**, **b**, **d**), and volumetric fractions (in percentages) (**c**, **e**), respectively. The ancestral node was estimated at 73 million years to present. $N = 34$ species in (**a**–**c**) and $N = 13$ in (**d**, **e**).

scaling relationships were slightly below isometry in both cases (slope$_{34}$ = 0.955 and slope$_{13}$ = 0.940), but statistically evolved isometrically. Ansiform volume scaled hyper-allometrically to the rest of cerebellar (ROC; Fig. 3c) and cerebral (Fig. 3d) volumes (slope$_{ROC}$ = 1.297; slope$_{cerebrum}$ = 1.245), with a lower intercept for the cerebrum (intercept$_{cerebrum}$ = −2.784) than for the ROC (intercept$_{ROC}$ = −1.833). Together, the ansiform scaled hyper-allometrically to both the ROC and cerebral volumes.

*No distinct allometries across major primate bifurcations.* To explore divergent patterns across primate groups, we split the data into strepsirrhine (9 species) and haplorhine (25 species) subsets (Fig. 4a–e). We also split the data into ape and non-ape subsets for cerebellum regressions (7 apes ($N = 24$); 27 non-apes ($N = 39$)) and ansiform (5 apes ($N = 22$); 8 non-apes ($N = 8$)) (Fig. 4f–i). Main cerebellar ~ cerebral regressions are replotted in (Fig. 4a, f).

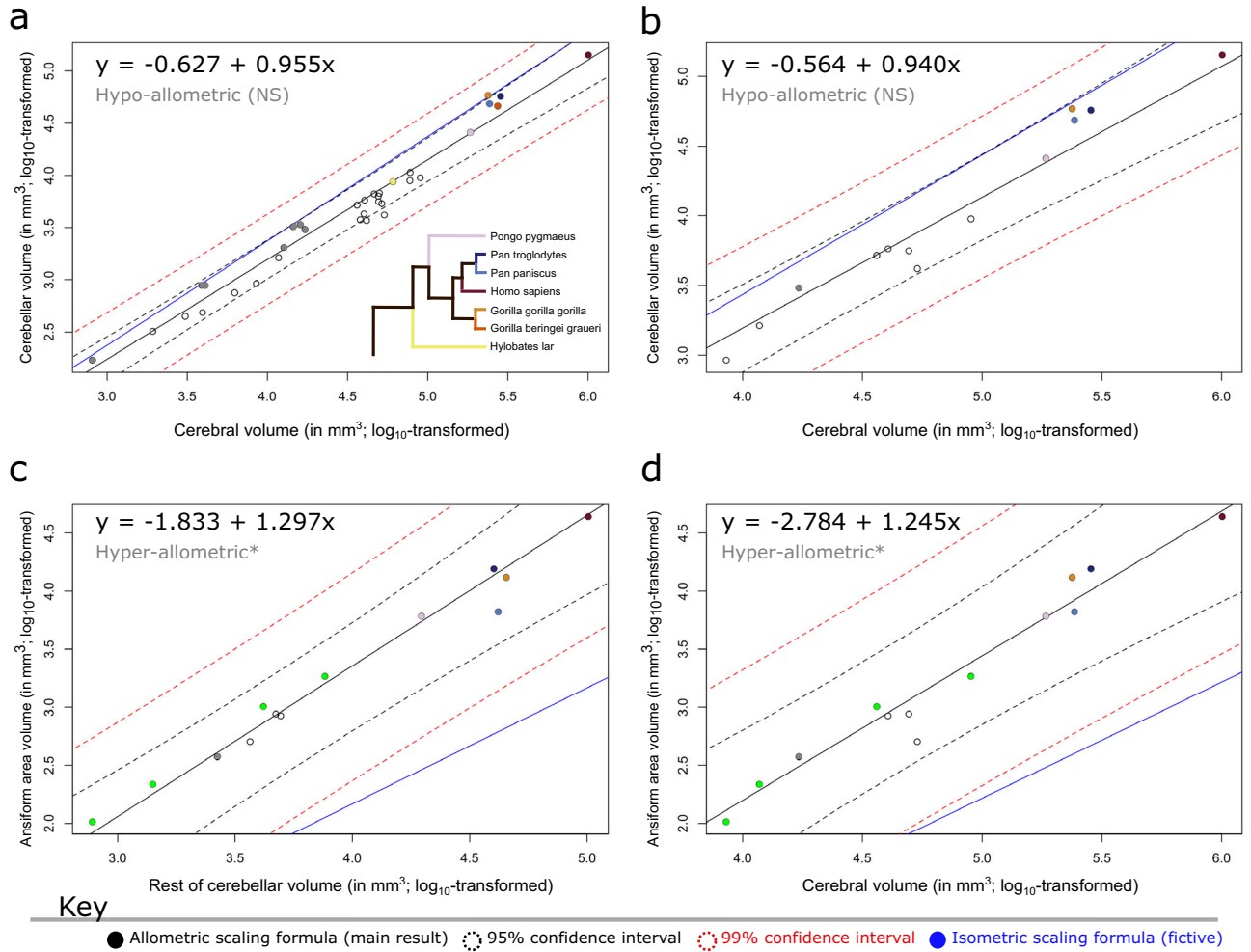

**Fig. 3 Allometric and phylogenetic regressions highlight ansiform hyperscaling relative to the cerebello-cerebral system.** Phylogenetic generalized least squares (PGLS) regressions for cerebellar volume regressed on cerebral volume (**a**, **b**), and ansiform regressed on rest of cerebellar (**c**) and cerebral (**d**) volumes. Median volumetric data were taken from 34-species (**a**) and 13-species (**b–d**). Log$_{10}$-transformed neuroanatomical measures were plotted and overlaid with regression lines obtained in the respective PGLS models. 95% (black dotted line) and 99% (red dotted line) confidence intervals are provided alongside a fictive isometric scaling relationship with the same intercept (blue line). Exclusion of the isometric scaling relationship from the confidence intervals was taken to indicate significant allometry, as indicated by the asterisks. **a**, **b** Cerebellar volumes regressed on cerebral volumes for full data (**a**) and for the 13 species with complete data (**b**) both illustrate isometric scaling trending towards hypo-allometry. Lemuriformes and Hominidea were the two clades with the most impressive cerebellar-to-cerebral volume ratios (Fig. 2c) and were thus specifically colored here (Lemuriformes colored in a bold gray; (**a–d**) to show how ratios were confounded by the primate-general allometry between areas of the cerebello-cerebral system. Although species belonging to these infraorders displayed slightly higher cerebellum-to-cerebrum scaling than the primate sample as a whole (**a**), most of the differences in ratio resulted directly from allometry. Zooming in, (**a**) illustrates that while most Hominoidea, including *Homo sapiens*, *Pongo pygmaeus*, and *Hylobates lar* fall on the regression line of the sample, all but one of the Lemuriformes (*Lemur catta*; on the line) exceed the general trend. Additionally, in both clades, several species approach isometry, with one member of each falling on the isometric line (*Gorilla gorilla gorilla* and *Mirza coquereli*). (**b**) Illustrates reducing of the PGLS slope, connected to the smaller sample of species. **c**, **d** Ansiform volume regressed on the rest of cerebellar volume (**c**) and cerebral volume (**d**) both illustrate hyper-allometric scaling relationships. Because of strong positive allometry, species with larger cerebella and cerebra are expected to have larger relative ansiforms, directly accounting for the high ratios reported in (Fig. 2e). **c**, **d** Both illustrate that Hominoidea do not have uniquely large ansiforms. Although some Hominoidea lie above this steep regression line, so do several other—smaller—species (as colored in green; *Mirza coquereli*, *Aotus trivirgatus*, *Cebus apella*, and *Papio hamadryas*). NS = non-significant. N = 34 in (**a**), and N = 13 in (**b–d**).

First, in the strepsirrhine-haplorhine analysis, we observed that although cerebellar volume scaled virtually identically across groups (Fig. 4b), cerebral volume appeared somewhat smaller relative to body mass in strepsirrhines (Fig. 4c). However, we show that relative to the rest of the brain, cerebella appeared larger in strepsirrhines (Fig. 4d). Conversely, cerebral scaling was virtually identical across groups (Fig. 4e). Differences were minor, with phylogenetic ANCOVA (pANCOVA) revealing no significant differences in scaling in any comparison. Supplementary Fig. 6 illustrates how cerebellar and cerebral volumes differ

relative to brain volume across these groups due to allometry. To further evaluate how brain-body scaling may contribute to our findings, we visualized encephalization in our data (Fig. 5). We observed a characteristically high encephalization quotient (EQ) for humans, in line with previous work (6.51; Jerison's EQ = 7.59[85]). We found that brain-body scaling differed across clades and was widely distributed between even closely related primates, e.g., between apes.

Second, for apes and non-apes, we observed no significant differences between cerebellar-cerebrum PGLS regressions (Fig. 4f),

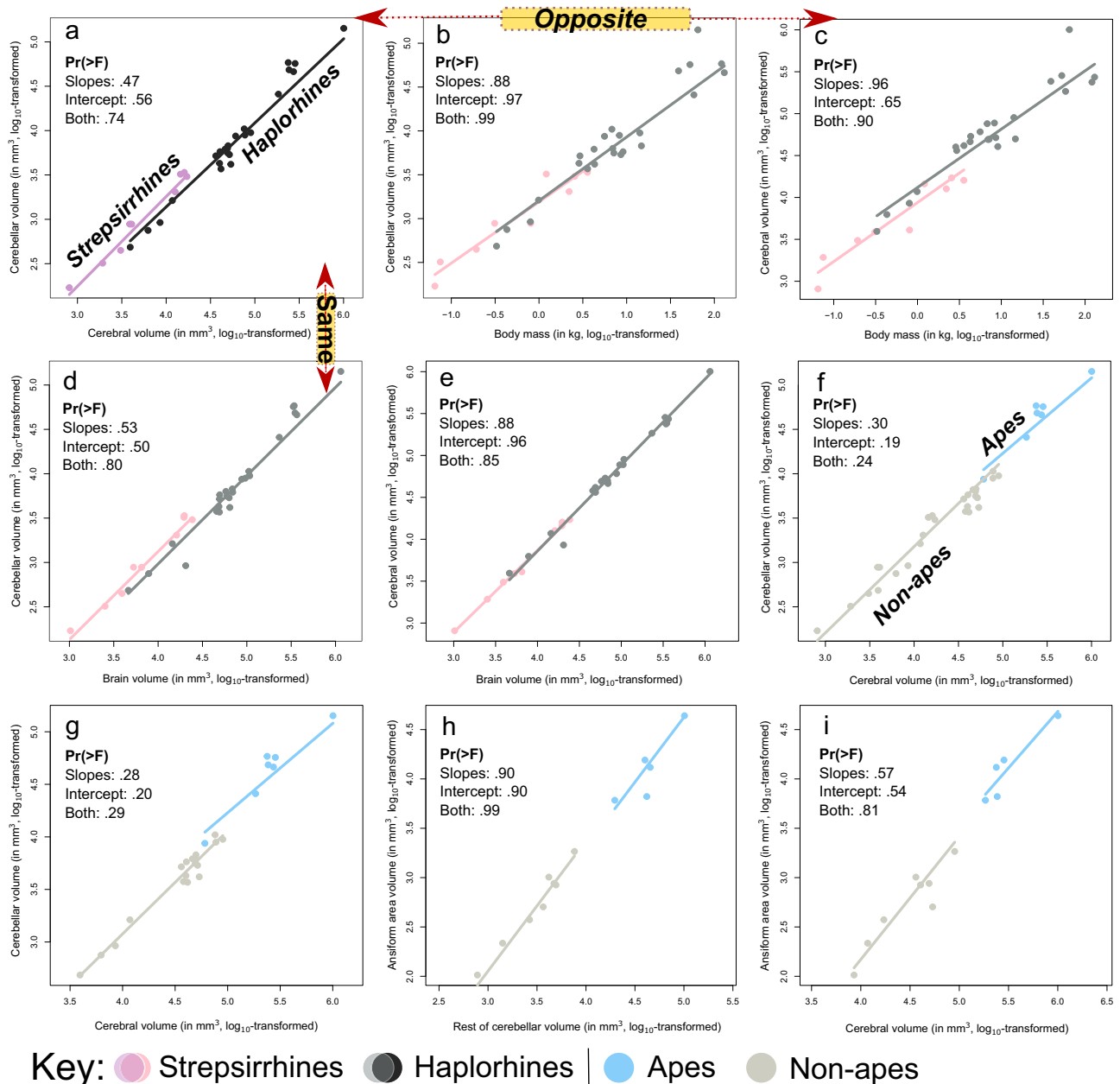

**Fig. 4 Cerebello-cerebral scaling may be primate-general.** Data were split into strepsirrhine and haplorhine (**a–e**) and ape and non-ape (**f–i**) subsets. Separate PGLS regressions were performed in these groups. Importantly, none of the grade shifts were significant as assessed by phylogenetic ANCOVA (**Pr > F** are given for the varying slopes, intercept, or both). For strepsirrhines, a slightly higher intercept was observed for cerebellar scaling relative to the cerebrum (**a**; main analysis). When regressed against body mass, cerebella scaled similarly between groups (**b**), whereas the cerebral PGLS had a slightly lower intercept in strepsirrhines relative to haplorhines (**c**). This showed the opposite pattern relative to the main PGLS (**a**). Within the brain however, cerebellar volumes appeared to be more influential on the regression in (**a**). The jump in intercept for strepsirrhines in cerebellar PGLS mirrored the main PGLS (**d**), while cerebra scaled virtually identically between groups (**e**). For apes/non-apes, it appeared that ape cerebella were relatively large (**f**; main analysis). This (merely visual) shift was slightly accentuated by restricting the analyses to haplorhines as in Barton and Venditti[35]. (**g**). Ansiform regressions on the rest of cerebellar (**h**) and cerebral (**i**) volumes revealed no differences. Together, the lack of significant differences in any pANCOVA argue for primate-general scaling. $N_{strepsirrhine} = 9$; $N_{haplorhine} = 25$ (**a–e**); $N_{ape} = 7$; and $N_{non-ape} = 27$ (**f**); $N_{ape} = 7$; and $N_{non-ape} = 18$ (**g**); and $N_{ape} = 5$ and $N_{non-ape} = 8$ (**h**, **i**).

even when restricting the analysis to haplorhines (Fig. 4g). No scaling differences were apparent for the ansiform (Fig. 4h, i). Fisher's R-to-Z transformation showed that fit was always worse when fitting separate regressions for apes and non-apes (Supplementary Note 1).

**Replication in the Stephan dataset**. To further validate the stability of our results, we replicated PGLS analysis in an alternative

set of primates, the Stephan dataset[82] (Fig. 5). This dataset contains postmortem primate brains, with volumes corrected for shrinkage. BM was again the best-supported model (AIC = −19.13), with BM ≈ Early Burst (AIC = −17.51) ≈ OU (diagonal α; AIC = −16.03). It significantly outperformed the star-model ($\chi^2$ = 84.60). We adopted BM to map the ancestral character estimations (ACEs) to the tree. Cerebellar and cerebral volume

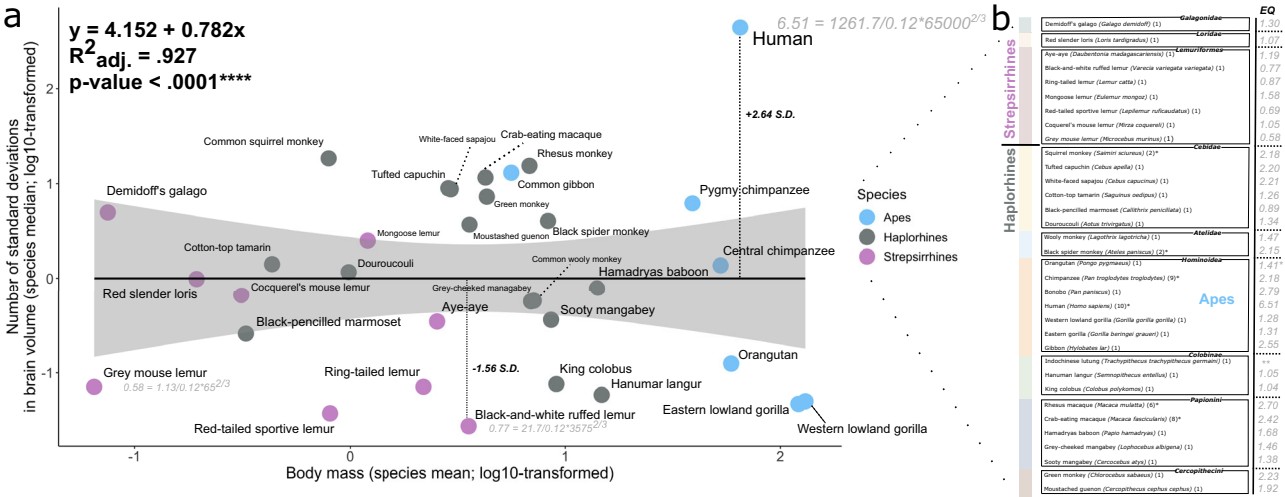

**Fig. 5 Brain-body scaling in the primate sample.** Here, we plotted the number of standard deviations from the brain volume–body mass regression versus body mass to visualize how they covary among clades and major primate bifurcations (**a**). Gray shading indicates the 95% confidence interval for the linear regression. We also replicated Jerison's encephalization quotient (EQ) in our data, describing brain-to-body scaling relative to expectation based on allometric scaling (**b**). We used the allometry described by Jerison[100] for mammals: $EQ = \frac{0.12}{P^{2/3}}$, with $P =$ body mass. EQ varied greatly between clades, and somewhat less within them. Deviations from our primate brain-body regression show little within-clade consistency or relation to grouping. *Indicates suspected outliers; **indicates missing values. $N = 34$ species (63 observations).

ACEs displayed only slightly larger spread and clade-wise distribution was similar. PGLS for cerebellar volume regressed on cerebral volume indicated significant hypo-allometry, with its slope of approximately 0.92 being slightly reduced relative to the main analysis (Fig. 5b). This slope was again able to explain observed ratios (Fig. 5a) well.

**Robustness analyses and exploration of data.** Lastly, we aimed to assess robustness of our results. First, we showed that cerebellar ~ cerebral scaling differed between data from the museum collection (MNHN) and rest of the data (Supplementary Fig. 7a). Notwithstanding, the museum PGLS scaling resembled the full data PGLS: $y = -0.274 + 0.977x$ (Supplementary Fig. 7b). We went on to simulate shrinkage across areas and brains (including other intraspecific variation). We showed that the cerebellar–cerebral regression supported a range of allometries: from relatively strong hypo-allometries (~0.8) to smaller hyper-allometries (~1.1) (Supplementary Fig. 7c, d). Conversely, ansiform regressions were resilient to this uncertainty and argued firmly for hyper-allometry (Supplementary Fig. 7g–j). Introducing artificial 2x intraspecific variability revealed that most PGLS analyses supported cerebello-cerebral hypo-allometry, despite intraspecific variation (Supplementary Fig. 7e, f).

Rerunning PGLS in a more conservative subset that excluded potential cerebellar volume outliers showed stronger cerebello-cerebral hypo-allometry (Supplementary Fig. 8a), tending toward significance while better matching the Stephan PGLS (Supplementary Fig. 8b). Ansiform regression on the rest of the cerebellar volume increased in hyperscaling (Supplementary Fig. 8c), while that on cerebral volume remained virtually identical (Supplementary Fig. 8d).

Lastly, considering our finding of primate-general scaling, we explored if ratio differences across clades (Fig. 2) could be directly related to body size or brain volume. Cerebellum-to-cerebrum ratios were neither related to body mass[86] ($R^2_{adj.} = -0.03$; $p = 0.88$) (Supplementary Fig. 9a) nor brain volume ($R^2_{adj.} = -0.01$; $p = 0.45$) (Supplementary Fig. 9b). Conversely, ansiform-to-cerebellum ratios were related to both body mass ($R^2_{adj.} = 0.50$; $p < 0.01**$) (Supplementary Fig. 9c) and brain volume ($R^2_{adj.} = 0.60$; $p < 0.0001****$) (Supplementary Fig. 9d). This

relationship between ansiform fraction and brain or body size results directly from strong positive allometric scaling.

## Discussion

In the current work, we studied cerebellar volumes for 34 primate species ($N = 63$) and ansiform volumes for 13 species ($N = 30$). The cerebellum, and the ansiform specifically, are part of important cerebello-cerebral loops and networks contributing to transmodal, integrative functions[12–14,31,39,45,46,50,61,62,69,87,88]. Together, our analyses showed support for isometric-to-hypo-allometric scaling between the cerebellum and cerebrum. Conversely, the ansiform displayed strong hyper-allometry versus the rest of the cerebello-cerebral system (CCS). We did not find statistical support for different scaling between strepsirrhines and haplorhines, nor between apes and non-apes. Given this primate-general scaling, species with larger brains are expected to have larger relative ansiforms, explaining previous reports of their high ratios in chimpanzees and humans[69].

Our analyses of CCS evolution showed the strongest support for Brownian Motion (BM) ($\lambda = 1.0$). This fits previous findings for the evolution of neocortical traits[89] and is further supported by additional analyses in data from the Stephan collection[82]. The BM model, describing random incremental changes predictable by time since divergence, fits with constrained scaling of the CCS in our study. Although departures from BM have been interpreted as indicating adaptive variation[90–92], BM does not rule out adaptation[93]. Recently, generalization of the BM model has been shown to represent both neutral drift and rapid adaptive evolutionary change. BM as implemented here has more difficulty describing the latter mode of evolution, because it assumes neutral and gradual change[94]. It remains open whether homogeneous models (a single regimen per tree, with useful statistical properties[93]) or heterogeneous models (different regimes on subtrees) will best elucidate primate evolutionary complexity[94].

Our main PGLS supports isometric cerebello-cerebral[31,32,95–97] evolution, tending towards hypo-allometry (Fig. 3a; ~0.95). To assess the effects of potential confounding factors, we ran several robustness analyses. These indicated that our main results generally hold when PGLS is rerun in data i) without potential outliers identified from the literature, ii) from only museum

specimens, and iii) accounting for shrinkage differences between brain areas. Introducing shrinkage differences within-brain showed that cerebello-cerebral scaling may range from rather strong hypo-allometries (~0.8) to slightly weaker hyper-allometries (~1.1) (Supplementary Fig. 7c, d). Introducing severe artificial shrinkage differences across-brains showed mostly reduced allometries relative to our main PGLS (Supplementary Fig. 7e, f), as did reanalysis without literature outliers (Supplementary Fig. 7a; ~0.93) and the Stephan replication (Supplementary Fig. 7b; ~0.92). A recent study reported similar and significant hypo-allometric cerebello-cerebral surface area scaling across mammals (~0.92)[97]. The museum (MNHN)-only PGLS was the exception (Supplementary Fig. 7a; ~0.98), with an increased slope relative to our main analysis. Although our results might hint at a slight cerebello-cerebral hypo-allometry, it becomes clear from these data how sparsely sampled, low-n, comparative data may lead to contradictory conclusions[98], especially when considering near-isometric scaling. Our robustness analyses showed that providing several complementary views on comparative data may reveal more than small sample p-values.

Previous studies have reported tight primate cerebello-cerebral scaling[31,32], as well as indications that cerebella may have become relatively large[95] and undergo accelerated growth in hominoidea[35]. Although we found no statistical evidence for relatively large cerebella in apes, restricting analyses to haplorhines (as in Barton and Venditti[35]) somewhat accentuates a small non-significant grade shift in apes (Fig. 4g). Additionally, apes' positive deviations in PGLS (Figs. 3, 6) hinted slightly towards accelerated cerebellar growth in apes[35]. A combination of small samples, near-isometric scaling, and relatively small differences between apes and non-apes may cause conflicting conclusions on cerebello-cerebral scaling. However, our study did not consider branch-wise evolutionary rates, which appeared more strongly accelerated in apes[35]. In line with the notion of accelerated cerebellar growth, relatively large cerebella are noted in contemporary humans and apes, but not in recent human ancestors[37].

Relatively large cerebella were also noted in lemurs, although this may result from smaller cerebra across strepsirrhines. Although strepsirrhines are smaller-brained and have smaller cerebra, their cerebella might be slightly larger than those of haplorhines (although not significantly), whereas cerebra scaled virtually identically relative to brain volume between groups (Fig. 4). Although brain-body scaling (Fig. 5) may be related to cognitive ability, as per the encephalization quotient hypothesis[85,99,100], we argue that reorganization of behavior-supporting systems within brains may be more relevant[63–65]. Brain reorganization[63–66,101] may paint a nuanced and complementary picture regarding links to comparative ability, alongside (relative) brain size[85,99,100,102,103]. Within the brain, both apes and lemurs appear to deviate positively from primate-general cerebello-cerebral scaling in PGLS. Future studies may seek to explore how relatively large cerebella may result from cerebral scaling, as well as investigate behavioral correlates of large cerebella. Both are likely quite different between strepsirrhines and haplorhines.

Relative to the CCS, the ansiform scaled significantly hyper-allometrically with a high slope (~1.3) (Fig. 3c, d). In contrast to cerebello-cerebral scaling, relative ansiform scaling appeared to remain hyper-allometric despite simulating unequal shrinkage across areas (~1.1 to ~1.6) (Supplementary Fig. 7g–j). Such hyperscaling likely drives much of the lateral cerebellar expansion observed in large-brained hominoidea[68], and primates more generally[67], and perhaps occurs alongside cerebral transmodal

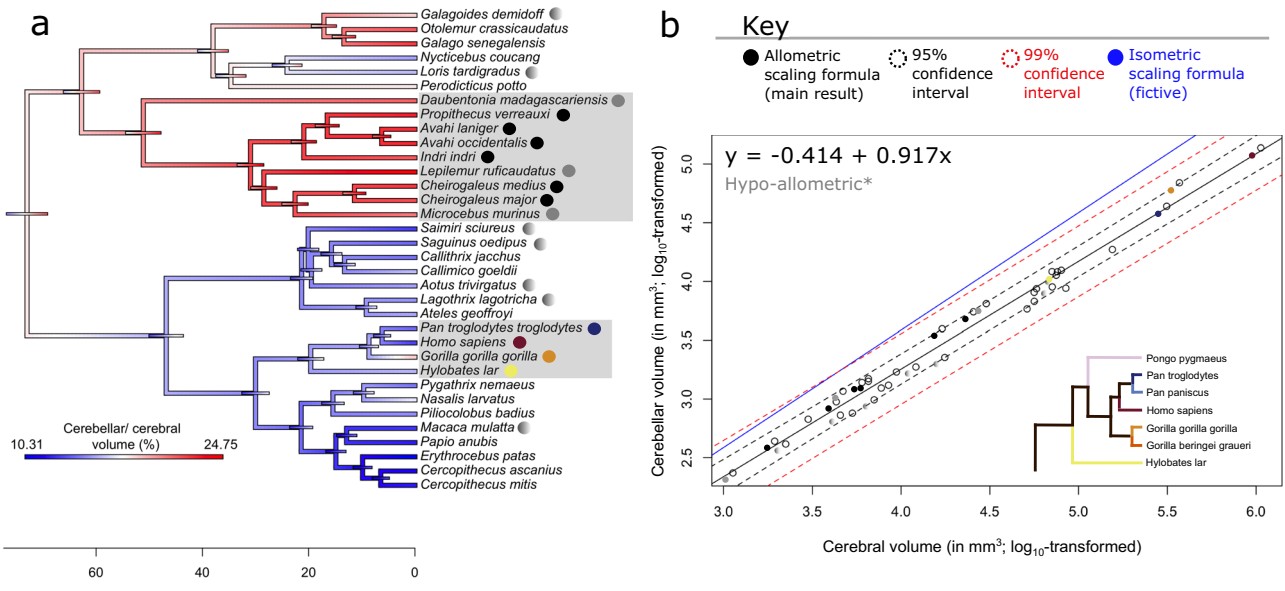

**Fig. 6 Replication of allometric relationship between the cerebellum and cerebrum in the Stephan collection.** The Stephan et al.[82] collection reported cerebellar and cerebral volumes for 34 species available in 10kTrees Arnold et al.[142]. Ancestral character estimations (ACEs) (**a**) and phylogenetic generalized squares regression (PGLS) (**b**) were repeated in this complementary dataset. Despite only partial overlap in species, ACEs for cerebellar-to-cerebral volume ratios (**a**) largely mirrored the main analysis (Fig. 2c). PGLS regression between cerebellar and cerebral volumes in the Stephan collection (**b**) illustrated generally similar scaling as the primary analysis (Fig. 3a), albeit reduced at approximately 0.92 (black line), and with its 95% confidence intervals (black dotted line), but not 99% confidence intervals (red dotted line), excluding isometry (blue line). All apes and lemurs fell on or above the regression line, approaching isometry and mirroring the main analysis (Fig. 3a). Altogether, the analyses again show how allometry causes diverse ratios between brain areas to arise across species. Legend: apes are colored as in Fig. 3a, with legend again shown here in (**b**). Lemuriformes are colored here as well, with solid grays representing species overlapping with the main analysis, and solid blacks indicating lemur species unique to the Stephan dataset (**a**, **b**). Lastly, other species overlapping between both datasets, that were not specifically considered, are colored a shaded gray (**a**, **b**). N = 34 species.

expansions (together part of a CCS functional module)[50,73]. It also explains larger ansiform fractions in larger-brained primates[69,83]. We found that the ansiform fraction, but not the cerebello-cerebral fraction, scales with brain volume (Supplementary Fig. 9), an expected outcome of a strong hyper-allometry. Together, these findings paint a coherent picture of primate-general hyperscaling of the ansiform, that exceeds the hyperscaling previously reported for the prefrontal cortex (1.18x relative to neocortex)[72].

The implications of such hyperscaling are manifold. First, it supports the notion of modular evolution[49] amidst anatomical constraints[104,105] of the CCS. Specifically, a combination of developmental and functional[48] constraints likely leads to tight scaling of cerebello-cerebral structures[31,32]. Amidst this isometric scaling, the ansiform-prefrontal link starts to scale at a steeper slope[50,67,73]. Small timing differences in cell division cycles have been found to explain hyperscaling of (neo)cortical surface area in primates[106]. Common neurodevelopmental mechanisms may thus underlie both tight cerebello-cerebral scaling across vertebrates[31,32,34,71,87,88] and modular ansiform expansions[107], suggesting two prominent evolutionary theories[49,104] are not mutually exclusive[50].

Connected to this, primate-general hyperscaling can reframe brain scaling as a matter of size. Humans may have specialized cognitive abilities and even exceptionally large relative brain size[85], but evolution has not acted to make humans exempt from its laws (the *scala naturæ*). Although specific anatomical specializations[108–110] may in part cause human abilities, large human brains[85] may also lead to some level of behavioral emergence, through an interaction of allometries that affect whole-brain scaling. Specifically, we speculate that the combination of many primate-general allometries might lead to increasingly abundant neuroanatomical systems that support the emergence of abilities that are more than the sum of their parts[111]. The ansiform-transmodal cortex module clearly contributes to integrative primate behaviors[10,12–14,24,50,67,69,112], but its evolution is primate-general. Anatomically, humans may merely represent an extreme of many allometries: their large CCSs[30] having enlarged modules supporting complex and abstract functional processing[51,71,72], increased white-matter connectivity[46,51,113], and a large neuronal pool[33,34]. The same architecture at a different scale may support relatively unique behaviors.

The anything-you-can-get-basis of comparative studies warrants some consideration. For example, most species in the current work are represented by one specimen, so that much of natural variation remains unaccounted for. First, it is important to consider that intraspecific variability in any brain volume is substantial[77,79,80]. Small intraspecific samples can thus not be representative of the range of natural variability. Although we show robustness of our general conclusions at the primate-level, this means that deviations for individual species cannot be used to infer adaptation, since precise volumes may differ. We show variability of volumes across our data to be substantial[78–80], as they include specimens across sexes and adult ages.

We took care to provide the most accurate volumetric measurements possible by manually segmenting cerebella. Relative to automated segmentations in CERES[114], volumes were systematically higher for the cerebellum. This resulted from under-inclusion at cerebellar edges by CERES and over-inclusion at the pontine-white matter border in manual segmentation, intending to isolate the cerebellum similarly across species. Universal guidelines for comparative segmentation might help decrease differences, crucial for relatively small-sized comparative studies.

Rerunning PGLS in data excluding potential outlying cerebellar volumes from the literature[31,68,82,95,115,116] (Supplementary Fig. 8) revealed that the main PGLS was robust, but had potentially somewhat overestimated cerebello-cerebral scaling. Discrepancies between these results and volumes reported across the literature show the impact of intraspecific variation.

Reassuringly, the ansiform was observed to consistently take up a large fraction of the cerebellum across humans ($N = 10$) and chimpanzees ($N = 9$), predicted by the main allometry (~1.3). Our data generally resemble prior observations for mean ansiform volumes in humans (current study: 43.94 cm$^3$; Balsters et al.: 53.65 cm$^3$; Makris et al.: 43.01 cm$^3$), intraspecific variability (5.00 cm$^3$; 8.01 cm$^3$; 6.38cm$^3$), and relative ansiform-to-cerebellum fraction (30.76%; 36.51%; 29.58%)[69,83]. Reanalysis of ansiform scaling in larger datasets will be necessary to confirm our observations as they may be partially driven by an underpowered sample and influenced by specimen-specific idiosyncrasies[81,98,117].

Our study compared brain areas within specimens, helping to minimize the influence of variability across brains. Nonetheless, shrinkage could alter exact scaling formulae and would ideally be accounted for. We assumed that intraspecific variability, compounded by measurement error due to unequal shrinkage, may lead to a twofold difference in volumetric estimates[77,79,80]. Simulating the effect of introducing this volumetric variability in cerebello-cerebral PGLS (Supplementary Fig. 7e, f), we found generally reduced allometries relative to our main PGLS. This result was consistent with analysis of the Stephan dataset, which used shrinkage-corrected volumes[82].

Although sometimes assumed to be of minor importance[101], within-brain shrinkage was expected to play a larger role. Cerebellar, cerebral, and ansiform shrinkage may differ[118], perhaps based on white-matter content[119]. Simulating this shrinkage (Supplementary Fig. 7c, d, g–j) revealed that conclusions on cerebello-cerebral scaling may differ depending on how areas have shrunk with respect to one another. This may also explain contradictory conclusions on near-isometric cerebellar scaling across datasets[35]. Ansiform hyperscaling was robust to this shrinkage. In the future, standard recording of fresh brain weights, and calculation of time-related shrinkage across brain areas may help combat uncertainty in volumetric estimates and, potentially, contradictory results[81,98].

Systematic analysis of larger datasets containing both many species and sizable intraspecific samples might resolve previous controversies[78,81,98,120,121]. Creating community-wide guidelines for data collection and statistical handling can facilitate the integration of data from primate studies[50,122–124]. Statistical methods to incorporate intraspecific samples already exist[125–127]. To further data integration, databases can be used to earmark primate brains with unique identifiers, linked to relevant meta-data including sex, age, body weight, captivity status, data availability, and inclusion in specific studies. Lastly, museum collections can greatly expand comparative samples[128].

Summarizing, we report cerebellar volumetric data for 34 species ($N = 63$) and ansiforms (crura I-II) for 13 species ($N = 30$). Our phylogenetic analyses provide corroborating evidence for isometry-to-hypo-allometry between cerebellar and cerebral volumes[31,32,97] and provide evidence for primate-general ansiform hyperscaling[69,83]. A combination of constrained[104] and modular[49] evolution may explain changes in the volume of the primate cerebello-cerebral system. The consistent finding of cerebello-cerebral coevolution argues for their structure and function being intimately linked. The strong positive allometry of the ansiform in primates, potentially alongside that of structurally connected cerebral transmodal cortices[51,71,72,129], illustrates modular evolution within the primate cerebello-cerebral system. However, larger datasets are needed to elucidate exact primate scaling and determine if species or groups of species deviate from allometry more than expected.

In conclusion, within the tightly coevolving cerebello-cerebral system, the ansiform-transmodal cerebrum functional module may be a primary area of interest in the study of human and comparative cognitive correlates in the future. Large datasets, in combination with relevant metadata and sizable intraspecific samples, will increase understanding of the behavioral implications of primate hyperscaling of this module within the cerebello-cerebral system.

## Methods

**Data provenance and acquisition**. Primate comparative data are rare and sizable datasets are difficult to obtain. We used the collection of primate species collated from different sources in the Brain Catalogue Primates project[89], available on BrainBox[83]. It included 35 species ($N = 66$), which covered the primate phylogenetic tree relatively well, with 9 infraorders from strepsirrhine and haplorhine suborders. Cebidae (7 species), Hominoidea (7), Lemuriformes (6), and Papionini (5) were the most extensively sampled (Fig. 1). The data combines MRI images obtained from diverse sources, including donations, open data, and acquisitions from Heuer et al. (2019)[89] (Supplementary Data 1). The current study followed the institutional review board guidelines of the corresponding institutions. Newer acquisitions included MRI scans from the Vertebrate Brain Collection of the National Museum of Natural History in Paris, France[89]. Scans included both fully abstracted brains and in situ brains, either full body scans or only including the skull, and were acquired on three scanners: two types of 3 T Siemens scanners (Tim Trio and Prisma) and a 11.4 T Bruker Bioscpec. Scanner. Beyond the new scans obtained by Heuer et al. (2019)[89], the Brain Catalogue Primates project[89] includes nine chimpanzee T1-weighted (T1w; 3 T) MRI scans from the National Chimpanzee Brain Resource (NCBR; chimpanzeebrain.org)[96], two crab-eating an one rhesus macaque (T1w; 7 T) from the Pruszinsky Lab[130], four crab-eating and four rhesus macaques (T1w; 3 T) from the Primate Data-Exchange (PRIME-DE)[131], a crab-eating macaque and eastern gorilla downloaded from braincantalogue.org, a bonobo, gibbon, and a western lowland gorilla (T1w; 1.5 T) from the NCBR[96], and finally 10 human brains (T1w; 3 T) from the ABIDE-I dataset[132]. ABIDE-I gained approval from the local Institutional Review Boards and had the 18 Health Insurance Portability and Accountability-protected health information identifiers removed[132]. No new primate brains were obtained in his study. Together, T1- or T2-weighted (T1w or T2w) MRI anatomical scans were obtained at either 1.5 T, 3 T, 7 T or 11.4 T for up to 12 h. For all brains, 3D gradient-echo sequences (FLASH) images were obtained[89]. MRI quality descriptors including scanning resolutions, and signal-to-noise (SNR) can be found in Supplementary Data 1. For full provenance and acquisition parameters, see Heuer et al.[89].

*Data preprocessing and quality control.* Human[132,133] and non-human primate (NHP)[89] data were already preprocessed. SNRs of the primate scans be found in Supplementary Data 1. Comparative datasets often vary in data quality, due to differences in brain size, tissue conservation (in vivo, in situ, or extracted brains), MRI resolution, and SNR, as well as geometrical brain differences, and more idiotypic differences related to individual specimens. During manual segmentation (see the following) extensive visual inspection of the data was performed by NM (as well as previously by KH and RT)[89]. The red howler monkey was excluded due to irreparable damage. Extensive damage was also noted in the black-and-white ruffed lemur, eastern gorilla, and gibbon. Additionally, two cerebellar were not noticeably missing

any tissue, but were deformed. Volumetric estimates for specimens with damaged brains need to be treated with due diligence.

Lastly, postmortem brains shrink. Comparing postmortem to fresh brain weight[82] is the general practice for estimating shrinkage. Unfortunately, we could not measure fresh brain volume, nor could we specifically determine brain tissue gravity in our data, necessary to determine unequal shrinkage across areas. We did thus not correct for shrinkage. Although one would ideally correct volumes for individualized specimen- and brain area-specific shrinkage factors, determined within a controlled experimental environment, this is usually not practically feasible and to our knowledge has not been done to this point (see Stephan et al.[82] for the closest approximation of this ideal). Previous comparisons between volumes obtained from in vivo MRI (Yerkes sample) and postmortem brains (Stephan sample) indicated no difference in reliability as it relates to the study of allometry[68]. Finally, our specific within-brain study design, comparing volume within specimens, should appreciably lessen the effect of shrinkage across brains.

**Manual segmentation procedure**. To provide the most accurate volumetric measurements in the highly variable primate data, we manually delineated the areas of interest directly on the MRI data[134]. Before this manual segmentation, we semi-automatically obtained initial cerebellar masks (serial 2D-outlines in MRI sections used to reconstruct the cerebellum in 3D). We did this by subtracting existing cerebral masks from whole brain masks with Stereo-taxicRAMON (github.com/neuroanatomy/StereotaxicRAMON) or using Thresholdmann (github.com/neuroanatomy/thresholdmann) to interpolate between thresholds at manually selected locations. Initial cerebellar masks were uploaded to BrainBox[135] for collaborative segmentation. Lastly, we used StereotaxicRAMON for mathematical morphology operations that preserve original mask topology (Fig. 7a).

*Cerebellum.* Manual segmentations (Fig. 7b) consisted of brainstem removal, outer cerebellar boundary determination, erroneously marked sulci erasure, and damaged tissue reconstruction. Segmentations ($N = 65$ specimens) were performed by NM, with SLV providing preliminary segmentations for a subset of specimens. Damaged tissue was reconstructed by methodical manual interpolation from adjacent, non-damaged slices. Segmentations were refined until satisfactory, using the three stereotaxic planes and alternating manual segmentation and mathematical morphology operations. All segmentations were approved by KH and SLV.

*Ansiform.* Segmentation (Fig. 7c) relied on identification of the superior posterior (SPF) and ansoparamedian (APMF) fissures[17,69,136]. Since data quality was variable, segmentation was performed in a subset of representative specimens where fissures were identifiable: 5 apes ($N = 22$; including 10 humans and 9 chimpanzees) and 8 non-apes ($N = 8$). Segmentations were again iteratively performed until satisfaction. An interobserver strategy validated segmentations made by NM. For six species, a secondary observer (MA) provided blinded segmentations alongside NM. Additionally, a tertiary observer (VS) segmented human and chimpanzee ansiforms in consultation with NM. All segmentations were checked by KH and SLV.

*Reliability of segmentations.* Reliability of ansiform segmentations was assessed by ANOVA:

$$Ansiform\ volume \sim species + observer \qquad (1)$$

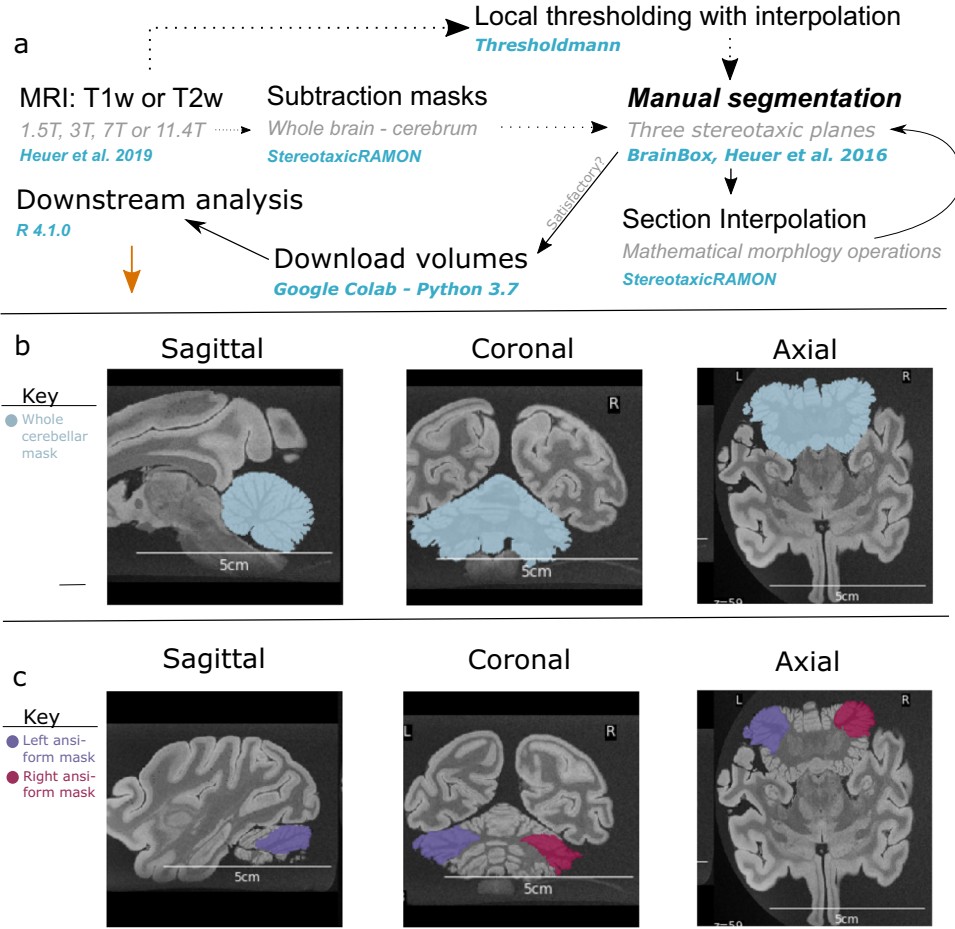

**Fig. 7 Manual segmentation method for primate brains. a** A schematic representation of the manual segmentation pipeline. T1- and T2-weighed (T1w and T2w) MRI scans previously described in Heuer et al.[89] were used to make initial cerebellar masks by subtracting cerebral masks from whole brain masks with StereotaxicRAMON. Some initial cerebellar masks were made through local thresholding and interpolation with Thresholdmann. Cerebellar masks were then uploaded to BrainBox for manual segmentation. Manual segmentation was performed iteratively with interpolation between slices through mathematical morphology operations, until segmentations reached satisfactory quality. Volumes were then downloaded with a custom python script for subsequent analysis in R. **b–c** Example segmentations in stereotaxic planes for the cerebellum and ansiform in the hamadryas baboon. **b** Cerebellar segmentations primarily involved removal of the brain stem, removing erroneously marked tissue and fissures, and reconstructing damaged tissue. **c** Ansiform segmentations additionally involved identification of superior posterior and ansoparamedian fissures, and segmentation between these. Segmentations were made so they did not enter vermal portions of lobule VII. Example masks are color-coded (see legend), but the meaning of the colors themselves is arbitrary. Scale bars are provided for each image.

Human cerebellar and ansiform manual segmentations were compared with automated segmentations from CERES[114] at volbrain.upv.es:

$$Cerebellar\ volume \sim method \qquad (2)$$

and

$$Ansiform\ volume \sim method \qquad (3)$$

Intraclass correlation (ICC) was additionally calculated[137]. ICC of 0.75–0.90 indicates good reliability and 0.90–1.00 excellent reliability[138].

Cerebellar volume was significantly related to method (Pr(>F) < 0.01). ICC was .26 (95% confidence interval (CI): −0.38 < ICC < 0.74). Inspection revealed that manual segmentations were on average 20.4 cm³ (16.2%) larger than CERES segmentations. Ansiform volume was not related to method (Pr(>F) = 0.286), with ICC of 0.84 (95% CI: .51 < ICC < 0.96). Manual volumes were only 2.7 cm³ (6.6%) larger. One specimen differed by 6.9 cm³ (18.2%). It represented an average-sized

cerebellum, unlikely to alter the results substantialy. Full results can be found on GitHub.

Manual ansiform volumes were significantly related to species (Pr(>F) < 0.001). No significant observer effect was detected (Pr(>F) = 0.468), with ICC of .99 (95% CI: 0.92 < ICC < 1.00).

**Neuroanatomical traits**. Traits of interest included cerebellar, cerebral[89], and ansiform volumes (Supplementary Fig. 1). Cerebellar-to-cerebral and ansiform-to-cerebellum ratios were provided merely to show that volume ratios observed within extant primate's CCS[69,101] result directly from allometric scaling (Supplementary Fig. 1). Ratios are strongly confounded by allometry, with hyper-allometries leading to high ratios with increasing size, and vice versa. Volumes could not be corrected for shrinkage. Absolute traits were log₁₀-transformed. Normality was examined for traits with multiple observations (≥4) per species (Fig. 1) by Shapiro–Wilk tests[139] and outliers were visualized on boxplots (± 1.5 IQR from Q1 and Q3). This led to

the exclusion from subsequent analyses of one rhesus and one crab-eating macaque (Supplementary Fig. 2).

*Uncertainties in comparative data.* Primate datasets are rare and generally work on an all-you-can-get-basis. Consequently, covariates known to have substantial influence on brain volumes, such as sex and age, are generally unavailable. In our study, we could not account for sex or age across the sample. Provenances of the data were diverse, potentially leading to differences in the extent of shrinkage of the included brains[82,140]. These sources of measurement variability may compound intraspecific variability in primate brain traits[80] and together lead to large uncertainties in volumetric estimates, a problem well-known to comparative primatologists[81]. These challenges may lead to spurious results when not appropriately addressed[98,117,141].

In our study, the comparison of traits within brains helps partially account for differences between them. Although minor relative to the cross-species scale of investigation, we investigated intraspecific variation where possible. We made sure to exclude any suspected juvenile specimens and examined the distribution of traits among the sexes in chimpanzees and humans (Supplementary Fig. 3). Lastly, we ran several robustness analyses to examine the potential effect of i) different provenances of the data, ii) shrinkage across tissues and brains, and iii) potentially outlying values based on previous literature.

**Statistics and reproducibility.** Consensus phylogenetic trees (the primate family structure) for the full (Fig. 1) and ansiform analyses were obtained from the 10kTrees[142] primate database (10ktrees.nunn-lab.org; version 3). They were constructed from 17 genes and 7 different loci[142] (10ktrees.nunn-lab.org/downloads/10kTrees_Documentation.pdf). We used R version R 4.1.0 for statistical analyses[143].

Extant primate traits are not statistically independent, sharing variable amounts of evolutionary history[144]. Brain traits of species that diverged more recently are more likely to be similar. We detected substantial phylogenetic signal, and correlations between cerebello-cerebral traits illustrated severe collinearity (Fig. 8), demonstrating the importance of accounting for phylogeny and allometry. Therefore, before performing evolutionary analyses, we assessed what evolutionary model best described trait evolution among our species. Fit of extant traits were tested with three common evolutionary models that describe modes of evolution, using Rphylopars[125]. Importantly, the best-supported evolutionary model would inform all subsequent analyses. Models included Brownian Motion (BM)[145], Ornstein-Uhlenbeck (OU)[146] (with single alpha, alpha per trait, or full multivariate alpha matrix), and Early Burst[92]. First, BM describes traits varying randomly over time in direction and extent, leading to differences between species predictable by time since divergence[144]. Second, OU processes behave like BM with a specific trait optimum[147,148]. Lastly, Early Burst processes[92] – which reflect rapid change after adaptive radiations (e.g., adaptation to a nocturnal niche)

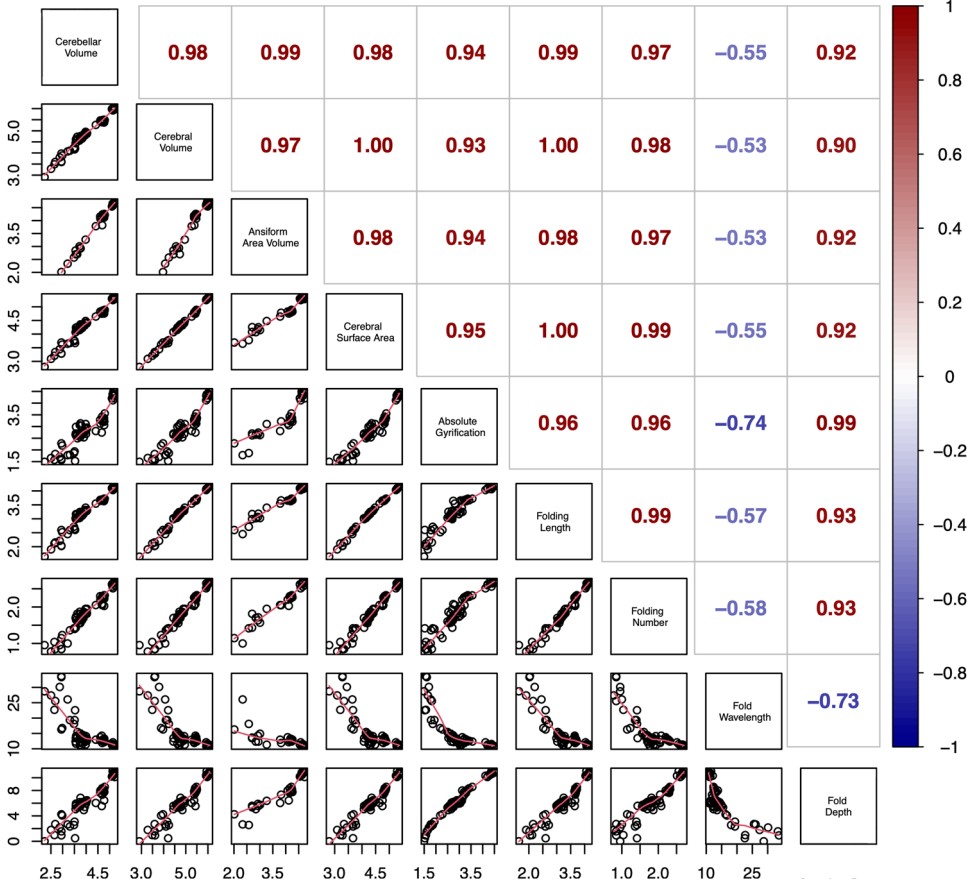

**Fig. 8 Correlations for neuroanatomical measurements.** Volumes recorded in the current study were correlated with the measurements from Heuer et al.[89]. The lower diagonal displays the correlations of $\log_{10}$-transformed variables, which were not corrected for phylogeny and included all specimens. These correlations serve to illustrate the strong collinearity between the traits, which are an expected outcome of allometric scaling. Corresponding $R^2$ coefficients for the correlations can be found on the upper diagonal. Cerebellar, cerebral, and ansiform volumes correlated strongly and positively with all other variables, except fold wavelength, for which a negative correlation was observed. $N = 34$ species.

followed by comparative trait stability – may explain trait evolution. To assess the influence of phylogeny within the data, we chi-squared tested BM with full (Pagel's $\lambda = 1.0$) versus a model without ($\lambda = 0.0$; star-model) phylogenetic signal. Fit was determined by Akaike information criterion (AIC; lower is better)[125,149]. A difference of four to seven points represents significantly less support, while a difference exceeding ten indicates no support for the worse-performing model.

We next mapped extant primate traits back to their ancestral nodes along the phylogenetic tree. This ancestral character estimation (ACE) combines information from extant traits and the tree, finding values for ancestor traits consistent with the best-supported evolutionary model. In other words, if traits evolved following this model, ACEs provide estimations of traits of ancestor species. Visualizing ACEs on the consensus phylogenetic tree facilitates qualitative examination of primate-general and clade-specific evolutionary trends that result from allometric scaling.

To account for trait non-independence and reveal allometric scaling between traits, we performed phylogenetic generalized least squares (PGLS)[144,150] regressions implemented in *nlme*[151]. Regressions were performed for cerebellar volume, regressed on cerebral volume for all species (1) and those with complete (including ansiform) data (2); and ansiform volume regressed on rest of cerebellar volume (3) and cerebral volume (4), all taking species medians.

To better understand potentially divergent patterns, we tested for differences in cerebello-cerebral scaling across two primate bifurcations: strepsirrhines and haplorhines, and apes and non-apes. Using a phylogenetic version of ANCOVA (pANCOVA; github.com/JeroenSmaers/evomap), separate regressions assessed the effects of allowing different intercepts (grade shifts), slopes (evolutionary rate of $y$ relative to $x$), or both. High cerebellar-to-cerebral scaling may be driven by i) large cerebella or ii) small cerebra. It is known that strepsirrhines have smaller brains than haplorhines[86]. Therefore, we performed PGLS for cerebellar and cerebral volumes against body mass data (obtained from a recent compilation[86]) and brain volume data. Strepsirrhine and haplorhine scaling differences were further illustrated by ancestral reconstructions of volumes normalized against brain volume.

Since we had the a priori hypothesis that apes and non-apes would have different cerebellar and ansiform scaling, we additionally calculated $R^2_{likelihood}$[152,153] for the full regressions, and for ape or non-ape regressions separately. Fisher's R-to-Z transformation (cran.r-project.org/web/packages/psych/index.html) determined the fit. To improve comparability with a previous study that reported a grade shift of cerebellar-to-neocortical scaling in apes[35], in a haplorrhine-only dataset, we repeated our PGLS analyses within haplorhines ($N = 25$ species).

*Replication and robustness analyses.* We repeated our main cerebellar–cerebral PGLS and ACE analyses in the Stephan collection[82]. Cerebellar and cerebral volumes were matched to primates in 10kTrees[142]. This led to a partially overlapping (species-wise) dataset of 34 species. This dataset corrected brain area volumes for shrinkage.

To examine the robustness of our main results, we ran a trio of additional analyses. First, we examined data provenance. Most data came from the MNHN, containing brains conserved for several decades. To assess whether unequal shrinkage between these and other data may have driven our results, we ran PGLS on MNHN and non-MNHN datasets separately. Secondly, we specifically examined shrinkage across areas, rerunning cerebellar–cerebral PGLS on 10.000 simulated datasets, in which we introduced random shrinkage factors between cerebellum and

cerebrum of 0.91 to 1.1 (assuming 10% different shrinkage difference). In a connected step, we introduced uncertainties in volumetric estimates, consistent with potentially unaccounted-for shrinkage[82], compounding intraspecific variability[79,80]. We again simulated 10.000 datasets, this time introducing random factors of 0.5 to 2.0 in PGLS. Thirdly, in lieu of intraspecific sample sizes for most species, we excluded potential outlier cerebellar volumes based on the literature[31,68,82,95,115,116], rerunning PGLS in a more conservative subsample. This analysis excluded observations differing more than a factor two from previous observations. Differences between PGLSs were quantified through pANCOVA.

**Reporting summary.** Further information on research design is available in the Nature Portfolio Reporting Summary linked to this article.

## Data availability

Data obtained in the current project, including volumes and ratios, but also intermediate text files and figures, are uploaded to the GitHub repository accompanying this project: github.com/NevMagi/34primates_cerebellum. Input files necessary to replicate our results can also be found here, alongside full-size figures to aid legibility. The source data behind the graphs can be found in Supplementary Data 2. MRI data have been collated from different sources, and published Open Access[89]. The Stephan collection (https://doi.org/10.1159/000155963) was not published Open Access. Therefore, access to the data was requested by the authors through the Copy Clearance Center Marketplace and facilitated through the Central Library of Forschungszentrum Jülich GmbH. Reuse permission for these data may be requested from the publisher.

## Code availability

Custom computer code was used at all stages of analysis. Some of the scripts used are adapted scripts[89]. Full custom code can be obtained from GitHub (github.com/NevMagi/34primates_cerebellum). Code is immediately available for usage under an Apache 2.0 License, which can be found on GitHub. A formal release of the data, as per 01-11-2023 was published to Zenodo (https://doi.org/10.5281/zenodo.10054902)[154]. Full accreditation of R software used in the project can be found in Supplementary Table 2 and their references[155–161] are added at the end of the reference list, unless the software was explicitly mentioned and cited earlier in the manuscript.

PUBINFO

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

## Acknowledgements

The authors would like to thank Dr. Carol MacLeod and Dr. Alexandra de Sousa for enthralling discussions about the cerebellum. We would furthermore like to thank Dr. Joshua Grant for their proofreading of the final version of the manuscript before acceptance. RT and KH are supported by the French Agence Nationale de la Recherche, projects NeuroWebLab (ANR-19-DATA-0025) and DMOBE (ANR-21-CE45-0016). KH received funding from the European Union's Horizon 2020 research and innovation program under the Marie Skłodowska-Curie grant agreement No101033485 (Individual Fellowship). Last, this work was funded in part by Helmholtz Association's Initiative and Networking Fund under the Helmholtz International Lab grant agreement InterLabs-0015, and the Canada First Research Excellence Fund (CFREF Competition 2, 2015–2016), awarded to the Healthy Brains, Healthy Lives initiative at McGill University, through the Helmholtz International BigBrain Analytics and Learning Laboratory (HIBALL), including NM, SBE, and SLV.

## Author contributions

N.M.: Conceptualized the manuscript; wrote and adapted custom computer code; performed manual segmentations; performed analyses; wrote the manuscript; incorporated coauthor revisions manuscript; incorporated reviewers' suggestions manuscript and analysis. R.T.: Provided MRI data; wrote original custom code; wrote used software; provided revision comments manuscript. V.S. and M.A. provided manual segmentations of ansiform areas for 12 and 6 specimens, respectively. S.B.E.: Provided revision comments manuscript; provided funding for the project. K.H.: Provided MRI data; provided semi-automated segmentation and software support; wrote original custom code; wrote used software; provided revision comments manuscript; provided supervision. S.L.V.: Conceptualized the manuscript; provided a subset of primary segmentations; provided revision comments on manual segmentations, the manuscript, and computer code (multiple occasions); provided supervision; provided funding for the project.

## Funding

## Competing interests

The authors declare no competing interests
