## [Peer Review File · Communications Biology]

Reviewers' comments:

Reviewer #1 (Remarks to the Author):

While the classical role of the cerebellum is in motor control, data over the last 20+ years have argued for a role in cognitive function as well, and the recognition that the cerebellum expands over evolution in parallel with the expansion of the cerebral cortex. The goal of the paper seems to be to provide quantitative evidence for the role of the ansiform lobule of the cerebellum in cognitive processes using quantitative species comparisons and correlation of cerebellar anatomy with functional sophistication. While I am very familiar with the literature on cerebellar connectivity and functions, I am much less familiar with the quantitative analysis techniques used in the paper and found it difficult to understand. Is the paper indeed meant for a rather narrow audience who are familiar with the techniques and assumptions behind them? If instead the paper is meant for a broader audience interested in the cerebellum, and brain evolution in general, but less familiar with the analysis techniques I think that more explanation of the techniques would be very helpful. For example please define "manual segmentation." The sentence "Manual segmentation provides ground truth for brain structure volumes" does not help. What is a cerebral mask? What is a cerebellar mask? What are ancestral character estimations? Specifically what traits are considered? A longer explanation of "evolutionary model testing" would be helpful as of ancestral character estimations.

I also have concerns about the measurement of volumes of brain structures from such diverse material. There are a lot of variables around tissue storage that could affect volumes of structures. Are there data showing that, for example, that in tissue stored in formalin the amount of shrinkage of cerebrum and cerebellum is the same? What about differences in brains stored for 6 months vs. 5 years?

I also have concerns about the characteristics of the subjects. There is a lot of individual variability in human brains. One major variable is sex- it is well established that men have bigger brains than women. This could be true of other apes as well: what are there data? Were the numbers of males and females of different species in this analysis balanced? I have not seen an analysis of whether there are male-female differences in the size of the cerebellum as well as of the cortex - that should at least be considered.

Throughout there is use of rather obscure language. For example it is stated that "It likely forms internal models encoding mental representations from separate streams, returning updated predictions". This sentence is impossible to decode without reading the reference. What is "transmodal integration in single granule cells"? Rather than just stating the abstract conclusions of these earlier papers some explanation of what the studies did and what they found would be very helpful. What does "scaffold the development of the brain" mean?

Reviewer #2 (Remarks to the Author):

After manually segmenting 63 cerebella (34 primate species; 9 infraorders) and 30 crura I-II (13 species; 8 infraorders), Magielse and colleagues show that there are constraints, as well as reorganization that may shape the evolution of the primate cerebello-cerebral system. Furthermore, they identify a positive allometry of the ansiform, suggesting a detachment from isometric cerebello-cerebral scaling. Altogether, the paper is sound and appropriate for the Special Issue on 'Brain Evolution.' Some suggestions are included below:

1. For a paper that includes a focus on the ansiform region, it was surprising that the authors do not mention Bolk's 1906 work, as he is credited with the ansiform definition (Glickstein and Voogd, 1995 TiNS for a summary of this work in 69 different species). Indeed, Larsell and Von Bartheldsdorf (Journal of Comparative Neurology; 1941) credit Bolk with this subdivision of the cerebellum. They write: "The lobulus ansiformis and the lobulus paramedianus were established by Bolk ('06) as subdivisions of the mammalian cerebellum."

Also directly relevant for the present paper, Glickstein and Voogd write: "Bolk made a comprehensive comparative study of the mammalian cerebellum, and used his findings to address two related questions: what is the basic organizational plan of the mammalian cerebellum, and can the similarities and differences in cerebellar structure among mammals provide clues about its functions?...Comparative anatomy of the cerebellum seems to have gone out of fashion, but it must return if such questions are to be answered. Differences in the relative size of different parts of the cerebellum await an explanation. That account, when it comes, will tell us more about the functional organization of the human cerebellum, and perhaps lead to a recognition of Bolk as the pathfinder."

Thus, it would be important to link the present findings to this classic work.

See here for a free version of Bolk's

atlas:<https://archive.org/details/dascerebellumde00bolkgoo/page/n49/mode/2up>).

2. The authors do a sufficient job acknowledging the small sample sizes within each species – for example, often only 1 brain in each different species aside from a few such as chimpanzees (9) and humans (10). As such:

a. Some statements could be toned down – for example, what if the 1 sample from the species is an outlier in terms of cerebellar or ansiform size relative to the norm of the species?

b. Additionally, age is not mentioned. What if the 1 brain in one species is from a very young individual, while the brain in another species is from a very old individual? Perhaps these concerns could also be mentioned in the section "Small sample sizes warrant caution."

c. These concerns could also be mentioned in the section "Community-wide sharing accelerates comparative neurosciences." However, the tone of this section is very "preachy" – for example rather than making suggestions, the section often uses words like "need" and "must," which I think does the opposite of what the authors intend. That is, rather than perhaps encouraging collaboration, open conversation, data sharing, and generating databases across research groups, it comes across as the authors telling readers what "must" be done. As such, the authors might consider modifying the tone in order to come closer to what I think the authors are intending.

Reviewer #3 (Remarks to the Author):

This is a potentially useful contribution to the literature on primate brain evolution, and in particular on cortico-cerebellar correlated and divergent evolution. However, I really struggled to take away clear messages from this study. I found the Introduction and Discussion lacking in a clear thread, with little obvious structure and some issues apparently conflated (e.g. differences in slopes versus intercepts in the analysis of apes versus other species) and/or not clearly explained (e.g. how evolution, structure, and function are related, the interpretation of Brownian motion models in relation to adaptive evolution). Some previous work may have been misinterpreted. As a result, the potential impact of this paper is not clear. I think the data are valuable but the analysis, presentation of results and how they relate to previous literature need to be reworked to provide a more cogent treatment.

Introduction

There are problems with the organization of the material and the structure and coherence of the argument. I don't find a clear thread. For example:

- Paragraph 1: tries to cover cerebellar functions, lesion studies, evolution, "involvement in cognitive modules" and functional connectivity. I found it really hard to decipher this paragraph or to understand its main point, given how many different points it touches on without really being clear on any of them (e.g. it isn't clear that involvement in "cognitive modules" – whatever they are - could "explain" functional connectivity – isn't it more the other way around?)
- Paragraph 2. It may be better to begin the Intro at Line 9, and then embed any missing info from the first paragraph in what follows. However, there are still issues: "a direct link between function and cerebello-cerebral connectivity" is argued for by "Cerebellar structural stereotypy and phylogenetic conservation" . I just don't know what this means. Surely it is general neuroscientific logic that argues for a direct link between function and connectivity? Similarly, the meaning of the final sentence of 2nd para is not very clear
- Paragraph 3 focuses on crura I and II/ansiform area, but again switches back and forth rather confusingly between different issues (connections, homology, functions, involvement in "associative networks" - aren't all neural networks associative? - language)
- Paragraph 4 tackles the involvement of cortico-cerebellar networks in primate brain evolution – this is OK, but there's a tension here between conservative patterns and primates being different, which could be more explicitly stated
- Paragraph 5 starts "Previous work has revealed phylogenetic axes of functional organization" – what this means isn't clear. Can you put it more simply? The rest of the paragraph again switches between issues a bit randomly (modularity, functional reorganization, ape-specific patterns, and comparisons of rats, macaques and humans) and doesn't make a clear impression of its main point

So, I think the Introduction needs to be re-written to be much clearer, better organised and in particular to better explain how the current study fits with the existing literature.

Methods

The methods seem generally OK, and it is good to have a new data set to address these issues. It is interesting that cerebellar vol estimation was significantly related to method (manual segmentation>automated)

- Did this interact with phylogeny? Did the methods deviate more for some taxa than others?

Results

Ratios: why report ratios, when we know these are confounded by allometry? This just seems to

confuse matters. This also applies to the ACE estimates – for these, you need to estimate values of $y \sim x$ estimated by your phylogenetic model. I can't see any justification for reporting or using ratios, which are biologically uninterpretable

Apes versus other primates ('No distinct allometries in apes'): here there is a lack of clarity about what you are testing

- what exactly is the model here, is it ANCOVA? Description of the results sounds like you just tested for an interaction effect, rather than main effect – if as stated the interaction is non-significant, this just means the slopes for apes and non-apes are homogenous. This wouldn't contradict Ref 70, in which a phylogenetic ANCOVA was used to show not only that slopes were homogenous but more interestingly that, after controlling for cortical volumes, apes have significantly larger cerebellar volumes, and that cerebellum volume evolved at a significantly higher rate in apes after accounting for rates of change in neocortex volumes. The analysis here doesn't obviously attempt to replicate that.
- In addition, Ref 70 was based on comparisons among haplorrhine primates only, which makes it difficult to compare with the current results. In the current study, I think these analyses may misinterpret the data for strepsirrhines. Strepsirrhines have small relative neocortex sizes (probably because of their predominantly nocturnal habits and reduced visual cortices), which means that apparently high ratios of cerebellum to neocortex in this group has a completely different basis (small neocortex rather than large cerebellum) compared to haplorrhine primates. These effects should be separated. The authors say they "did not test lemur uniqueness to prevent hypothesizing after results are known", but you do need some way of accounting for divergent patterns in the data (e.g. dummy code the taxa in your models)

Discussion:

First, to pick up the comments on the results above

- The Discussion reinforces the impression that scaling (slopes) and grade shifts (intercepts) are being conflated (or at least not clearly distinguished). For example, "Accelerated cerebellar volumetric evolution [70] in great apes may be a response to cerebral expansion" – as noted above, Ref 70 specifically ruled this out (the model accounted for evolutionary rates in cortical volume) . This part of the discussion is confusing
- Again, on high cerebellar-cerebrum ratios in lemurs. In the Discussion this is interpreted as a function of large brain size ("lemurs may have large brains for their bodies, a trait shared with hominoids and especially humans") – this is not true; lemurs are relatively small brained among primates and, as noted above, the ratios alluded to are ambiguous and likely reflect smaller neocortex volumes.

Second, the discussion of Brownian motion – this is confusing and possibly muddled. The authors say that the fact that BM was the best model "at first glance, clashes with notions of adaptive evolution". This is incorrect (as has been appreciated for a long time), The authors then seem to imply as much, but I didn't find this clear. This section of the discussion doesn't therefore seem to add anything useful, but merely confuse the issues

The following material on scaling and on methodological points could be better focussed, and more concise

Once the issues with the analyses/results and interpretation of the data are resolved, the discussion could then be reworked accordingly

We would like to thank the Reviewers for their positive evaluations, constructive comments, and for the opportunity to submit a Revised manuscript. We feel that the comments and suggestions have greatly improved the manuscript. In this covering letter, we outline the steps we have taken to address the suggestions of the Reviewers, in a point-by-point fashion below and highlighting the corresponding changes in the manuscript in yellow. The Revised manuscript may be used to access references, but a bibliography is included at the bottom for quick reference.

Reviewers' comments:

Reviewer #1 (Remarks to the Author):

While the classical role of the cerebellum is in motor control, data over the last 20+ years have argued for a role in cognitive function as well, and the recognition that the cerebellum expands over evolution in parallel with the expansion of the cerebral cortex. The goal of the paper seems to be to provide quantitative evidence for the role of the ansiform lobule of the cerebellum in cognitive processes using quantitative species comparisons and correlation of cerebellar anatomy with functional sophistication.

We thank the Reviewer for the insightful comments, which we have addressed below.

While I am very familiar with the literature on cerebellar connectivity and functions, I am much less familiar with the quantitative analysis techniques used in the paper and found it difficult to understand. Is the paper indeed meant for a rather narrow audience who are familiar with the techniques and assumptions behind them?

We apologize to the Reviewer if the quantitative analysis techniques introduced in the paper were hard to understand due to lack of provided background on techniques and their associated assumptions. Indeed, the paper is intended for a broader scientific audience. Hence, we rewrote the Introduction and hope to have addressed concerns with the complexity of the Methods through further clarifications. We also rewrote the Discussion, to integrate previous literature better in the current paper. Please find the highlighted text in our Revised manuscript to find updates to the Introduction (p.3), Methods (p.6) and Discussion (p.22).

If instead the paper is meant for a broader audience interested in the cerebellum, and brain evolution in general, but less familiar with the analysis techniques I think that more explanation of the techniques would be very helpful. For example please define "manual segmentation." The sentence "Manual segmentation provides ground truth for brain structure volumes" does not help. What is a cerebral mask? What is a cerebellar mask? What are ancestral character estimations? Specifically what traits are considered? A longer explanation of "evolutionary model testing" would be helpful as of ancestral character estimations.

We thank the Reviewer for this remark and have further explained the techniques and assumptions in the current work to make it accessible for a broader audience. We have ensured the Revised manuscript strikes a better balance between brevity and clarity when describing methods and approaches. All the specific points raised in the Reviewer's comments are explained in plain language in the updated manuscript's Methods, and their rationale more clearly explained by adding a clear purpose at the start of every paragraph:

Manual segmentations:

“To provide most accurate volumetric measurements in the highly variable primate data, we manually delineated the areas of interest directly on the MRI data⁸⁸.”

“Manual segmentations (Figure 2b) consisted of brainstem removal, outer cerebellar boundary determination, erroneously marked sulci erasure, and damaged tissue reconstruction.”

Cerebellar/cerebral masks:

“Before this manual segmentation, we obtained initial cerebellar masks - serial 2D-outlines in MRI sections, used to reconstruct the cerebellum in 3D - semi-automatically.”

Ancestral character estimations:

“We next mapped extant primate traits back to their ancestral nodes along the phylogenetic tree. This ancestral character estimation (ACE) combines information from extant traits and the tree, finding values for ancestor traits consistent with the best-supported evolutionary model. In other words, if traits evolved following this model, ACEs provide estimations of traits of ancestor species. Visualizing ACEs on the consensus phylogenetic tree facilitates qualitative examination of primate-general and clade-specific evolutionary trends.”

What traits are under consideration:

“Traits of interest included full cerebellar, cerebral⁸³, and ansiform volumes, and cerebellum-to-cerebrum and ansiform-to-cerebellum ratios (Figure S1). We report ratios for illustrative purposes, facilitating better understanding of underlying scaling and hypothesizing about differences between clades. Ratios are confounded by allometry, with steep allometries leading to high ratios and vice versa. However, we aim to visualize how allometries may lead to reorganizations of fractions observed within extant primate’s CCS^{60,94}, and thus provide ratios alongside our PGLS results”

Evolutionary model testing:

“Extant primate traits are not statistically independent, sharing variable amounts of evolutionary history¹⁰²: brain traits of species that diverged more recently are more likely to be similar. We detected substantial phylogenetic signal and correlations between cerebello-cerebral traits illustrated severe collinearity (Figure 3). Therefore, before performing evolutionary analyses, we assessed what evolutionary model best described trait evolution along the primate tree. Fit of extant traits were thus tested with three common evolutionary models that describe modes of evolution, using Rphylopars¹⁰³. Importantly, the best-supported evolutionary model would inform all subsequent analyses. Models included Brownian Motion (BM)¹⁰⁴, Ornstein-Uhlenbeck (OU)¹⁰⁵ (with single alpha, alpha per trait, or full multivariate alpha matrix), and Early Burst (EB)¹⁰⁶. First, BM describes traits varying randomly over time in direction and extent, leading to differences between species predictable by time since divergence¹⁰². BM does not equal pure genetic drift but may represent weak selection within BM parameters or selection towards specific adaptive trait values, whose distribution is described by BM¹⁰⁷. Second, OU processes behave like BM with a specific trait optimum^{108,109}. Lastly, EB processes¹⁰⁶ – rapid change after adaptive radiations (e.g., adaptation to a nocturnal niche) followed by comparative trait stability – may explain trait evolution. To assess influence of phylogeny within the data, we chi-squared tested BM

with full phylogenetic signal (Pagel's $I=1.0$) versus one without ($I=0.0$; star-model). Fit was determined by Akaike information criterion (AIC; lower is better)^{103,110}. A difference of four to seven points represents significantly less support while a difference exceeding ten indicates no support for the lower-performing model.”

I also have concerns about the measurement of volumes of brain structures from such diverse material. There are a lot of variables around tissue storage that could affect volumes of structures.

We agree with the Reviewer's assessment and are happy to further clarify. Indeed, differences in treatment can lead to differences in shrinkage between brains and within them. Brains mostly come from one collection, the National Museum of Natural History in Paris (MNHN). Exact preservation history may differ among the brains (Herbin et al. 2021). However, all brains have been preserved for several decades (most around a century) in good quality. The age of the specimens is in most cases not known, but assumed to be adult: we excluded outliers based on intraspecific variability for the species where this was possible. Additionally, we reran our PGLS analysis excluding species that had cerebellar volumes more than two times smaller/larger than previously reported volumes. This showed that cerebellar ~ cerebral PGLS results remained intact and even tended more towards hypo-allometry, matching the Stephan dataset analysis that corrected volumes for shrinkage (Figure S9a; and here for easy accessibility):

Figure S9a: Removing potential outliers, identified based on cerebellar volumes from the literature, shows that there are scaling differences in these data (green). Running PGLS with the remaining data (grey), shows a scaling formula highly similar to our main analysis, tending slightly more towards hypo-allometry.

Additionally (and as the above analysis shows), our analyses, focusing on within-brain scaling of different areas of the cerebello-cerebral system, should be relatively less susceptible to shrinkage-related differences across brains. For illustrative purposes, we included ancestral character estimations (ACEs) focusing on ratios between structures, as these are purely within-brain (Figure 4c,e).

We believe that looking at within-brain scaling can help account for the effect of differences across brains. Hanson et al., faced with similar concerns, specifically examined lobular fractions of dolphin cerebella, and their statement in **bold** highlights our shared assumption on shrinkage affecting cerebellar structures similarly. Moreover, in *italics*, they mention that Montie et al. 2010

found little discernible difference in white/grey matter shrinkage in live vs. fixated brains in sea lions.

“Concerns over potential discrepancies between datasets are reduced by the fact that our comparisons are based on relative measurements within each specimen. **It is unlikely that the process of fixation would affect separate lobules differently; rather, it is more probable that processing artifacts created changes in volume and contrast that affected the cerebellum as a whole [bold added].** *Furthermore, a study by Montie, et al. (2010) on the effects of fixation on brain structure volume in sea lions found little discernible difference in volumes of white or gray matter in the cerebrum between scans of live sea lions compared to freshly deceased brains and brains that had undergone fixation [italics added].*”

Are there data showing that, for example, that in tissue stored in formalin the amount of shrinkage of cerebrum and cerebellum is the same?

We thank the Reviewer for this question on tissue-specific shrinkage. We believe this type of shrinkage is important to the implications of our study. Although tissue-specific shrinkage is generally expected to be rather small, especially relative to the scale of investigation (cross-species), it becomes more important since we focus on differences in scaling *within* brains.

Although assumed to be minor, there may be shrinkage differences between grey and white matter, as short-term shrinkage in paraffin appears to differ between cerebral cortex (51%) and white matter (42%) [Kretschmann et al., 1982] in humans. Here, different water-content could drive differences in shrinkage between cerebellum and cerebrum. Indeed, paraffin embedding seems to reduce cerebellar volume by 42% and cerebral volume by 45% [Gellért 1971; Quester and Schröder 1997]. Additional statements on cerebellar/cerebral shrinkage come from Quester and Schröder (1997), and Treff and Kraus (1960). Although interested in differential shrinkage of the brainstem, Quester and Schröder mention:

“As Treff and Kraus (1960) already discovered, this swelling pattern (7% increase in weight) can be less distinct than in other brain structures, as for instance the cerebellum (12% increase in weight)”.

They referred specifically to initial swelling due to formalin fixation. Unfortunately, we were not able to find a copy of *Treff WM, Kraus C. Formalinfixation des Gehirns. Psychiatr Neurol Neurochir 1960;63:116 – 24*, so we will have to take the author's word for the statement.

We believe these differences likely reflect small differences in water-content between areas [Kretschmann et al., 1982]. In general terms, this data does not prove directly useful to provide specific estimates of relative cerebellar (+ansiform) and cerebral shrinkage. It does give us some confidence that if there is a systematic scaling difference between any of the volumes we aimed to measure in our study, it should be minor.

We thus have the following assumptions:

- Across cerebella, cerebra, and ansiforms, shrinkage differences are relatively minute. We think that assuming a 10% difference in shrinkage would be liberal;
- Although more relevant to our analyses (comparing volumes *within* brains) this amount of shrinkage difference should not greatly alter the general picture of our main PGLS.

In order to test our second assumption, that shrinkage of 10% would not alter our main PGLS, we reran PGLS on 10,000 randomly transformed dataframes. Cerebellar, cerebral, and ansiform volumes were multiplied by a random value between 0.91 and 1.1 for each species. We then ran PGLS on these transformed dataframes. The simulation results are included here for accessibility (see also Figure S8c,d,g-j):

Figure S8: Introducing random shrinkage between tissues in our dataset leads to the following distributions of PGLS intercepts and slopes for cerebellum ~ cerebrum (c,d), ansiform area ~ rest of cerebellum (g,h), and ansiform area ~ cerebrum (i,j) regressions. The PGLS analyses center on our main analysis (vertical black lines), and show the amount of variability that could be supported through introducing tissue-specific shrinkage differences of up to 10%. The cerebellum-cerebrum analysis shows a pattern that ranges between a relatively strong hypo-allometry, to a slightly more moderate hyper-allometry. We discuss in the manuscript how our analysis cannot, by itself, be conclusive on cerebellar ~ cerebral scaling, based on previous contradictory results and previously voiced statistical concerns (Wartel et al., 2019; Hooper et al., 2002). However, our ansiform area regressions argue for anywhere from a relatively minor hyper-allometry, to a very strong one (>1.5). We thus report higher confidence in these results.

Changes in manuscript text:

After the section 'Neuroanatomical traits', we add a new section 'Uncertainties in comparative data', voicing potential uncertainties about the data and how we made sure to test for robustness of our results. We decided to group together all potential concerns with volumetric estimates in this section, to give a cogent treatment. This section also serves to show difficulties with comparative data, leading into a more natural and positive discussion of future perspective for the comparative community in 'Community-wide sharing accelerates comparative neurosciences':

“Uncertainties in comparative data

Primate datasets are rare and generally work on an ‘*all-you-can-get*’-basis. Consequently, covariates known to have substantial influence on brain volumes, such as sex and age, are generally unavailable. In our study, we could not account for sex or age. Provenances of the data are diverse, potentially leading to differences in the extent of shrinkage of the included brains^{82,95}. These sources of variability may compound intraspecific variability in primate brain traits⁸⁰ and together lead to large uncertainties in volumetric estimates, a problem well-known to comparative primatologists⁸¹. These challenges may lead to spurious results when not appropriately addressed^{96–98}.

“In our study, the comparison of traits within brains helps partially account for differences between them. Although minor relative to the cross-species scale of investigation, we investigated intraspecific variation where possible, made sure to exclude any suspected juvenile specimens, and examined distribution of traits among sexes in chimpanzees and humans (Figure S4). Lastly, we ran several robustness analyses to examine the potential effect of i. different provenances of the data, ii. shrinkage across tissues and brains, and iii. potentially outlying values based on previous literature. ”

We updated our discussion and conclusions based on the findings in our shrinkage-simulation analysis and discussed its effect in a separate section:

“Accounting for shrinkage

Our study compares brain areas within specimens, helping to minimize influence of variability across brains. Notwithstanding, shrinkage could alter exact scaling formulae and would ideally be accounted for. We assumed intraspecific variability, compounded by measurement error due to unequal shrinkage, may lead to a twofold difference in volumetric estimates^{77,79,80}. Simulating the effect of introducing this volumetric variability in cerebello-cerebral PGLS (Figure S8e,f), we found generally reduced allometries relative to our main PGLS. This result was consistent with analysis of the Stephan dataset, with shrinkage-corrected volumes⁸².

Within-brain shrinkage was expected to play a larger role. Although area-specific shrinkage differences are assumed to be minor⁹⁴, cerebella, cerebra, and ansiforms may scale differently¹⁴⁷, perhaps based on white-matter content¹⁴⁸. Simulating this shrinkage (Figure S8c,d,g-j) revealed that conclusions on cerebello-cerebral scaling may be different depending on how areas have shrunk relative to another. Ansiform hyperscaling was robust to shrinkage. In the future, standard recording of fresh brain weights, and calculation of time-related shrinkage across different brain components may help combat uncertainty in volumetric estimates and potentially contradictory results^{81,97}.”

We increase emphasis on the comparison of our cerebellar ~ cerebral results with previous literature, which is important since single dataset analyses may be somewhat inconclusive, as we also find within our study. We outline the new text here (please find this text highlighted in the Revised manuscript as well):

“To assess effects of potential confounding factors, we ran several robustness analyses, indicating that our main results hold when PGLS is rerun in data i. without potential outliers identified from the literature; ii. from only museum specimens; and iii. accounting for shrinkage differences between brain areas. Introducing shrinkage differences within-brain showed that cerebello-cerebral scaling may range

from rather strong hypo-allometries (~ 0.8) to slightly weaker hyper-allometries (~ 1.1) (Figure S8c,d). Introducing severe artificial shrinkage differences across-brains showed mostly reduced allometries relative to our main PGLS (Figure S8e,f; ~ 0.95), as did reanalysis without literature outliers (S9a; ~ 0.93), and the Stephan replication (Figure S8b; ~ 0.92) but not the MNHN-only PGLS (Figure S8a; ~ 0.98). A recent study reported similarly significant hypo-allometric cerebello-cerebral scaling across mammals (~ 0.92)¹²⁸. Although our results might hint towards a slight cerebello-cerebral hypo-allometry, it becomes clear from these data how sparsely sampled comparative data may lead to contradictory results⁹⁷, especially when considering near-isometric scaling. Our robustness analyses show that providing several complementary views on comparative data may reveal more than small sample p-values.”

What about differences in brains stored for 6 months vs. 5 years?

We thank the Reviewer for this relevant comment. We lack detailed metadata - no (tissue-specific) fresh brain weights are available to us - which does not allow us to incorporate these shrinkage uncertainties decisively. In the Stephan dataset [Stephan et al. 1981], which is one-of-a-kind in many ways, whole-brain shrinkage was calculated by comparing freshly extracted and preserved brain weights. Due to the lack of these measurements in our more diverse (in provenance) data, we could try to assume that preserved tissues shrink roughly to the same extent, but this seems unwarranted.

Nevertheless, we have taken various approaches that, at least in part, should account for the main effect of shrinkage *across* brains. 1. By taking ratios (Figure 4) and 2. by regressing brain areas in a *within*-brain manner (Figure 5), our conclusions should be affected by the between-area, rather than the between-brain (and with that, species and specific preparation) shrinkage. Such a within-brain focus is a regularly used strategy in comparative neuroanatomy, aiming to deal with uncertainties about the data that seem to be inherent to the field of study [Mars et al. 2014; Neubert et al. 2014; Passingham and Smaers 2014; Halley and Krubitzer 2019]. It removes much of the variability related to between-brain shrinkage differences, but also related to overall brain size and its many biological correlates such as sex and age (see also our later response to the question on sex/age).

Our data includes one primary source of long-conserved brains, the National Museum of Natural History in Paris (MNHN). All these brains have been in fluid for long times, and were well past the point of immediate shrinkage. Time-related shrinkage differences between brains should therefore be relatively minor. Unfortunately, not much data exists that examines time-related shrinkage of specific brain tissues in the different solutions that were used (at least, to our knowledge), so it is not possible to test this assumption directly. Soft tissue shrinkage in museum collections is generally found to be fast initially and to then slow down considerably [Hedrick et al. 2018]. This property seems to apply rather similarly to soft-tissues more divergent than just brain areas: brains, livers, lungs, and kidneys show similar temporal volumetric reduction dynamics in mice [Dawood et al. 2021]. Moreover, in mummies, preparation strategy seems to affect soft tissue shrinkage, but age of the mummy seems relatively unimportant [Sydler et al., 2015]. As long as one is not comparing freshly prepared (order of magnitude: hours to days) brains with long-stored ones (years-to-decades; all brains in Paris Collection are preserved for many decades) the factor of time in unequal shrinkage should be relatively minute.

This does leave the comparison between MNHN-brains and brains that may have shrunk less, as one to investigate further. We reran PGLS in data from the MNHN directly against data from other sources (here for accessibility; Figure S8a):

RESULT:

Splitting our main analysis up into MNHN (Paris) and non-MNHN species resulted in 24 MNHN species and 9 non-MNHN species.

We observed the following allometric scaling between cerebellum (y) and cerebrum (x) in PGLS:

MNHN: $y = -0.724 + 0.977x$

Non-MNHN: $y = -0.252 + 0.885x$

For reference, the full 34-species PGLS: $y = -0.627 + 0.955x$

Figure S8a: Splitting up PGLS analysis provided different formulae for MNHN and non-MNHN groups. A scaling difference can be observed when allowing both intercepts and slopes to vary. Our non-MNHN data (grey) scaled at a quite strong hypo-allometric grade, perhaps driven by the smaller sample size which was partly *in vivo*, partly *ex vivo*. This grade may also represent to a large extent actual cerebellar ~ cerebral scaling in this group, which is very ape-heavy. The museum collection data (green) show scaling comparable to our main analysis, albeit slightly steeper. This may be caused by the removal of species with shallower scaling (green). Together, although the MNHN data might provide somewhat different scaling, our shrinkage simulations may explain this. Due to the different species included in both groups, we cannot be conclusive on what drives the non-MNHN scaling difference.

Interpretation:

In contrast to our expectation, the MNHN-only PGLS differed *more* from the Stephan dataset PGLS than did our main analysis. Under the assumption that shrinkage is more similar amongst brains in the MNHN-only data than the full data (and thus likely more similar to the Stephan data in relative cerebellar-cerebral scaling), this points to intraspecific variability being able to drive large part in the difference between the main ($y = -0.627 + 0.955x$) and Stephan PGLSs ($-0.414 + 0.917x$). However, we have insufficient knowledge of the complete preparation of tissues in both datasets, and so cannot assess their exact effects on relative cerebellar-cerebral scaling. It may therefore be possible that the preparation strategies in MNHN vs. Stephan may introduce relative scaling differences between cerebellum and cerebrum. These differences are expected to be extremely minute relative to the level of analysis, and should not alter our main results based on our simulations.

Conclusion: in all robustness analyses, including the replication analysis of the Stephan data, we find hypo-allometries. Our data do not argue for a significant relationship (and thus argue for isometry), whereas the Stephan data does (and thus for hypo-allometry). This points to the role of intraspecific variability in determining the exact scaling formulae, as is somewhat expected.

However, despite our general results being robust, it remains difficult to compare individual brains across different preparations, of which we do not have complete knowledge in our data. We included *in vivo* brains in our studies, including humans, chimpanzees, the eastern gorilla, bonobo, and gibbon. These volumes represent unshrunk values. Despite this, we do not notice these species deviation from our trends more than would be expected (Figures 5a,c from the Revised manuscript):

Figure 5a,c: PGLS analysis in the original manuscript did not show indication that *in vivo* brains scaled differently from the other brains in analysis.

Manuscript changes:

Based on the MNHN/non-MNHN analyses, we conclude that it is important that specific deviations from allometry (or exact volumes) for a species are examined with caution. We dedicate a new section to this shortcoming/crucial note in the Revised manuscript:

“Volumes and ratios should not be taken as representative of a species

First, it is important to consider that intraspecific variability in any brain volume is substantial^{77,79,80}. Small intraspecific samples can thus not be representative of the range of natural variability. Although we show

robustness of our general conclusions at the primate-level, this means that deviations for individual species cannot be used to infer adaptation, since precise volumes and ratios may differ. We show variability of absolute and relative volumes across our data to be substantial^{78–80}, as they include specimens across sexes and adult ages.”

..and hope that this in combination with the ancestral character estimations (and Figure 6 provide optimal understanding of the underlying data:

“We report ratios for illustrative purposes, facilitating better understanding of underlying scaling and hypothesizing about differences between clades. Ratios are confounded by allometry, with steep allometries leading to high ratios and vice versa. However, we aim to visualize how allometries may lead to reorganizations of fractions observed within extant primate’s CCS^{60,94}, and thus provide ratios alongside our PGLS results.”

The point that volumes are not to be taken as **the** volume for a species can also be made by comparing cerebellar volumes with those previously reported in the Stephan collection [Stephan et al., 1981] and other datasets (see manuscript). Over two-fold differences exist in cerebellar volumetric estimates for some species. These effects may be driven in a very substantial part by both i. natural intraspecific variability and ii. shrinkage specific to preparation of the brain. In the Revised manuscript, we make explicit this challenge (in comparative primatology) and speculate that most of it may come from intraspecific variability (see next section for our discussion on this topic):

“Primate datasets are rare and generally work on an ‘*all-you-can-get*’-basis. Consequently, covariates known to have substantial influence on brain volumes, such as sex and age, are generally unavailable. In our study, we could not account for sex or age. Provenances of the data are diverse, potentially leading to differences in the extent of shrinkage of the included brains^{82,96}. These sources of variability may compound intraspecific variability in primate brain traits⁸⁰ and together lead to large uncertainties in volumetric estimates, a problem well-known to comparative primatologists⁸¹. These challenges may lead to spurious results when not appropriately addressed^{97–99}.”

+

“Volumes and ratios should not be taken as representative of a species

First, it is important to consider that intraspecific variability in any brain volume is substantial^{77,79,80}. Small intraspecific samples can thus not be representative of the range of natural variability. Although we show robustness of our general conclusions at the primate-level, this means that deviations for individual species cannot be used to infer adaptation, since precise volumes and ratios may differ. We show variability of absolute and relative volumes across our data to be substantial^{78–80}, as they include specimens across sexes and adult ages.”

..and how we believe that combining datasets and collaborative science may help overcome key challenges of evolutionary neurosciences:

“Community-wide sharing accelerates comparative neurosciences

Systematic analysis of larger datasets containing both many species and sizable intraspecific samples might resolve previous controversies^{81,97,149,150}. Creating community-wide guidelines for data collection and statistical handling can facilitate integration of data from primate studies^{34,151–153}. Access to intraspecific variability could combat uncertainties about evolutionary scaling^{78,81}, and statistical methods

to incorporate intraspecific samples already exist^{103,113,154}. To further data integration, primate databases can be used to earmark primate brains with unique identifiers, linked to relevant metadata including sex, age, body weight, captivity status, data availability, and inclusion in specific studies. Lastly, museum collections can greatly expand comparative samples¹⁵⁵.”

We thus mention importance of considering potential incongruence between data provenances/unequal shrinkages in the section ‘Uncertainties in comparative data’, and discuss/show our PGLS results in the robustness analysis sections of the Results/Discussion in the Revised manuscript, as well as in Figure S8a,b.

We additionally explain how partially removing the effect of shrinkage across brain preparations led to a PGLS that more closely resembled the results in the Stephan collection and a previous cerebello-cerebral scaling study in a wider mammal sample [Heuer et al. 2022]. Both studies include brains prepared in systematic ways. It is interesting that, if anything, removing shrinkage from brains might produce results more consistent with the literature. We speculate on potential controversy around exact scaling:

“To assess effects of potential confounding factors, we ran several robustness analyses, indicating that our main results hold when PGLS is rerun in data i. without potential outliers identified from the literature; ii. from only museum specimens; and iii. accounting for shrinkage differences between brain areas. Introducing shrinkage differences within-brain showed that cerebello-cerebral scaling may range from rather strong hypo-allometries (~.8) to slightly weaker hyper-allometries (~1.1) (Figure S8c,d). Introducing severe artificial shrinkage differences across-brains showed mostly reduced allometries relative to our main PGLS (Figure S8e,f; ~.95), as did reanalysis without literature outliers (S9a; ~.93), and the Stephan replication (Figure S8b; ~.92) but not the MNHN-only PGLS (Figure S8a; ~.98). A recent study reported similarly significant hypo-allometric cerebello-cerebral scaling across mammals (~0.92)¹²⁸. Although our results might hint towards a slight cerebello-cerebral hypo-allometry, it becomes clear from these data how sparsely sampled comparative data may lead to contradictory results⁹⁷, especially when considering near-isometric scaling. Our robustness analyses show that providing several complementary views on comparative data may reveal more than small sample p-values.”

I also have concerns about the characteristics of the subjects. There is a lot of individual variability in human brains. One major variable is sex- it is well established that men have bigger brains than women. This could be true of other apes as well: what are there data? Were the numbers of males and females of different species in this analysis balanced? I have not seen an analysis of whether there are male-female differences in the size of the cerebellum as well as of the cortex - that should at least be considered.

We thank the Reviewer for this comment, which is relevant since sex (and age) introduce(s) great variability in volumetric estimates. Therefore, it is important to consider how our conclusions on primate-general scaling can hold in the face of this variability. We do not aim to make conclusive statements on specific species, but due to the scale of investigation, primate-general trends should hold. We have made this more explicit in the manuscript, making sure that the reader knows to i. not consider volumes as ‘the’ volume for a species; and ii. take uncertainty in conclusions, especially on species-specific volumes/ratios/deviations, into account:

“We report ratios for illustrative purposes, facilitating better understanding of underlying scaling and hypothesizing about differences between clades. Ratios are confounded by allometry, with steep allometries leading to high ratios and vice versa. However, we aim to visualize how allometries may lead to reorganizations of fractions observed within extant primate’s CCS^{60,94}, and thus provide ratios alongside our PGLS results.”

“Volumes and ratios should not be taken as representative of a species

First, it is important to consider that intraspecific variability in any brain volume is substantial^{77,79,80}. Small intraspecific samples can thus not be representative of the range of natural variability. Although we show robustness of our general conclusions at the primate-level, this means that deviations for individual species cannot be used to infer adaptation, since precise volumes and ratios may differ. We show variability of absolute and relative volumes across our data to be substantial^{78–80}, as they include specimens across sexes and adult ages.”

In other words, the reader should know that although there are strong indications that primate scaling of cerebellum-to-cerebrum volume may be isometric-to-hypo-allometric and ansiform area scaling hyper-allometric, individual data points do not represent the large variability within the species’ population. We aimed to minimize one primary driver of variation by performing manual segmentations. By reporting ratios and within-brain regressions, we furthermore aimed to remove the effect of between-brain differences, as evidenced by the general similarity of our main analyses (Figure 5) and analysis in the Stephan data (Figure 8) (despite very different underlying data).

We assume that the range of intraspecific variability in brain traits based on sex and age is similar between non-human primates and humans [Croxon et al. 2018]. This means that there may be a fold-difference in brain volumes of up to ~1.8 [Giedd et al. 2015]. This fold-difference should translate quite directly to the fold-difference between areas (i.e., cerebellar, cerebral, and ansiform) volumes (although every species may have different variability per area [Croxon et al. 2018]). In the manuscript we thus mention the large range of intraspecific variability that our data cannot directly account for within the ‘Uncertainties in comparative data’ subsection, citing the literature above:

“Uncertainties in comparative data

Primate datasets are rare and generally work on an ‘all-you-can-get’-basis. Consequently, covariates known to have substantial influence on brain volumes, such as sex and age, are generally unavailable. In our study, we could not account for sex or age. Provenances of the data are diverse, potentially leading to differences in the extent of shrinkage of the included brains^{82,95}. These sources of variability may compound intraspecific variability in primate brain traits⁸⁰ and together lead to large uncertainties in volumetric estimates, a problem well-known to comparative primatologists⁸¹. These challenges may lead to spurious results when not appropriately addressed^{96–98}.

In our study, the comparison of traits within brains helps partially account for differences between them. Although minor relative to the cross-species scale of investigation, we investigated intraspecific variation where possible, made sure to exclude any suspected juvenile specimens, and examined distribution of traits among sexes in chimpanzees and humans (Figure S4). Lastly, we ran several robustness analyses to examine the potential effect of i. different provenances of the data, ii. shrinkage across tissues and brains, and iii. potentially outlying values based on previous literature. We

took great care not to overstate the implications of our results, acknowledging where replication may be necessary to support findings.”

In the manuscript we mention the following:

AGE

Only brains that were expected to be adult brains were included in the study. Clear outliers based on intraspecific volumes were excluded where possible:

“Normality was examined for traits with multiple observations (³⁴) per species (Figure 1) by Shapiro-Wilk tests⁹⁴ and outliers were visualized on boxplots (± 1.5 IQR from Q1 and Q3). This led to the exclusion from subsequent analyses of one rhesus and one crab-eating macaque (Figure S2).”

Connected, we reran PGLS excluding potential outliers based on the literature (Figure S9a,b). Figure S9a is provided as response to the earlier question on the museum collection and shrinkage.

SEX

In our data, the human (5F;5M) and chimpanzee (4F;5M) data (both higher N) were balanced in terms of sex, and selected to be representative of natural variability in brain size. Despite this, volumes fit well with the PGLS slope created from mostly N=1 species:

“The ansiform was observed to consistently take up a disproportionately large fraction of the cerebellum across humans and chimpanzees, fitting well with the main PGLS slope.”

..and resemble prior data on the ansiform:

“Our data generally resemble prior observations for mean ansiform volumes in humans (current study: 43.94 cm³; Balsters et al.: 53.65 cm³; Makris et al.: 43.01 cm³), intraspecific variability (5.00 cm³; 8.01 cm³; 6.38cm³), and relative ansiform-to-cerebellum fraction (30.76%; 36.51%; 29.58%)^{60,134}. Reanalysis of ansiform scaling in larger datasets will be necessary to confirm our observations as they may be partially driven by an underpowered sample and influenced by specimen-specific idiosyncrasies^{81,96,98}.”

We also added Figure S4, that shows spread of traits per sex for chimpanzees and humans. It shows although in the sample traits may vary by sex, that ratios (within-brain measurements) appear less sensitive to its influence (with sex not being represented in N=1 species):

Figure S4: Spread of traits with sex.

In-text we added:

“Moreover, we examined distribution of traits among sexes in chimpanzees and humans (Figure S4).”

+

“Human absolute volumes were systematically higher in males, but ratios were practically identical between the sexes. Chimpanzee data was generally similar between sexes (Figure S4).”

Anything available

For most species, we operated under more of an ‘*anything-you-can-get*’-strategy. We mention this more explicitly in the manuscript:

“Methodological considerations

The ‘*anything-you-can-get*’-basis of comparative studies warrants some consideration. For example, most species in the current work are represented by one specimen, so that much of natural variation remains unaccounted for.”

+

“We show variability of absolute and relative volumes across our data to be substantial^{78–80}, as they include specimens across sexes and adult ages.”

We mention possible analytical / scientific solutions to advance our understanding of evolution:

“Community-wide sharing accelerates comparative neurosciences

Systematic analysis of larger datasets containing both many species and sizable intraspecific samples might resolve previous controversies^{81,97,149,150}. Creating community-wide guidelines for data collection and statistical handling can facilitate integration of data from primate studies^{34,151–153}. Access to intraspecific variability could combat uncertainties about evolutionary scaling^{78,81}, and statistical methods to incorporate intraspecific samples already exist^{103,113,154}. To further data integration, primate databases can be used to earmark primate brains with unique identifiers, linked to relevant metadata including sex, age, body weight, captivity status, data availability, and inclusion in specific studies. Lastly, museum collections can greatly expand comparative samples¹⁵⁵.”

Lastly, to examine if results of PGLS may hold despite intraspecific variability, we ran another simulation analysis. Here, we assumed that our volumes may be anywhere from two times larger/smaller: a rather extreme assumption even when combining natural brain variability [Croxson et al. 2018; Giedd et al. 2015] and potential shrinkage differences in brains. Practically, we multiplied every volume in our data by a factor between .5 and 2.0, did this 10.000 times, running PGLS on all transformed datasets. Results are presented below (and in Figure S8e,f):

Figure S8e,f: Running 10.000 simulations, assuming a combination of intraspecific variability + shrinkage between brains (here together called interspecific variability) amounting to a variance of a factor of .5-to-2.0. Our PGLS results (black vertical line) are within the normal expectation of the simulations’ distribution. Generally, the results argue that introducing interspecific variance may be most likely to provide stronger hypo-allometries than our main PGLS did. This is consistent with the Stephan data, robustness analysis with literature-outliers removed (Figure S9a), and mammal cerebellar ~ cerebral scaling (Heuer et al. 2022)

We discuss in the manuscript how together our results may hint towards hypo-allometry, but also how contradictions could be lifted by larger datasets:

“Isometric to hypo-allometric cerebello-cerebral scaling

Our main PGLS supports primate isometric cerebello-cerebral^{145,46,87,118,128} evolution, tending towards hypo-allometry (Figure 5a; ~.95). To assess effects of potential confounding factors, we ran several

robustness analyses, indicating that our main results hold when PGLS is rerun in data i. without potential outliers identified from the literature; ii. from only museum specimens; and iii. accounting for shrinkage differences between brain areas. Introducing shrinkage differences within-brain showed that cerebello-cerebral scaling may range from rather strong hypo-allometries (~ 0.8) to slightly weaker hyper-allometries (~ 1.1) (Figure S8c,d). Introducing severe artificial shrinkage differences across-brains showed mostly reduced allometries relative to our main PGLS (Figure S8e,f; ~ 0.95), as did reanalysis without literature outliers (S9a; ~ 0.93), and the Stephan replication (Figure S8b; ~ 0.92) but not the MNHN-only PGLS (Figure S8a; ~ 0.98). A recent study reported similarly significant hypo-allometric cerebello-cerebral scaling across mammals (~ 0.92)¹²⁸. Although our results might hint towards a slight cerebello-cerebral hypo-allometry, it becomes clear from these data how sparsely sampled comparative data may lead to contradictory results⁹⁷, especially when considering near-isometric scaling. Our robustness analyses show that providing several complementary views on comparative data may reveal more than small sample p-values.”

Throughout there is use of rather obscure language. For example it is stated that “It likely forms internal models encoding mental representations from separate streams, returning updated predictions”. This sentence is impossible to decode without reading the reference. What is “transmodal integration in single granule cells”? Rather than just stating the abstract conclusions of these earlier papers some explanation of what the studies did and what they found would be very helpful. What does “scaffold the development of the brain” mean?

We thank the Reviewer for pointing out examples of unnecessarily contrived language used in the previous manuscript. In the Revised manuscript, we have made sure to connect the different domains we touch on more clearly, and in more plain language. The specific examples provided by the Reviewer are no longer present in the Revised manuscript.

Reviewer #2 (Remarks to the Author):

After manually segmenting 63 cerebella (34 primate species; 9 infraorders) and 30 crura I-II (13 species; 8 infraorders), Magielse and colleagues show that there are constraints, as well as reorganization that may shape the evolution of the primate cerebello-cerebral system. Furthermore, they identify a positive allometry of the ansiform, suggesting a detachment from isometric cerebello-cerebral scaling. Altogether, the paper is sound and appropriate for the Special Issue on 'Brain Evolution.' Some suggestions are included below:

We thank the Reviewer for appreciation of our work and the helpful comments and suggestions, which we have addressed below.

1. For a paper that includes a focus on the ansiform region, it was surprising that the authors do not mention Bolk's 1906 work, as he is credited with the ansiform definition (Glickstein and Voogd, 1995 TiNS for a summary of this work in 69 different species). Indeed, Larsell and Von Berthelsdorf (Journal of Comparative Neurology; 1941) credit Bolk with this subdivision of the cerebellum. They write: "The lobulus ansiformis and the lobulus paramedianus were established by Bolk ('06) as subdivisions of the mammalian cerebellum."

Also directly relevant for the present paper, Glickstein and Voogd write: "Bolk made a comprehensive comparative study of the mammalian cerebellum, and used his findings to address two related questions: what is the basic organizational plan of the mammalian cerebellum, and can the similarities and differences in cerebellar structure among mammals provide clues about its functions?...Comparative anatomy of the cerebellum seems to have gone out of fashion, but it must return if such questions are to be answered. Differences in the relative size of different parts of the cerebellum await an explanation. That account, when it comes, will tell us more about the functional organization of the human cerebellum, and perhaps lead to a recognition of Bolk as the pathfinder."

Thus, it would be important to link the present findings to this classic work.

See here for a free version of Bolk's atlas:<https://archive.org/details/dascerebellumde00bolkgoog/page/n49/mode/2up>).

We thank the Reviewer for pointing our attention to Louis Bolk's important work. Although we were somewhat familiar with his 1906 work, we were unaware that the original definition of the ansiform area came from his pen. We have now added mention to his work on lobular definitions in the manuscript. We added the following passages:

"Already in 1906, Louis Bolk asked the question of how large-scale cerebellar structure and (evolutionary) reorganization may reveal cerebellar function⁴¹⁻⁴³."

+

"The large lateral cerebellar crura I-II area is of elevated interest in primates. A cross-species homolog of human crura I-II has been referred to as the ansiform area^{56,57}, a term accredited to Louis Bolk's

work⁴¹ on the mammal cerebellum where he first defined cerebellar lobular terminology⁴³. We adopt the term 'ansiform' here to refer to this area."

2. The authors do a sufficient job acknowledging the small sample sizes within each species – for example, often only 1 brain in each different species aside from a few such as chimpanzees (9) and humans (10). As such:

a. Some statements could be toned down – for example, what if the 1 sample from the species is an outlier in terms of cerebellar or ansiform size relative to the norm of the species?

We thank the Reviewer for the suggestion! The point to tone down statements (considering the variability the data is not able to describe), is well taken and is integrally incorporated in the Revised manuscript. In the Revised manuscript we have made clearer how we do not aim to make statements on deviation from allometry for specific species:

"Volumes and ratios should not be taken as representative of a species

First, it is important to consider that intraspecific variability in any brain volume is substantial^{77,79,80}. Small intraspecific samples can thus not be representative of the range of natural variability. Although we show robustness of our general conclusions at the primate-level, this means that deviations for individual species cannot be used to infer adaptation, since precise volumes and ratios may differ. We show variability of absolute and relative volumes across our data to be substantial^{78–80}, as they include specimens across sexes and adult ages."

By spending additional effort on segmenting all human and chimpanzee ansiform areas, we aimed to illustrate that the general conclusions that we draw - concerning ansiform area hyperscaling and this being a primate-general trend - hold regardless of the variability in the trait. We are aware of and acknowledge the massive intraspecific variability of any brain trait (e.g., a 1.7-to-1.8 factor between smallest to largest humans brains [Giedd et al. 2015]) and the lack of knowledge of metadata in this sample, that likely drives a large part of this variability (i.e., sex, age). However, chimpanzees and humans do not appear to take unusual positions in our PGLS (Figure 5), despite being sex-balanced, and including some intraspecific variability into their median value.

On the scale of our investigation, namely between species, the importance of intraspecific variability is somewhat more minute. For all absolute traits that we were able to examine in several specimens for a species, we found no evidence of overlap between species. For example, the human ansiform area volume actually overlaps with the chimpanzee's cerebellar volume rather than its ansiform area volume. This does not provide a direct solution to the Reviewers' concern, as the problem lies more with the species that have just one observation, and that may be much closer together in size. It is inherently impossible to know within this sample to what extent the specific (N=1) specimens are representative of the species. Please also find our response to Reviewer 1, where we reran PGLS after removing potential outliers that we identified based on cerebellar volumes from the literature (Figure S9a,b).

We used two guiding assumptions: i. variability in our volumes varies within ranges on the order of magnitude of the largest described ranges in brain traits (max. ~1.8x in humans [Giedd et al. 2015], and similar in non-human primates [Croxson et al. 2018]); and ii. the distribution of our specimen measurements relative to their population values are normally distributed. Together, we

have reasonable certainty in the significant hyper-allometric slope that we describe for the ansiform area, while the non-significant hypo-allometric slope for cerebellar-to-cerebral volume is less certain. Please also see our response to reviewer 1, where we ran simulations testing the robustness of our results when subject to different sources of variability (Figure S8).

Extra confidence comes from the fact that the regression based on many N=1 species approximates humans and chimpanzees' median values (with higher N, mixed sex) very well (especially in the ansiform area analyses):

“Rerunning PGLS in data excluding potential outlying cerebellar volumes based on the literature^{45,59,82,117–119} (Figure S9) revealed that the main PGLS was robust, and if anything, overestimated cerebello-cerebral scaling. Nonetheless, discrepancies between cerebellar volumes across studies show impact of intraspecific variation.

Reassuringly, the ansiform was observed to consistently take up a large fraction of the cerebellum across humans and chimpanzees, fitting well with the main PGLS slope.”

What the data do not give us certainty on, is exact scaling in these primate species, let alone all primates. We also feel the data is inappropriate to make any conclusions on species-specific deviations from the main PGLS trend, which would normally facilitate at least speculation on adaptive variations on this primate general trend. We have refrained from making conclusive statements on species, and make more explicit for the reader that this is inappropriate:

“First, it is important to consider that intraspecific variability in any brain volume is substantial^{77,79,80}. Small intraspecific samples can thus not be representative of the range of natural variability. Although we show robustness of our general conclusions at the primate-level, this means that deviations for individual species cannot be used to infer adaptation, since precise volumes and ratios may differ. We show variability of absolute and relative volumes across our data to be substantial^{78–80}, as they include specimens across sexes and adult ages.”

To provide the best understanding of the underlying data, we reported ratios. These show relative changes of absolute traits across clades that may be characteristic for groups of species. Because ratios come from within-brain measurements, and can be plotted to the phylogenetic tree, we can visualize potential dynamics of relative scaling following BM. Two groups stand out qualitatively: apes and lemurs. We focus our interpretation at this level, instead of the species-level, where we have more uncertainty. Figure 4 from the original (and current) manuscript, and Figure 6 and Figure S7 support this clade-level interpretation.

Like for PGLS deviations, it is important to not overinterpret ratios that come from just one specimen, as we show that the range of these values is substantial. We believe that the fact that apes and lemurs show consistently high cerebellar-to-cerebral scaling relative to the primate sample, is most telling. Certainly, the consistent position of humans and chimpanzees, who were more deeply sampled, provides confidence.

We aimed to have the revised manuscript reflect these assumptions more explicitly. Additionally, we stated explicitly that the ratios and PGLS-deviations need to be cautiously interpreted and cannot be taken to directly support a species' adaptive evolution. What they can do, however, is visualize primate trends that lead to testable hypotheses in (future) datasets appropriate for testing species' deviations from the general primate allometry. We also make more explicit our

guiding assumptions, and that regardless of unaccounted-for intraspecific variation or variation in the rest of the primate tree, the general principle should hold. This arguments are supported by our robustness analysis as described in:

“To examine robustness of our main results, we ran a trio of additional analyses. First, we examined data provenance. Most data come from the MNHN, containing brains conserved for several decades. To assess whether unequal shrinkage between these and other data may have driven our results, we ran PGLS on MNHN and non-MNHN datasets separately. Secondly, we specifically examined shrinkage across tissues, rerunning cerebellar ~ cerebral PGLS on 10.000 simulated datasets, in which we introduced random shrinkage factors between cerebellum and cerebrum of between .91 and 1.1 (assuming maximally 10% different shrinkage between tissues). Connected, we examined how introducing uncertainties in volumetric estimates consistent with potentially unaccounted for shrinkage and intraspecific variability may affect PGLS results. We again simulated 10.000 datasets, this time introducing random factors of .5 to 2.0. Thirdly, in lieu of intraspecific sample sizes for most species, we excluded potential outlier cerebellar volumes based on the literature^{45,59,82,117–119}, rerunning PGLS in a more conservative subsample.”

b. Additionally, age is not mentioned. What if the 1 brain in one species is from a very young individual, while the brain in another species is from a very old individual? Perhaps these concerns could also be mentioned in the section “Small sample sizes warrant caution.”

We thank the Reviewer and agree that age, along with sex, are the primary drivers of volumetric variability within species. Please see the response to point a. for our main answer to this question. We made sure to check, where possible, for outlier values clearly indicative of specimens unrepresentative of adult individuals of the species. This concern, especially in N=1 species, has been made more explicit. However, our PGLS analysis without potential outliers based on the literature (Figure S9a,b) provides confidence in the isometric-tohypo-allometric relationship for cerebellum ~ cerebrum, as well as hyper-allometric ansiform scaling.

We agreed with the Reviewer that the “Small sample sizes warrant caution” was an appropriate place to voice this limitation. We reorganized this section into a section called “*Volumes and ratios should not be taken as representative of a species*”:

“Volumes and ratios should not be taken as representative of a species

First, it is important to consider that intraspecific variability in any brain volume is substantial^{77,79,80}. Small intraspecific samples can thus not be representative of the range of natural variability. Although we show robustness of our general conclusions at the primate-level, this means that deviations for individual species cannot be used to infer adaptation, since precise volumes and ratios may differ. We show variability of absolute and relative volumes across our data to be substantial^{78–80}, as they include specimens across sexes and adult ages.”

c. These concerns could also be mentioned in the section “Community-wide sharing accelerates comparative neurosciences.” However, the tone of this section is very “preachy” – for example rather than making suggestions, the section often uses words like “need” and “must,” which I think does the opposite of what the authors intend. That is, rather than perhaps encouraging collaboration, open conversation, data sharing, and generating databases across research groups, it comes across as the authors telling

readers what “must” be done. As such, the authors might consider modifying the tone in order to come closer to what I think the authors are intending.

We thank the Reviewer for this critical remark, we did not intend the section to have this tone. To make this section more stimulating, we have now removed words such as ‘need’ and ‘must’ from the section. We discuss how community-wide sharing can be the solution to the concerns in our (and others’) data:

“Community-wide sharing accelerates comparative neurosciences

Systematic analysis of larger datasets containing both many species and sizable intraspecific samples might resolve previous controversies^{81,97,149,150}. Creating community-wide guidelines for data collection and statistical handling can facilitate integration of data from primate studies^{34,151–153}. Access to intraspecific variability could combat uncertainties about evolutionary scaling^{78,81}, and statistical methods to incorporate intraspecific samples already exist^{103,113,154}. To further data integration, primate databases can be used to earmark primate brains with unique identifiers, linked to relevant metadata including sex, age, body weight, captivity status, data availability, and inclusion in specific studies. Lastly, museum collections can greatly expand comparative samples¹⁵⁵.”

Reviewer #3 (Remarks to the Author):

This is a potentially useful contribution to the literature on primate brain evolution, and in particular on cortico-cerebellar correlated and divergent evolution.

We thank the Reviewer for the acknowledgement of contribution of our work to the literature, and the helpful comments and suggestions which we have addressed below. We feel that they have massively improved the manuscript by clearing up previous unclarities and inconsistencies.

However, I really struggled to take away clear messages from this study. I found the Introduction and Discussion lacking in a clear thread, with little obvious structure and some issues apparently conflated (e.g. differences in slopes versus intercepts in the analysis of apes versus other species) and/or not clearly explained (e.g. how evolution, structure, and function are related, the interpretation of Brownian motion models in relation to adaptive evolution). Some previous work may have been misinterpreted. As a result, the potential impact of this paper is not clear. I think the data are valuable but the analysis, presentation of results and how they relate to previous literature need to be reworked to provide a more cogent treatment.

We thank the Reviewer for raising these points and apologize for any unclarity. We have reworked the Introduction to make the motivation and background of the current work more accessible and organized. Additionally, more emphasis was placed on describing the relation of its results to previous literature. To better represent the meaning of our Results, we also made much more explicit the limitations and assumptions of our methods/analyses.

Introduction

There are problems with the organization of the material and the structure and coherence of the argument. I don't find a clear thread. For example:

We decided to rewrite the Introduction to get to the point more quickly and enhance clarity. Additionally, every paragraph within it now has a clear focus:

- Definition of area of investigation (cerebellar role in brain function)
- Pertinent literature to orient the reader (previous primate evolutionary cerebellar studies)
- Definition of specific focus (reorganizations of cerebello-cerebral system)
 - Comparative findings in ansiform
 - Functional relevance of ansiform
- Method and principal results of the current study

- Paragraph 1: tries to cover cerebellar functions, lesion studies, evolution, “involvement in cognitive modules” and functional connectivity. I found it really hard to decipher this paragraph or to understand its main point, given how many different points it touches on without really being clear on any of them (e.g. it isn't clear that involvement in “cognitive modules” – whatever they are - could “explain” functional connectivity – isn't it more the other way around?)

We agree with the Reviewer that this (and really all paragraphs) was(/were) unnecessarily confusing. We thus decided to have this paragraph simply aim to describe the functional diversity of the cerebellum and the notion that the cerebellum may play a role in many domains in a similar manner.

- Paragraph 2. It may be better to begin the Intro at Line 9, and then embed any missing info from the first paragraph in what follows. However, there are still issues: “a direct link between function and cerebello-cerebral connectivity” is argued for by “Cerebellar structural stereotypy and phylogenetic conservation” . I just don’t know what this means. Surely it is general neuroscientific logic that argues for a direct link between function and connectivity? Similarly, the meaning of the final sentence of 2nd para is not very clear

We agree with the Reviewer that the original paragraph misrepresented what we intended to say. Our new introduction mentions how the cerebellum is an especially interesting structure for large scale evolutionary studies, since the cytoarchitecture is highly organized/ similar across the entire cortex and is phylogenetically conserved - as is the connectional system it is involved in. That means that volumetric variation and reorganization can be quite directly linked to potential (selection on the) functions of specific cerebellar areas:

From paragraph 2:

“Already in 1906, Louis Bolk asked the question of how large-scale cerebellar structure and (evolutionary) reorganization may reveal cerebellar function^{41–43}. This question is still as relevant today and has motivated evolutionary studies aiming to examine how the cerebellum has expanded and reorganized: the cerebellum likely has close mapping between connectivity and function.”

- Paragraph 3 focuses on crura I and II/ansiform area, but again switches back and forth rather confusingly between different issues (connections, homology, functions, involvement in “associative networks” - aren’t all neural networks associative? - language)

We acknowledge the original switching between a wide range of topics. Our intention, to bridge these different scales, becomes confusing without prior knowledge of all specific literature, which should not be the case. The paragraph focussed on the ansiform now just focuses on chief comparative and functional findings around the ansiform:

“Reorganizations^{52–55} within the CCS and its associated connectivity are hypothesized to be important drivers of global functional divisions within the cerebellum^{29,34,40}. Indeed, reorganizations within primate brains, rather than brain size alone, may best characterize anthropoids^{34,52}. The large lateral cerebellar crura I-II area is of elevated interest in primates. A cross-species homolog of human crura I-II has been referred to as the ansiform area^{56,57}, a term accredited to Louis Bolk’s work⁴¹ on the mammal cerebellum where he first defined cerebellar lobular terminology⁴³. We adopt the term ‘ansiform’ here to refer to this area. Relatively large lateral cerebellar hemispheres (mostly consisting of the ansiform) can be noted in four vertebrate clades including primates⁵⁸ and especially hominoids⁵⁹. Specifically, ansiform volume fractions of the cerebellum are larger in humans than in macaques⁶⁰, and appear generally larger in primates than in mice and rats⁵⁶. These gross anatomical expansions in primates, alongside cerebral transmodal expansions^{34,61–64}, could reflect selection on large-scale CCS networks^{45,58,60,65,66}. Indeed, expansion of the ansiform from macaques to chimps to humans⁶⁰, occurs alongside expansion of the PFC⁶⁴ and the termination zone that connects them in the cerebral peduncles expands in primates⁴⁰.

fMRI studies in humans indicate that the ansiform is activated in a diverse set of cognitive abilities including language, social cognition, and executive functions^{10,67}. It connects reciprocally to PFC, as well as temporal and parietal cortices in humans^{68,69}, and at least the PFC in cebus apella (tufted capuchin; a new world monkey)^{29,36,38,39}. Recent tractography work also illustrates high similarity

of human and chimpanzee CCSs³³. Human resting-state profiles support ansiform connectivity with cerebral transmodal areas¹²⁻¹⁴. Moreover, the area has a uniquely prolonged developmental trajectory⁷⁰. Its disruption is often connected to brain disorders with cognitive alterations⁷¹⁻⁷³ and developmental disruption strongly alters normal socio-cognitive development⁷⁴. Currently, it remains unclear how the ansiform scales relative to the CCS when taking evolutionary relationships among species into account.”

- Paragraph 4 tackles the involvement of cortico-cerebellar networks in primate brain evolution – this is OK, but there’s a tension here between conservative patterns and primates being different, which could be more explicitly stated

We thank the Reviewer for noting this tension, and tried to be explicit on when findings relate to primates, or subgroups being different (primarily see Revised manuscript methods/results). We aimed to focus on important findings of i. cerebellum-to-cerebral scaling (how they relate to each other and the brain) and ii. of relative ansiform scaling, both in primates specifically:

For cerebella:

“Comparative studies of the primate CCS have evidenced that: i. CCS hyperscaling may drive large primate brain size⁴⁴; ii. Components of the primate CCS scale together rather tightly in volume^{45,46} and neuron numbers^{47,48}; iii. The rate of cerebellar volumetric⁴⁹ and surface area⁵⁰ evolution may have increased relative to the neocortex (specifically towards humans); and iv. Contemporary humans and apes have large cerebella relative to recent human ancestors⁵¹. Together, these findings show that the cerebellum may have become of increased importance over the primate lineage.”

For the ansiform area see our comment above.

- Paragraph 5 starts “Previous work has revealed phylogenetic axes of functional organization” – what this means isn’t clear. Can you put it more simply? The rest of the paragraph again switches between issues a bit randomly (modularity, functional reorganization, ape-specific patterns, and comparisons of rats, macaques and humans) and doesn’t make a clear impression of its main point

So, I think the Introduction needs to be re-written to be much clearer, better organised and in particular to better explain how the current study fits with the existing literature.

We thank the Reviewer for this comment and believe the edits described above have positively impacted the motivation for the current work.

Methods

The methods seem generally OK, and it is good to have a new data set to address these issues. It is interesting that cerebellar vol estimation was significantly related to method (manual segmentation>automated)

- Did this interact with phylogeny? Did the methods deviate more for some taxa than others?

Unfortunately, we are not able to answer those questions since we were only able to perform automatic segmentations for the human brains. Applying automatic segmentation on even the chimpanzee brain images provided very unsatisfactory masks. It was noteworthy to us that the human automatic segmentation provided systematically lower values than manual segmentations

(although there was a high rank correlation). We explain this through under-inclusion of voxels towards the edges by CERES, and our own over-inclusion toward the cerebellar-pontine white matter border (which was done to lessen segmentation variability *across species* due to differences in data quality, brain geometry etc.).

On the topic of volumetric estimates, we did find some slight differences reported in volume by two different observers, but these were insignificant in ANOVA (volumes themselves did vary with phylogeny, but that is only expected).

Results

Ratios: why report ratios, when we know these are confounded by allometry? This just seems to confuse matters. This also applies to the ACE estimates – for these, you need to estimate values of $y \sim x$ estimated by your phylogenetic model. I can't see any justification for reporting or using ratios, which are biologically uninterpretable

We thank the Reviewer for raising this comment. First, we agree that ratios themselves can be hard to interpret biologically. Indeed, the hyper-allometry we find is directly able to account for large fractions of the ansiform within the cerebellum, the larger the CCS becomes. The fraction of the ansiform is by itself a result, and facilitates comparison with previous studies e.g., [Balsters et al., 2010; Hanson et al., 2013]. We believe the combination of showing the ratios and how they are explained by allometric scaling gives the most coherent and complete treatment of the data. In that sense, in the revised manuscript we have made explicit how ratios serve an illustrative purpose, furthering interpretation of evolutionary dynamics - supporting PGLS analyses.

“Traits of interest included full cerebellar, cerebral⁸³, and ansiform volumes, and cerebellum-to-cerebrum and ansiform-to-cerebellum ratios (Figure S1). We report ratios for illustrative purposes, facilitating better understanding of underlying scaling and hypothesizing about differences between clades. Ratios are confounded by allometry, with steep allometries leading to high ratios and vice versa. However, we aim to visualize how allometries may lead to reorganizations of fractions observed within extant primate's CCS^{60,94}, and thus provide ratios alongside our PGLS results.”

Using within-brain scaling (or remapping factors) is an often used strategy to deal with uncertainties about the data that seem to be inherent to the field of study [Hanson et al. 2013; Mars et al. 2014; Neubert et al. 2014; Passingham and Smaers 2014; Halley and Krubitzer 2019]. Fractions have previously been reported within the cerebellum with some speculation on how they are related to (comparative) cognition [Balsters et al., 2010; Hanson et al., 2010]. We aim to show how these fractions come about, because we believe the relatively enlarged areas do hold importance for species [Smaers et al., 2018]. Beside the concerns raised by the Reviewer, the exact ratios are as volatile as the volumes they are based on, so they are not intended to report a *ground truth* ratio of A-to-B. Because ratios themselves may be uninterpretable, they are also not used for phylogenetic model construction.

Ratios mapped to the phylogenetic tree (Figure 4) can thus complement PGLS regressions (Figure 5), by for example showing that those clades that end up above the primate-sample PGLS regression line, also display high ratios (for e.g. cerebellum to cerebrum) . By making explicit in the revised manuscript that we do not aim to predict exact ratios of traits for specific species - instead focusing on trends across primates and clades - we hope that the figures can also illustrate how distinct groups may have elevated scaling of a structure relative to another.

For example, we found the high cerebellum-to-cerebral ratios in lemurs and apes illustrative (especially in combination with their scaling in PGLS), since they may drive further hypothesizing on what has driven these relative scaling differences. We make explicit the purpose of the ratios in the results as well:

“We found no statistical evidence for relatively large cerebella in apes. Restricting analyses to haplorhines, as in Barton & Venditti’s 2014 study⁴⁹, somewhat accentuates a small non-significant difference in intercept in apes (Figure 6g). However, ancestral character estimations (ACEs; Figure 4) help illustrate that apes have relatively large cerebella relative to the primate sample, reflected in generally positive deviations in PGLS (Figure 5). Despite lack of significance, these results somewhat support acceleration of hominoid cerebellar expansion⁴⁹. In line with this notion, relatively large cerebella are noted in contemporary humans and apes, but not in recent human ancestors⁵¹⁹”

We also illustrate how ratios are not systematically influenced by sex as much as absolute measures (Figure S4). This is because our analyses aim to consider within-brain reorganization:

“Although brain-body scaling (Figure 7) may hold some connection to cognitive ability, as per the encephalization quotient hypothesis^{121,129,130}, we argue that reorganizations of behavior-supporting systems within brains may be more relevant. Furthermore, regressing against body mass may lead to inferential biases¹³³. Within the brain, both apes and lemurs appear to deviate positively from primate-general cerebello-cerebral scaling. Future studies may seek to explore behavioral correlates of large cerebella, which are likely quite different between both groups.”

To this end, we also included ACEs normalized against brain volume, to illustrate potential underlying patterns in brain reorganization:

Figure S7: ancestral reconstructions based on the same BM model, but now mapping back absolute volumes of cerebellum and cerebrum normalized to brain volume. This aims to illustrate potential underlying scaling differences across clades that may drive the main scaling relationship in Figure 5 (and 6).

As the Reviewer pointed out in their later comment on different drivers of high ratios (small cerebra versus large cerebella), our ratios may specifically fuel investigation into these mechanisms. We believe that the visualization of ratios (Figure 4) can be a useful addition to understanding the underlying data, as can volumes normalized against brain volume (Figure S7).

The point is very well taken: we believe that making it more explicit within the text that ratios may be driven by different scaling (smaller cerebra, or larger cerebella) is important. Additionally showing this through supplementary analysis (see answer to the comment on strepsirhine versus haplorhine scaling) will improve understanding of the data. This should be the primary objective since comparative data often provides contradictory results [Wartel et al. 2019].

Apes versus other primates ('No distinct allometries in apes'): here there is a lack of clarity about what you are testing

- what exactly is the model here, is it ANCOVA? Description of the results sounds like you just tested for an interaction effect, rather than main effect – if as stated the interaction is non-significant, this just means the slopes for apes and non-apes are homogenous. This wouldn't contradict Ref 70, in which a phylogenetic ANCOVA was used to show not only that slopes were homogenous but more interestingly that, after controlling for cortical volumes, apes have significantly larger cerebellar volumes, and that cerebellum volume evolved at a significantly higher rate in apes after accounting for rates of change in neocortex volumes. The analysis here doesn't obviously attempt to replicate that.

We thank the Reviewer for this comment and realize that we initially only tested for an interaction effect. In the revised manuscript, we used pANCOVA to test for differences in intercept, slope, or both. Splitting our data up in ape and non-ape subsets, we found no statistical support for either a grade shift, differences in slope, or an interaction:

Figure 6f-i: apes and non-apes do not scale significantly differently as assessed by pANCOVA (f). When restricting our analyses to haplorhines only (to improve comparability with Barton & Venditti 2014), we observe no significant scaling differences in our cerebellar ~ cerebral PGLS (g). Anisofirm area PGLSs show no differences in ape scaling either (h,i).

- In addition, Ref 70 was based on comparisons among haplorhine primates only, which makes it difficult to compare with the current results. In the current study, I think these analyses may misinterpret the data for strepsirrhines.

In order to make our data more comparable to this haplorhine-only dataset, we performed the same comparison of apes and non-apes in the haplorhine species only (Figure 6g). Although visual inspection of the PGLS slopes might suggest a slight grade shift (although with shallower slope; see Figure 6g), there was no statistical evidence for this:

(1) Differences in slopes, holding intercept constant:"

	df	Pr(>F)
Full	3	0.2849

Reduced	2	
---------	---	--

"(2) Differences in intercept, holding slopes constant:"

	df	Pr(>F)
Full	3	0.1995
Reduced	2	

"(3) Differences in slopes and differences in intercept:"

	df	Pr(>F)
Full	3	0.2896
Reduced	2	

Strepsirrhines have small relative neocortex sizes (probably because of their predominantly nocturnal habits and reduced visual cortices), which means that apparently high ratios of cerebellum to neocortex in this group has a completely different basis (small neocortex rather than large cerebellum) compared to haplorrhine primates. These effects should be separated. The authors say they “did not test lemur uniqueness to prevent hypothesizing after results are known”, but you do need some way of accounting for divergent patterns in the data (e.g. dummy code the taxa in your models)

As pointed out by the Reviewer, lemur and ape (and more generally: strepsirrhine and haplorrhine) relative scaling could be driven by different relative area sizes. We were interested in reporting the scaling of the cerebello-cerebral system in the primate sample as a whole (and testing if apes are different in slope or intercept as we specifically hypothesized they would be). However, although we were previously unaware of strepsirrhine species having testably smaller cerebra, the finding of a different intercept versus haplorrhine primates when ECV was scaled to body mass may point in this direction [Isler et al. 2008]. However, this finding needs the addition that cerebra actually have become smaller in this lineage.

Our ancestral character estimations show that our strepsirrhines have characteristically smaller cerebra than the haplorrhines. As the body mass data available to us comes from different specimens, we refrained from generally correcting for it in the original manuscript, to prevent potentially spurious relationships. Notwithstanding, when separating strepsirrhines from haplorrhines, and regressing cerebellar and cerebral volumes on body mass, we indeed find a specifically lower intercept for strepsirrhines cerebral volume (Figure 6b,c). This ‘grade shift’ was not significant:

CEREBELLUM

(1) Differences in slopes, holding intercept constant:"

	df	Pr(>F)
Full	3	0.8847

Reduced	2	
---------	---	--

"(2) Differences in intercept, holding slopes constant:"

	df	Pr(>F)
Full	3	0.9729
Reduced	2	

"(3) Differences in slopes and differences in intercept:"

	df	Pr(>F)
Full	3	0.9891
Reduced	2	

CEREBRUM

(1) Differences in slopes, holding intercept constant:"

	df	Pr(>F)
Full	3	0.9627
Reduced	2	

"(2) Differences in intercept, holding slopes constant:"

	df	Pr(>F)
Full	3	0.6496
Reduced	2	

"(3) Differences in slopes and differences in intercept:"

	df	Pr(>F)
Full	3	0.9028
Reduced	2	

Partly because encephalization can only hold partial information about cognitive ability and poorly distinguishes primate clades (Figure 7; previous critiques of Jerison's EQ), we instead consider within-brain organization by regressing against brain volume. When doing this, we find something interesting. Relative to the whole brain, strepsirrhines display a (very much non-significant; see below) positive 'grade shift' of cerebellar scaling relative to the rest of the brain versus haplorhines (Figure 6d):

CEREBELLUM

	df	Pr(>F)
Full	3	0.5282
Reduced	2	

"(2) Differences in intercept, holding slopes constant:"

	df	Pr(>F)
Full	3	0.4962
Reduced	2	

"(3) Differences in slopes and differences in intercept:"

	df	Pr(>F)
Full	3	0.7963
Reduced	2	

Cerebral scaling versus the rest of the brain is virtually identical in strepsirrhines and haplorhines (and hence even less significant):

CEREBRUM

(1) Differences in slopes, holding intercept constant:"

	df	Pr(>F)
Full	3	0.8766
Reduced	2	

"(2) Differences in intercept, holding slopes constant:"

	df	Pr(>F)
Full	3	0.9629
Reduced	2	

"(3) Differences in slopes and differences in intercept:"

	df	Pr(>F)
Full	3	0.8549
Reduced	2	

We are highly thankful that the Reviewer pointed out these effects, which we had not foreseen. The data somewhat hint at differential contributions of cerebellar and cerebral scaling in

strepsirrhines and haplorhines to our main effect, which we are now able to illustrate. Cerebellar scaling might trump cerebral scaling in contribution to the relatively large cerebellum found in Figures 4 and 5. We do not want to make conclusive statements, due to the clear lack of statistical significance. Since our primary goal is to give the fairest treatment of the data, we include Figure 6 into the Revised manuscript's main text, to show how different scaling may build up the main result.

Discussion:

First, to pick up the comments on the results above

- The Discussion reinforces the impression that scaling (slopes) and grade shifts (intercepts) are being conflated (or at least not clearly distinguished). For example, "Accelerated cerebellar volumetric evolution [70] in great apes may be a response to cerebral expansion" – as noted above, Ref 70 specifically ruled this out (the model accounted for evolutionary rates in cortical volume) . This part of the discussion is confusing

We thank the Reviewer for this comment. We corrected our misphrasing of the 2014 study of Barton & Venditti. We have now integrated the updated results:

"Second, for apes and non-apes, we observed only very minor potential differences between cerebellar-cerebrum PGLS regressions (f), even when restricting analysis to haplorhines (g). There were no apparent scaling differences for the ansiform (h,i). None of these comparisons were significant in pANCOVA, and Fisher's R-to-Z transformation showed that fit was always worse for separate regressions across apes/non-apes versus the whole sample (See Supplemental Text; Results)."

and updated the discussion:

"Previous studies have reported tight primate cerebello-cerebral scaling^{45,46}, as well as indications that cerebella may have become relatively large¹¹⁸ and experience accelerated growth in *hominoidea*⁴⁹. We found no statistical evidence for distinct allometries between apes and non-apes. Restricting analyses to haplorhines, as in Barton & Venditti's 2014 study⁴⁹, somewhat accentuates a small non-significant cerebellar grade shift in apes (Figure 6g). However, ancestral character estimations (ACEs; Figure 4) help illustrate that apes have relatively large cerebella relative to the primate sample, reflected in generally positive deviations in PGLS (Figure 5). Despite lack of significance, these results somewhat support acceleration of hominoid cerebellar expansion⁴⁹. In line with this notion, relatively large cerebella are noted in contemporary humans and apes, but not in recent human ancestors⁵¹."

- Again, on high cerebellar-cerebrum ratios in lemurs. In the Discussion this is interpreted as a function of large brain size ("lemurs may have large brains for their bodies, a trait shared with hominoids and especially humans") – this is not true; lemurs are relatively small brained among primates and, as noted above, the ratios alluded to are ambiguous and likely reflect smaller neocortex volumes.

We thank the Reviewer for pointing out our mistake. Our data indeed show that lemurs are generally small-brained primates that may have relatively large cerebella relative to cerebra and

the whole brain (Figure 6, Figure 7, Figure S7). What the ultimate reasons are for small/large brains, we feel, is beyond the scope of our current work.

Second, the discussion of Brownian motion – this is confusing and possibly muddled. The authors say that the fact that BM was the best model “at first glance, clashes with notions of adaptive evolution”. This is incorrect (as has been appreciated for a long time), The authors then seem to imply as much, but I didn’t find this clear. This section of the discussion doesn’t therefore seem to add anything useful, but merely confuse the issues

We have straightened out this part of the discussion. In short, we removed the initial confusing statement, instead positing simply that BM does not clash with adaptive evolution:

“The BM model, describing random incremental changes predictable by time since divergence, fits with constrained scaling of the CCS in our study. Although departures from BM have been interpreted to indicate adaptive variation^{106,125,126}, BM does not rule out adaptation¹⁰⁷. Recently, generalization of the BM model has been shown to represent both neutral drift and rapid adaptive evolutionary change, the latter of which BM does not do well¹²⁷”

We also mention in our discussion that our results - especially species’ deviations from allometry - should not be taken to support adaptive evolution either:

“Small intraspecific samples can thus not be representative of the range of natural variability. Although we show robustness of our general conclusions at the primate-level, this means that deviations for individual species cannot be used to infer adaptation, since precise volumes and ratios may differ. We show variability of absolute and relative volumes across our data to be substantial^{78–80}, as they include specimens across sexes and adult ages.”

The following material on scaling and on methodological points could be better focussed, and more concise

We have made this paragraph more to-the-point.

Once the issues with the analyses/results and interpretation of the data are resolved, the discussion could then be reworked accordingly

We have updated our discussion to fit with the updated interpretation of our results and to fit in better with the Revised manuscript.

Bibliography of literature mentioned in the covering letter (not formatted)

- Barton, R. A. & Venditti, C. Rapid evolution of the cerebellum in humans and other great apes. *Current Biology* 24, 2440–2444 (2014).
- Croxson, P. L., Forkel, S. J., Cerliani, L. & Thiebaut de Schotten, M. Structural Variability Across the Primate Brain: A Cross-Species Comparison. *Cerebral Cortex* 28, 3829–3841 (2018).
- Dawood Y, Hagoort J, Siadari BA, Ruijter JM, Gunst QD, Lobe NHJ, Strijkers GJ, de Bakker BS, van den Hoff MJB. Reducing soft-tissue shrinkage artefacts caused by staining with Lugol's solution. *Sci Rep.* 2021 Oct 5;11(1):19781. doi: 10.1038/s41598-021-99202-2. Erratum in: *Sci Rep.* 2022 Feb 7;12(1):2366. PMID: 34611247; PMCID: PMC8492742.
- Gellért A. and Csernovszki E. (1971). Anwendung der Paraffintechnik bei der Herstellung anatomischer Präparate. Aus einem NachlassMaterial zusammengestellt. *Studia medica Szegedinsia*, 8, 24.
- Giedd, J. N. et al. Child Psychiatry Branch of the National Institute of Mental Health Longitudinal Structural Magnetic Resonance Imaging Study of Human Brain Development. *Neuropsychopharmacology* 40, 43–49 (2015).
- Halley, A. C. & Krubitzer, L. Not all cortical expansions are the same: the coevolution of the neocortex and the dorsal thalamus in mammals. *Current Opinion in Neurobiology* 56, 78–86 (2019).
- Hanson, A., Grisham, W., Sheh, C., Annese, J. & Ridgway, S. Quantitative Examination of the Bottlenose Dolphin Cerebellum. *The Anatomical Record* 296, 1215–1228 (2013).
- Hedrick, B.P. et al., Assessing Soft-Tissue Shrinkage Estimates in Museum Specimens Imaged With Diffusible Iodine-Based Contrast-Enhanced Computed Tomography (diceCT), *Microscopy and Microanalysis*, Volume 24, Issue 3, 1 June 2018, Pages 284–291, <https://doi.org/10.1017/S1431927618000399>
- Herbin, M. et al. Do not Dispose of Historic Fluid Collections: Evaluating Research Potential and Range of Use. *Collection Forum* 34, 157–169 (2021)
- Heuer, K., Traut, N., Sousa, A. A. de, Valk, S. & Toro, R. Diversity and evolution of cerebellar folding in mammals. 2022.12.30.522292 Preprint at <https://doi.org/10.1101/2022.12.30.522292> (2022).
- Isler, K. et al. Endocranial volumes of primate species: scaling analyses using a comprehensive and reliable data set. *Journal of Human Evolution* 55, 967–978 (2008).
- Kretschmann, H. J., Tafesse, U. & Herrmann, A. Different volume changes of cerebral cortex and white matter during histological preparation. *Microsc Acta* 86, 13–24 (1982).
- Mars, R. B. et al. Primate comparative neuroscience using magnetic resonance imaging: Promises and challenges. *Frontiers in Neuroscience* 8, 298 (2014).
- Montie EW, Wheeler E, Pussini N, Battey TW, Barakos J, Dennison S, Colegrove K, Gulland F. Magnetic resonance imaging quality and volumes of brain structures from live and postmortem imaging of California sea lions with clinical signs of domoic acid toxicosis. *Dis Aquat Organ.* 2010 Sep 17;91(3):243-56. doi: 10.3354/dao02259. PMID: 21133324.
- Neubert, F.-X., Mars, R. B., Thomas, A. G., Sallet, J. & Rushworth, M. F. S. Comparison of Human Ventral Frontal Cortex Areas for Cognitive Control and Language with Areas in Monkey Frontal Cortex. *Neuron* 81, 700–713 (2014).
- Passingham, R. E. & Smaers, J. B. Is the Prefrontal Cortex Especially Enlarged in the Human Brain? Allometric Relations and Remapping Factors. *Brain, Behavior and Evolution* 84, 156–166 (2014).
- Quester, R. & Schröder, R. The shrinkage of the human brain stem during formalin fixation and embedding in paraffin. *Journal of Neuroscience Methods* 75, 81–89 (1997).
- Smaers, J. B., Turner, A. H., Gómez-Robles, A. & Sherwood, C. C. A cerebellar substrate for cognition evolved multiple times independently in mammals. *eLife* 7, e35696 (2018).
- Stephan, H., Frahm, H. & Baron, G. New and revised data on volumes of brain structures in insectivores and primates. *Folia Primatologica* 35, 1–29 (1981).
- Sydler C, Öhrström L, Rosendahl W, Woitek U, Rühli F. CT-Based Assessment of Relative Soft-Tissue Alteration in Different Types of Ancient Mummies. *Anat Rec (Hoboken).* 2015 Jun;298(6):1162-74. doi: 10.1002/ar.23144. PMID: 25998649.
- Treff WM, Kraus C. Formalinfixation des Gehirns. *Psychiatr Neurol Neurochir* 1960;63:116 – 24, so we will have to take the author's word for the statement.
- Wartel, A., Lindenfors, P. & Lind, J. Whatever you want: Inconsistent results are the rule, not the exception, in the study of primate brain evolution. *PLoS ONE* 14, e0218655 (2019).

Revised manuscript bibliography for quick reference (please find the sections and references in-text).

References

1. Van Overwalle, F. *et al.* Consensus Paper: Cerebellum and Social Cognition. *The Cerebellum* **19**, 833–868 (2020).
2. Koziol, L. F. *et al.* Consensus paper: The cerebellum's role in movement and cognition. *The Cerebellum* **13**, 151–177 (2014).
3. Mariën, P. *et al.* Consensus paper: Language and the cerebellum: An ongoing enigma. *Cerebellum* **13**, 386–410 (2014).
4. Adamaszek, M. *et al.* Consensus Paper: Cerebellum and Emotion. *Cerebellum* **16**, 552–576 (2017).
5. Botez, M. I., Gravel, J., Attig, E. & Vézina, J. L. Reversible chronic cerebellar ataxia after phenytoin intoxication: Possible role of cerebellum in cognitive thought. *Neurology* **35**, 1152–1157 (1985).
6. Schmahmann, J. D. & Sherman, J. C. The cerebellar cognitive affective syndrome. *Brain* **121**, 561–79 (1998).
7. Luciani, L. *Das Kleinhirn: neue Studien zur normalen und pathologischen Physiologie.* (E. Besold, 1893).
8. Kruithof, E. S., Klaus, J. & Schutter, D. J. L. G. The human cerebellum in reward anticipation and outcome processing: An activation likelihood estimation meta-analysis. *Neuroscience & Biobehavioral Reviews* **149**, 105171 (2023).
9. Stoodley, C. J. The cerebellum and cognition: evidence from functional imaging studies. *Cerebellum* **11**, 352–365 (2012).
10. Stoodley, C. J. & Schmahmann, J. D. Functional topography in the human cerebellum: A meta-analysis of neuroimaging studies. *NeuroImage* **44**, 489–501 (2009).
11. Buckner, R. L., Krienen, F. M., Castellanos, A., Diaz, J. C. & Thomas Yeo, B. T. The organization of the human cerebellum estimated by intrinsic functional connectivity. *Journal of Neurophysiology* **106**, 2322–2345 (2011).
12. Xue, A. *et al.* The Detailed Organization of the Human Cerebellum Estimated by Intrinsic Functional Connectivity Within the Individual. *Journal of Neurophysiology* **125**, 358–384 (2020).
13. Marek, S. *et al.* Spatial and Temporal Organization of the Individual Human Cerebellum. *Neuron* **100**, 977–993.e7 (2018).
14. Guell, X., Schmahmann, J. D., Gabrieli, J. D. E. & Ghosh, S. S. Functional gradients of the cerebellum. *eLife* **7**, e36652 (2018).
15. Leiner, H., Leiner, A. & Dow, R. Cerebro-cerebellar learning loops in apes and humans. *The Italian Journal of Neurological Sciences* **8**, 423–436 (1987).
16. Leiner, H., Leiner, A. & Dow, R. Does the Cerebellum Contribute to Mental Skills? *Behavioral Neuroscience* **100**, 443–454 (1986).
17. Schmahmann, J. D. From movement to thought: Anatomic substrates of the cerebellar contribution to cognitive processing. *Human Brain Mapping* **4**, 174–198 (1996).
18. Schmahmann, J. D. Disorders of the cerebellum: Ataxia, dysmetria of thought, and the cerebellar cognitive affective syndrome. *Journal of Neuropsychiatry and Clinical Neurosciences* **16**, 367–378 (2004).
19. Marr, D. A theory of cerebellar cortex. *The Journal of Physiology* **202**, 437–470 (1969).
20. Albus, J. S. A theory of cerebellar function. *Mathematical Biosciences* **10**, 25–61 (1971).
21. Kawato, M., Ohmae, S., Hoang, H. & Sanger, T. 50 Years Since the Marr, Ito, and Albus Models of the Cerebellum. *Neuroscience* **462**, 151–174 (2020).
22. Ito, M. & Kano, M. Long-lasting depression of parallel fiber-Purkinje cell transmission induced by conjunctive stimulation of parallel fibers and climbing fibers in the cerebellar cortex. *Neuroscience Letters* **33**, 253–258 (1982).
23. Eccles, J. C., Ito, M. & Szentágothai, J. *The Cerebellum as a Neuronal Machine. The Cerebellum as a Neuronal Machine* (Springer, 1967). doi:10.1007/978-3-662-13147-3.
24. Nieuwenhuys, R. Comparative Anatomy of the Cerebellum. *Progress in Brain Research* **25**, 1–93 (1967).
25. Larsell, O. & Jansen, J. *The Comparative Anatomy and Histology of the Cerebellum: Vol. 2. From Monotremes through Apes.* vol. 2 (University of Minnesota Press, 1970).
26. Larsell, O. & Jansen, J. *The Comparative Anatomy and Histology of the Cerebellum: Vol. 1. From Myxinoidea through Birds.* vol. 1 (University of Minnesota Press, 1967).
27. Marr, D. & Poggio, T. From Understanding Computation to Understanding Neural Circuitry. *A.I. memo* **357**, 1–22 (1976).
28. Marr, D. *Vision: A Computational Investigation of Visual Representation in Man.* vol. 8 (Freeman and Company, 1982).
29. Ramnani, N. The primate cortico-cerebellar system: Anatomy and function. *Nature Reviews Neuroscience* **7**, 511–523 (2006).
30. Diedrichsen, J., King, M., Hernandez-Castillo, C., Sereno, M. & Ivry, R. B. Universal Transform or Multiple Functionality? Understanding the Contribution of the Human Cerebellum across Task Domains. *Neuron* **102**, 918–928 (2019).
31. King, M., Shahshahani, L., Ivry, R. B. & Diedrichsen, J. A task-general connectivity model reveals variation in convergence of cortical inputs to functional regions of the cerebellum. *eLife* **12**, e81511 (2023).
32. Shahshahani, L., King, M., Nettekoven, C., Ivry, R. & Diedrichsen, J. Selective recruitment: Evidence for task-dependent gating of inputs to the cerebellum. 2023.01.25.525395 Preprint at <https://doi.org/10.1101/2023.01.25.525395> (2023).
33. Chauvel, M. Singularity of the white matter structural connectivity of the human brain compared to the chimpanzee brain. (Université Paris-Saclay, 2023).
34. Magielse, N., Heuer, K., Toro, R., Schutter, D. J. L. G. & Valk, S. L. A Comparative Perspective on the Cerebello-Cerebral System and Its Link to Cognition. *Cerebellum* (2022) doi:10.1007/s12311-022-01495-0.
35. Doya, K. What are the computations of the cerebellum, the basal ganglia and the cerebral cortex? *Neural Networks* **12**, 961–974 (1999).
36. Kelly, R. M. & Strick, P. L. Cerebellar loops with motor cortex and prefrontal cortex of a nonhuman primate. *Journal of Neuroscience* **23**, 8432–8444 (2003).

37. Dum, R. P. & Strick, P. L. An unfolded map of the cerebellar dentate nucleus and its projections to the cerebral cortex. *Journal of Neurophysiology* **89**, 634–639 (2003).
38. Middleton, F. A. & Strick, P. L. Cerebellar projections to the prefrontal cortex of the primate. *Journal of Neuroscience* **21**, 700–712 (2001).
39. Strick, P. L., Dum, R. P. & Fiez, J. A. Cerebellum and Nonmotor Function. *Annual Review of Neuroscience* **32**, 413–434 (2009).
40. Ramnani, N. *et al.* The evolution of prefrontal inputs to the cortico-pontine system: Diffusion imaging evidence from macaque monkeys and humans. *Cerebral Cortex* **16**, 811–818 (2006).
41. Bolk, L. *Das Cerebellum der Säugetiere: eine vergleichend anatomische Untersuchung.* (Fischer, 1906).
42. Glickstein, M. & Voogd, J. Lodewijk Bolk and the comparative anatomy of the cerebellum. *Trends in Neurosciences* **18**, 206–210 (1995).
43. Larsell, O. & Von Bartheldsdorf, S. The ansoparamedian lobule of the cerebellum and its correlation with the limb-muscle masses. *Journal of Comparative Neurology* **75**, 315–340 (1941).
44. Smaers, J. B. & Vanier, D. R. Brain size expansion in primates and humans is explained by a selective modular expansion of the cortico-cerebellar system. *Cortex* **118**, 292–305 (2019).
45. Smaers, J. B., Steele, J. & Zilles, K. Modeling the evolution of cortico-cerebellar systems in primates. *Annals of the New York Academy of Sciences* **1225**, 176–190 (2011).
46. Whiting, B. A. & Barton, R. A. The evolution of the cortico-cerebellar complex in primates: anatomical connections predict patterns of correlated evolution. *Journal of Human Evolution* **44**, 3–10 (2003).
47. Herculano-Houzel, S. The remarkable, yet not extraordinary, human brain as a scaled-up primate brain and its associated cost. *PNAS* **109**, 10661–10668 (2012).
48. Azevedo, F. A. C. *et al.* Equal numbers of neuronal and nonneuronal cells make the human brain an isometrically scaled-up primate brain. *Journal of Comparative Neurology* **513**, 532–541 (2009).
49. Barton, R. A. & Venditti, C. Rapid evolution of the cerebellum in humans and other great apes. *Current Biology* **24**, 2440–2444 (2014).
50. Sereno, M. I. *et al.* The human cerebellum has almost 80% of the surface area of the neocortex. *Proceedings of the National Academy of Sciences of the United States of America* **117**, 19538–19543 (2020).
51. Weaver, A. H. Reciprocal evolution of the cerebellum and neocortex in fossil humans. *Proceedings of the National Academy of Sciences of the United States of America* **102**, 3576–3580 (2005).
52. Smaers, J. B. & Soligo, C. Brain reorganization, not relative brain size, primarily characterizes anthropoid brain evolution. *Proceedings of the Royal Society B: Biological Sciences* **280**, 20130269 (2013).
53. Passingham, R. E. & Smaers, J. B. Is the Prefrontal Cortex Especially Enlarged in the Human Brain? Allometric Relations and Remapping Factors. *Brain, Behavior and Evolution* **84**, 156–166 (2014).
54. Halley, A. C. & Krubitzer, L. Not all cortical expansions are the same: the coevolution of the neocortex and the dorsal thalamus in mammals. *Current Opinion in Neurobiology* **56**, 78–86 (2019).
55. Neubert, F.-X., Mars, R. B., Thomas, A. G., Sallet, J. & Rushworth, M. F. S. Comparison of Human Ventral Frontal Cortex Areas for Cognitive Control and Language with Areas in Monkey Frontal Cortex. *Neuron* **81**, 700–713 (2014).
56. Luo, Y. *et al.* Lobular homology in cerebellar hemispheres of humans, non-human primates and rodents: a structural, axonal tracing and molecular expression analysis. *Brain Structure and Function* **222**, 2449–2472 (2017).
57. Sugihara, I. Crus I in the Rodent Cerebellum: Its Homology to Crus I and II in the Primate Cerebellum and Its Anatomical Uniqueness Among Neighboring Lobules. *The Cerebellum* **17**, 49–55 (2018).
58. Smaers, J. B., Turner, A. H., Gómez-Robles, A. & Sherwood, C. C. A cerebellar substrate for cognition evolved multiple times independently in mammals. *eLife* **7**, e35696 (2018).
59. MacLeod, C. E., Zilles, K., Schleicher, A., Rilling, J. K. & Gibson, K. R. Expansion of the neocerebellum in Hominoidea. *Journal of Human Evolution* **44**, 401–429 (2003).
60. Balsters, J. H. *et al.* Evolution of the cerebellar cortex: The selective expansion of prefrontal-projecting cerebellar lobules. *NeuroImage* **49**, 2045–2052 (2010).
61. Sherwood, C. C., Bauernfeind, A. L., Bianchi, S., Raghanti, M. A. & Hof, P. R. Human brain evolution writ large and small. *Progress in brain research* **195**, 237–254 (2012).
62. Hill, J. *et al.* Similar patterns of cortical expansion during human development and evolution. *Proceedings of the National Academy of Sciences of the United States of America* **107**, 13135–13140 (2010).
63. Buckner, R. L. & Krienen, F. M. The evolution of distributed association networks in the human brain. *Trends Cogn. Sci.* **17**, 648–665 (2013).
64. Bush, E. C. & Allman, J. M. The scaling of frontal cortex in primates and carnivores. *Proceedings of the National Academy of Sciences of the United States of America* **101**, 3962–3966 (2004).
65. Smaers, J. B. Modeling the Evolution of the Cerebellum. From Macroevolution to Function. *Progress in Brain Research* **210**, 193–216 (2014).
66. Smaers, J. B., Steele, J., Case, C. R. & Amunts, K. Laterality and the evolution of the prefronto-cerebellar system in anthropoids. *Annals of the New York Academy of Sciences* **1288**, 59–69 (2013).
67. Stoodley, C. J., Valera, E. M. & Schmahmann, J. D. An fMRI study of intra-individual functional topography in the human cerebellum. *Behavioural Neurology* **23**, 65–79 (2010).
68. Palesi, F. *et al.* Contralateral cerebello-thalamo-cortical pathways with prominent involvement of associative areas in humans in vivo. *Brain Structure and Function* **220**, 3369–3384 (2015).

69. Palesi, F. *et al.* Contralateral cortico-ponto-cerebellar pathways reconstruction in humans in vivo: Implications for reciprocal cerebro-cerebellar structural connectivity in motor and non-motor areas. *Scientific Reports* **7**, 1–13 (2017).
70. Liu, X. *et al.* A multifaceted gradient in human cerebellum of structural and functional development. *Nature Neuroscience* **25**, 1129–1133 (2022).
71. Moberget, T. *et al.* Cerebellar Gray Matter Volume Is Associated With Cognitive Function and Psychopathology in Adolescence. *Biological Psychiatry* **86**, 65–75 (2019).
72. Dong, D. *et al.* Compression of cerebellar functional gradients in schizophrenia. *Schizophrenia Bulletin* **46**, 1282–1295 (2020).
73. Morimoto, C. *et al.* Volumetric differences in gray and white matter of cerebellar Crus I/II across the different clinical stages of schizophrenia. *Psychiatry and Clinical Neurosciences* **75**, 256–264 (2021).
74. Badura, A. *et al.* Normal cognitive and social development require posterior cerebellar activity. *eLife* **7**, e36401 (2018).
75. Rilling, J. K. Human and NonHuman Primate Brains: Are They Allometrically Scaled Versions of the Same Design? *Evolutionary Anthropology* **15**, 65–77 (2006).
76. Schoenemann, P. T., Budinger, T. F., Sarich, V. M. & Wang, W. S. Y. Brain size does not predict general cognitive ability within families. *Proceedings of the National Academy of Sciences of the United States of America* **97**, 4932–4937 (2000).
77. Giedd, J. N. *et al.* Child Psychiatry Branch of the National Institute of Mental Health Longitudinal Structural Magnetic Resonance Imaging Study of Human Brain Development. *Neuropsychopharmacology* **40**, 43–49 (2015).
78. Charvet, C. J., Darlington, R. B. & Finlay, B. L. Variation in Human Brains May Facilitate Evolutionary Change toward a Limited Range of Phenotypes. *Brain Behav Evol* **81**, 74–85 (2013).
79. Reardon, P. K. *et al.* Normative brain size variation and brain shape diversity in humans. *Science* **360**, 1222–1227 (2018).
80. Croxson, P. L., Forkel, S. J., Cerliani, L. & Thiebaut de Schotten, M. Structural Variability Across the Primate Brain: A Cross-Species Comparison. *Cerebral Cortex* **28**, 3829–3841 (2018).
81. Harmon, L. J. & Losos, J. B. The effect of intraspecific sample size on type I and type II error rates in comparative studies. *Evolution; international journal of organic evolution* **59**, 2705–10 (2005).
82. Stephan, H., Frahm, H. & Baron, G. New and revised data on volumes of brain structures in insectivores and primates. *Folia Primatologica* **35**, 1–29 (1981).
83. Heuer, K. *et al.* Evolution of neocortical folding: A phylogenetic comparative analysis of MRI from 34 primate species. *Cortex* **118**, 275–291 (2019).
84. Arbuckle, S. A., Diedrichsen, J. & Pruszyński, J. A. Non-human primate anatomicals. (2018) doi:10.5281/ZENODO.1319671.
85. Craddock, C. *et al.* The Neuro Bureau Preprocessing Initiative: open sharing of preprocessed neuroimaging data and derivatives. *Frontiers in Neuroinformatics* **7**, (2013).
86. Milham, M. P. *et al.* An Open Resource for Non-human Primate Imaging. *Neuron* **100**, 61–74.e2 (2018).
87. Rilling, J. K. & Insel, T. R. The primate neocortex in comparative perspective using magnetic resonance imaging. *Journal of Human Evolution* **37**, 191–223 (1999).
88. Morey, R. A. *et al.* A comparison of automated segmentation and manual tracing for quantifying hippocampal and amygdala volumes. *Neuroimage* **45**, 855–866 (2009).
89. Heuer, K., Ghosh, S., Robinson Sterling, A. & Toro, R. Open Neuroimaging Laboratory. *Research Ideas and Outcomes* **2**, e9113 (2016).
90. Faber, J. *et al.* Manual sub-segmentation of the cerebellum. 2022.05.09.22274814 Preprint at <https://doi.org/10.1101/2022.05.09.22274814> (2022).
91. Romero, J. E. *et al.* CERES: A new cerebellum lobule segmentation method. *NeuroImage* **147**, 916–924 (2017).
92. Gamer, M., Lemon, J. & Singh, I. *irr: Various Coefficients of Interrater Reliability and Agreement*. (2010).
93. Koo, T. K. & Li, M. Y. A Guideline of Selecting and Reporting Intraclass Correlation Coefficients for Reliability Research. *Journal of Chiropractic Medicine* **15**, 155–163 (2016).
94. Hanson, A., Grisham, W., Sheh, C., Annese, J. & Ridgway, S. Quantitative Examination of the Bottlenose Dolphin Cerebellum. *The Anatomical Record* **296**, 1215–1228 (2013).
95. Shapiro, S. S. & Wilk, M. B. An Analysis of Variance Test for Normality (Complete Samples) on JSTOR. *Biometrika* **52**, 591–611 (1965).
96. Hedrick, B. P. *et al.* Assessing Soft-Tissue Shrinkage Estimates in Museum Specimens Imaged With Diffusible Iodine-Based Contrast-Enhanced Computed Tomography (diceCT). *Microsc Microanal* **24**, 284–291 (2018).
97. Wartel, A., Lindenfors, P. & Lind, J. Whatever you want: Inconsistent results are the rule, not the exception, in the study of primate brain evolution. *PLoS ONE* **14**, e0218655 (2019).
98. Freckleton, R. P. The seven deadly sins of comparative analysis. *J Evol Biol* **22**, 1367–1375 (2009).
99. Hooper, R., Brett, B. & Thornton, A. Problems with using comparative analyses of avian brain size to test hypotheses of cognitive evolution. *PLOS ONE* **17**, e0270771 (2022).
100. Arnold, C., Matthews, L. J. & Nunn, C. L. The 10kTrees website: A new online resource for primate phylogeny. *Evolutionary Anthropology* **19**, 114–118 (2010).
101. R Core Team. R: A Language and Environment for Statistical Computing. (2021).
102. Felsenstein, J. Phylogenies and the comparative method. *Am. Nat* **125**, 1–15 (1985).
103. Goolsby, E. W., Bruggeman, J. & Ané, C. Rphylopar: fast multivariate phylogenetic comparative methods for missing data and within-species variation. *Methods in Ecology and Evolution* **8**, 22–27 (2017).
104. Freckleton, R. P., Harvey, P. H. & Pagel, M. Phylogenetic analysis and comparative data: a test and review of evidence. *Am Nat* **160**, 712–726 (2002).

105. Uhlenbeck, G. E. & Ornstein, L. S. On the Theory of the Brownian Motion. *Physical Review* **36**, 823–841 (1930).
106. Harmon, L. J. *et al.* Early bursts of body size and shape evolution are rare in comparative data. *Evolution; international journal of organic evolution* **64**, 2385–2396 (2010).
107. Harmon, L. Phylogenetic Comparative Methods: Learning From Trees. Preprint at <https://doi.org/10.32942/OSF.IO/E3XNR> (2019).
108. Lande, R. Natural Selection and Random Genetic Drift in Phenotypic Evolution. *Evolution* **30**, 314–334 (1976).
109. Hansen, T. F. Stabilizing Selection and the Comparative Analysis of Adaptation. *Evolution* **51**, 1341–1351 (1997).
110. Akaike, H. A new look at the statistical model identification. *IEEE Transactions on Automatic Control* **19**, 716–723 (1974).
111. Symonds, M. R. E. & Blomberg, S. P. A primer on phylogenetic generalised least squares. in *Modern Phylogenetic Comparative Methods and their Application in Evolutionary Biology* 105–130 (Springer Berlin Heidelberg, 2014). doi:10.1007/978-3-662-43550-2_5.
112. Pinheiro, J., Bates, D. & R Core Team. nlme: Linear and Nonlinear Mixed Effects Models. (2022).
113. Paradis, E. & Schliep, K. ape 5.0: an environment for modern phylogenetics and evolutionary analyses in R. *Bioinformatics* **35**, 526–528 (2019).
114. Isler, K. *et al.* Endocranial volumes of primate species: scaling analyses using a comprehensive and reliable data set. *Journal of Human Evolution* **55**, 967–978 (2008).
115. Ives, A. R. R2s for Correlated Data: Phylogenetic Models, LMMs, and GLMMs. *Systematic Biology* **68**, 234–251 (2019).
116. Ives, A. & Li, D. rr2: An R package to calculate R2s for regression models. *Journal of Open Source Software* **3**, 1028 (2018).
117. Maseko, B. C., Spocter, M. A., Haagenen, M. & Manger, P. R. Elephants Have Relatively the Largest Cerebellum Size of Mammals. *The Anatomical Record: Advances in Integrative Anatomy and Evolutionary Biology* **295**, 661–672 (2012).
118. Rilling, J. K. & Insel, T. R. Evolution of the Cerebellum in Primates: Differences in Relative Volume among Monkeys, Apes and Humans. *Brain, Behavior and Evolution* **52**, 308–314 (1998).
119. Navarrete, A. F. *et al.* Primate Brain Anatomy: New Volumetric MRI Measurements for Neuroanatomical Studies. *Brain, Behavior and Evolution* **91**, 109–117 (2018).
120. Burnham, K. P. & Anderson, D. R. Multimodel inference: Understanding AIC and BIC in model selection. *Sociological Methods and Research* **33**, 261–304 (2004).
121. Jerison, H. J. *Evolution of the brain and intelligence*. (Academic Press, 1973).
122. Gutiérrez-Ibáñez, C., Iwaniuk, A. N. & Wylie, D. R. Parrots have evolved a primate-like telencephalic-midbrain-cerebellar circuit. *Scientific Reports* **8**, 9960 (2018).
123. Muller, A. S. & Montgomery, S. H. Co-evolution of cerebral and cerebellar expansion in cetaceans. *Journal of Evolutionary Biology* **32**, 1418–1431 (2019).
124. Shine, J. M. & Shine, R. Delegation to automaticity: The driving force for cognitive evolution? *Frontiers in Neuroscience* **8**, 90 (2014).
125. O’Meara, B. C., Ané, C., Sanderson, M. J. & Wainwright, P. C. Testing for different rates of continuous trait evolution using likelihood. *Evolution* **60**, 922–933 (2006).
126. Harmon, L. J., Schulte, J. A., Larson, A. & Losos, J. B. Tempo and Mode of Evolutionary Radiation in Iguanian Lizards. *Science* **301**, 961–964 (2003).
127. Elliot, M. G. & Mooers, A. Ø. Inferring ancestral states without assuming neutrality or gradualism using a stable model of continuous character evolution. *BMC Evolutionary Biology* **14**, 226 (2014).
128. Heuer, K., Traut, N., Sousa, A. A. de, Valk, S. & Toro, R. Diversity and evolution of cerebellar folding in mammals. 2022.12.30.522292 Preprint at <https://doi.org/10.1101/2022.12.30.522292> (2022).
129. Jerison, H. J. The theory of encephalization. *Annals of the New York Academy of Sciences* **299**, 146–160 (1977).
130. Jerison, H. J. Brain, body and encephalization in early primates. *Journal of Human Evolution* **8**, 615–635 (1979).
131. Van Schaik, C. P., Triki, Z., Bshary, R. & Heldstab, S. A. A Farewell to the Encephalization Quotient: A New Brain Size Measure for Comparative Primate Cognition. *Brain, Behavior and Evolution* **96**, 1–12 (2021).
132. Deaner, R. O., Isler, K., Burkart, J. & Van Schaik, C. Overall brain size, and not encephalization quotient, best predicts cognitive ability across non-human primates. *Brain, Behavior and Evolution* **70**, 115–124 (2007).
133. Rogell, B., Dowling, D. K. & Husby, A. Controlling for body size leads to inferential biases in the biological sciences. *Evolution Letters* **4**, 73–82 (2020).
134. Makris, N. *et al.* MRI-based surface-assisted parcellation of human cerebellar cortex: an anatomically specified method with estimate of reliability. *Neuroimage* **25**, 1146–1160 (2005).
135. Barton, R. A. & Harvey, P. H. Mosaic evolution of brain structure in mammals. *Nature* **405**, 1055–1058 (2000).
136. Finlay, B. L. & Darlington, R. B. Linked regularities in the development and evolution of mammalian brains. *Science* **268**, 1578–1584 (1995).
137. Yopak, K. E. *et al.* A conserved pattern of brain scaling from sharks to primates. *Proceedings of the National Academy of Sciences of the United States of America* **107**, 12946–12951 (2010).
138. Montgomery, S. H., Mundy, N. I. & Barton, R. A. Brain evolution and development: Adaptation, allometry and constraint. *Proceedings of the Royal Society B: Biological Sciences* **283**, 20160433 (2016).
139. Rakic, P. A small step for the cell, a giant leap for mankind: a hypothesis of neocortical expansion during evolution. *Trends Neurosci* **18**, 383–388 (1995).
140. Charvet, C. J., Striedter, G. F. & Finlay, B. L. Evo-Devo and Brain Scaling: Candidate Developmental Mechanisms for Variation and Constancy in Vertebrate Brain Evolution. *Brain Behav Evol* **78**, 248–257 (2011).
141. Wong, C. H. Y. *et al.* Fronto-cerebellar connectivity mediating cognitive processing speed. *NeuroImage* **226**, 117556 (2021).

142. Herculano-Houzel, S. Numbers of neurons as biological correlates of cognitive capability. *Current Opinion in Behavioral Sciences* **16**, 1–7 (2017).
143. Bush, E. C. & Allman, J. M. The scaling of white matter to gray matter in cerebellum and neocortex. *Brain, Behavior and Evolution* **61**, 1–5 (2003).
144. Semendeferi, K., Armstrong, E., Schleicher, A., Zilles, K. & Van Hoesen, G. W. Prefrontal Cortex in Humans and Apes: A Comparative Study of Area 10. *American Journal of Physical Anthropology* **114**, 224–241 (2001).
145. Eichert, N. *et al.* Cross-species cortical alignment identifies different types of anatomical reorganization in the primate temporal lobe. *eLife* **9**, e53232 (2020).
146. Amiez, C. *et al.* The relevance of the unique anatomy of the human prefrontal operculum to the emergence of speech. *Commun Biol* **6**, 1–12 (2023).
147. Quester, R. & Schröder, R. The shrinkage of the human brain stem during formalin fixation and embedding in paraffin. *Journal of Neuroscience Methods* **75**, 81–89 (1997).
148. Kretschmann, H. J., Tafesse, U. & Herrmann, A. Different volume changes of cerebral cortex and white matter during histological preparation. *Microsc Acta* **86**, 13–24 (1982).
149. Sherwood, C. C. & Smaers, J. B. What’s the fuss over human frontal lobe evolution? *Trends in Cognitive Sciences* **17**, 432–433 (2013).
150. Barton, R. A. & Venditti, C. Reply to Smaers: Getting human frontal lobes in proportion. *Proceedings of the National Academy of Sciences* **110**, E3683–E3684 (2013).
151. Bakker, R., Wachtler, T. & Diesmann, M. CoCoMac 2.0 and the future of tract-tracing databases. *Frontiers in Neuroinformatics* **6**, 30 (2012).
152. Mars, R. B. *et al.* Primate comparative neuroscience using magnetic resonance imaging: Promises and challenges. *Frontiers in Neuroscience* **8**, 298 (2014).
153. Friedrich, P. *et al.* Imaging evolution of the primate brain: the next frontier? *NeuroImage* **228**, 117685 (2021).
154. Powell, L. E., Isler, K. & Barton, R. A. Re-evaluating the link between brain size and behavioural ecology in primates. *Proceedings of the Royal Society B: Biological Sciences* **284**, 20171765 (2017).
155. Herbin, M. *et al.* Do not Dispose of Historic Fluid Collections: Evaluating Research Potential and Range of Use. *Collection Forum* **34**, 157–169 (2021).
156. Donahue, C. J., Glasser, M. F., Preuss, T. M., Rilling, J. K. & Van Essen, D. C. Quantitative assessment of prefrontal cortex in humans relative to nonhuman primates. *Proceedings of the National Academy of Sciences of the United States of America* **115**, E5183–E5192 (2018).

Reviewers' comments:

Reviewer #1 (Remarks to the Author):

The authors have responded to the previous reviews and rewritten the paper extensively trying to address the problems raised. However, there are still many obscure sentences and undefined terms. A few examples:

Abstract: what are "advanced associative abilities?"

What are functional modules?

Humans do indeed have unique cognitive abilities. They also have unique motor skills, being bipedal means the hands are freed and there is independent control of the fingers. Could the cerebellar expansion also be necessary for use of the hand?

Introduction: what does "cerebellar functional topography may be related to extracerebellar connectivity" mean?

Are you excluding vestibulo-cerebellar functions?

The anatomical relationship between cortex and cerebellum is critical but nowhere do you say that the cerebral cortex projects to cerebellum via relays in the pontine nuclei and that cerebellum projects back to cortex via thalamic relays.

Also you do not really make clear that there are regions of the cerebellum that are not interconnected with cortex.

It would be helpful to explain what isometric scaling is and means. Please define hypo-allometric.

Please define "exceptional scaling." Please define hyperscaling.

What does behavioral emergence mean?

The methods and results seem sound but there are major problems in extracting the hypotheses driving the study, the neuroanatomical basis for the study and the implications of the results.

Reviewer #3 (Remarks to the Author):

The revised version is substantially improved - in particular, the rewritten introduction is a vast improvement with much greater clarity and focus on the key issues. This is a good paper.

I will leave the other referees to address the issues they picked up on, such as data quality/reliability. The two substantive remaining issues from my perspective are as follows:

- responding to my query on the point of including ratios of cerebellum to neocortex size, the authors give, in my opinion, a tortuous set of reasons that do not justify including ratios. They suggest that the ratios help to interpret the allometric patterns ("We report ratios for illustrative purposes, facilitating better understanding of underlying scaling and hypothesizing about differences between clades"), but it doesn't make sense to say that ratios help you understand allometry: it is the other way around - the only way we can interpret the ratios is by reference to the allometries, and the latter is all we need to know about. They also mention comparability with previous studies, but the use of ratios in those studies was flawed for the same reason. In my view, including ratios and analyses/figures based on these (i) simply muddies the water, and (ii) encourages the unfortunate habit in the literature of treating them as though they are functionally meaningful. This has created huge problems in the field of brain evolution, where people fail to use allometric correction when making claims about the human brain. Just because Balsters et al, for example, discussed ratios,

doesn't justify continuing to do so and perpetuating misunderstandings. I recommend removal of the analyses and figures involving ratios unless simply and clearly to make the point that they give a false impression

- the authors have responded to my comments about lack of comparability with previous work on hominoid cerebellar enlargement by running phylogenetic ANCOVAs (as in Barton & Venditti 2014). This is fine, but they need to measure their conclusions in relation to the overall weight of evidence. Here they do not replicate the finding of a significant grade shift, although the trend is in that direction. Their data set for haplorrhine primates is, however, substantially smaller than that of B&V. In addition, I think they have included strepsirrhines in the relevant ANCOVA (hominoids versus all other primates) - as discussed in my original review and in their response, strepsirrhines appear to have a large cerebellum relative to neocortex, because they have a small neocortex (compared to haplorhines) not because they have a large cerebellum (as shown by the analyses controlling for body size). So the basis of cerebellar relative to neocortex size is completely different. Hence, this analysis will militate against finding a significant grade shift between apes and monkeys, and the replication needs to be done excluding strepsirrhines, ie. ANCOVA comparing hominoids and other haplorrhines – they don't do this analysis. If they don't have the statistical power to do this convincingly they should say so. It's important to note that B&V not only demonstrated the grade shift across a larger data set, they also replicated it across 6 out of 8 of the underlying data sets (B&V, supplementary information), and in terms of neuron numbers as well as volumes, so it is pretty well established (and by others since). Hence, the authors' statement at the end of the introduction "Contrary to our hypothesis based on previous work, hominoids did not exhibit exceptional scaling, showing that aniform hyperscaling may be primate-general" needs to be carefully contextualised so as not to give the impression that their analyses disprove relative cerebellar enlargement in apes.

Minor points:

I recommend removing "cognitive" from the title. The definition of what qualifies as 'cognitive' versus sensory, motor etc is contentious and probably it is not a sensible distinction, particularly when applied to a brain area.

Abstract: "advanced associative abilities" – here, the word 'advanced' implies progressive evolution towards a primate-like condition. Evolution is not progressive and the definition of 'advanced' is problematic. Primates are not on a 'more advanced' branch of evolution. It tends to perpetuate anthropocentric myths about evolution. May I suggest "specialised associative abilities"?

Response to Reviewers' comments:

Reviewer #1 (Remarks to the Author):

The authors have responded to the previous reviews and rewritten the paper extensively trying to address the problems raised. However, there are still many obscure sentences and undefined terms.

We thank the Reviewer for their consideration and for pointing out still unclear phrasing. We have made sure to simplify and clearly explain the relevant phrases. We have proofread the paper personally to identify potentially difficult to understand parts and have additionally asked our in-house proofreader to specifically simplify difficult phrases. We hope this helped further clarify obscure sentences and undefined terms. Please see detailed considerations and changes below.

A few examples:

**Abstract: what are “advanced associative abilities?”
What are functional modules?**

These terms are simplified in the Revised manuscript and given additional explanation. For the ‘advanced associative abilities’ in the abstract, we think the current phrasing better reflects the focus of the study:

“The reciprocal connections between the cerebellum and the cerebrum have been suggested to simultaneously play a role in brain size increase and to support a broad array of brain functions in primates.”

We explain early what is meant with this array of brain functions when considering macroscale evolution of the ansiform, without trying to provide an exhaustive list of functions. Here, we decide to give a global description of the functions that the area is suggested to support:

“Although it is impossible to summarize a brain area’s function in a single term – especially in an evolutionary context – human crura I-II appear to primarily be involved in coordinating more distributed processes associated with social cognition, language, and emotion. This idea is furthered by functional imaging, which reveals a strong involvement in the multi-demand and socio-linguistic networks, together suggesting the area’s general involvement in complex integrative computation. For example, crura I-II shows fMRI activations in tasks involving language and theory of mind, but also executive and emotional processing”

At the macroscale level it has been suggested that the cerebello-cerebral connection can be divided into functional modules which are areas that are reciprocally connected, structurally and functionally. Importantly, these connections are relatively separate from other ones at each level (cerebellum, cerebrum, dentate, pons, thalamus), prompting the ‘module’ term. To differentiate this usage from that describing the cerebellar computational unit, we briefly mention the term’s alternative usage by Apps et al. (2018):

“Within the highly connected CCS, motor and prefrontal cortex (PFC) streams are largely separated into distinct projection zones. Such an example of a separated information stream can be described as a distinct *functional module*, supporting relatively distinct brain functions.”

... “Note that this macroscale description of functional modules contrasts descriptions of the smallest operational unit that performs the cerebellar computation.”

Therefore, a module represents a much-simplified large-scale brain network. Despite only being able to study gross morphological changes in this sample, we can make functional inferences based on the strongly conserved connectivity of this system and its correspondence between structural and functional connectivity.

In the current work, we focus on the whole cerebellum and on the ansiform area, because previous work has shown the latter to become larger towards humans (Balsters et al., 2010). Additionally, many of the more complex, integrative functional processes related to socio-cognitive functioning and language activate these regions of the cerebellum in human task-based fMRI studies (Stoodley, 2012). Moreover, the ansiform area primarily connects (functionally and structurally) to transmodal cerebral cortical areas. We make explicit that this (structural and functional) connection/module guides our interpretation of the macroscale evolution of the ansiform:

“The cerebellum is not only connected to many areas of the cerebral cortex but also to the subcortex and other areas of the body. Its macroscale evolution could thus reflect selection on a combination of behaviors including sensorimotor, vestibular, and socio-cognitive abilities, as well as changes in social niche, diet, and mode of locomotion. However, macroscale evolutionary analyses of specific functional modules may reflect somewhat more specific selective forces relevant to primate evolution. Although it is impossible to summarize a brain area’s function in a single term – especially in an evolutionary context – crura I-II appear to primarily be involved in coordinating more distributed processes associated with social cognition, language, and emotion. This idea is furthered by functional imaging in humans, that reveals a strong involvement in the multi-demand and socio-linguistic networks, together suggesting the area’s general involvement in complex integrative computation. For example, crura I-II shows fMRI activations in tasks involving language and theory of mind, but also executive and emotional processing. Resting-state profiles and meta-analytic connectivity studies show that these functions may mainly come from cerebral connectivity, as the area is connected with cerebral areas also involved in such transmodal, integrative functions.”

“We focus our current study on both the whole cerebellar volume and, more specifically, the volume of the ansiform.”

Humans do indeed have unique cognitive abilities. They also have unique motor skills, being bipedal means the hands are freed and there is independent control of the fingers. Could the cerebellar expansion also be necessary for use of the hand?

We thank the Reviewer for raising this point. We agree that cerebellar expansion could have to do with selection on other functions/behaviors such as bipedality. As bipedality is expected to mostly alter cerebellar areas involved in motor processing, it is unlikely that the ansiform (primarily involved in non-motor processing) expansion came from this human characteristic. The ansiform is an area - and part of a functional module - that is mostly thought to play a role

in more transmodal, abstract processes such as theory of mind and language. Our observations argue against a role of bipedality in selection on this area, as humans do not appear to be unique in their ansiform scaling (they fall perfectly on the allometric line). Rather, quadruped primates also scale up their ansiform as their brains get bigger, so this scaling relationship might be suspected to be related to a more primate-general ability/behavior, as well.

Since the evolution of the ansiform does not appear exceptional in humans (but instead predictable based on the primate-general trend), it might be speculated that selection on the more abstract cognitive abilities ascribed to the ansiform may be primate-general, as well. See later our speculation on how expected (non-exceptional) scaling could nonetheless lead to new behavior. We try to explain this speculation clearly.

To further clarify how our findings may relate to evolutionary selection on behaviors, we have dedicated more room to explaining the functional modules within the cerebello-cerebral system. We mention* that while evolution of the whole cerebellum may reflect selection on a great host of behaviors, focusing on the functional module (and its cerebellar part, the ansiform) may yield a more specific interpretation:

**This paragraph was mentioned as a previous response as well.*

“The cerebellum is not only connected to many areas of the cerebral cortex but also to the subcortex and other areas of the body. Its macroscale evolution could thus reflect selection on a combination of behaviors including sensorimotor, vestibular, and socio-cognitive abilities, as well as changes in social niche, diet, and mode of locomotion. However, macroscale evolutionary analyses of specific functional modules may reflect somewhat more specific selective forces relevant to primate evolution. Although it is impossible to summarize a brain area’s function in a single term – especially in an evolutionary context – crura I-II appear to primarily be involved in coordinating more distributed processes associated with social cognition, language, and emotion. This idea is furthered by functional imaging in humans, that reveals a strong involvement in the multi-demand and socio-linguistic networks, together suggesting the area’s general involvement in complex integrative computation. For example, crura I-II shows fMRI activations in tasks involving language and theory of mind, but also executive and emotional processing. Resting-state profiles and meta-analytic connectivity studies show that these functions may mainly come from cerebral connectivity, as the area is connected with cerebral areas also involved in such transmodal, integrative functions.”

After finding that the ansiform area hyperscales in a primate-general fashion, we can thus more specifically suggest that selection on the functions supported area may not be specific to humans. When considering the evolutionary slope of the whole cerebellum, such speculation would necessarily be more aspecific, as its evolution is confounded by many more behaviors.

Introduction: what does “cerebellar functional topography may be related to extracerebellar connectivity” mean?

We are happy to explain. We meant to say that the whole cerebellar cortex may perform the same computation, leading to the function of a cerebellar area being closely related to its input/outputs.

Even if there are distinct computations, the high structural similarity of cerebellar cortical areas argues that inputs and outputs are essential in defining where functions are located in the cerebellum. We have reworded the paragraph to be much simpler:

“The cerebellum may encode errors, with the highly repetitive and evolutionarily conserved cerebellar circuit supporting the same computational process across behavioral domains (the universal cerebellar transform theory). This has prompted the idea that the location of cerebellar functions may be chiefly related to its inputs and outputs. Nonetheless, functional diversity may exist at the algorithmic level, with cerebellar areas integrating information from distributed brain areas, instead of simply mapping one-to-one to inputs. In either scenario, cerebellar function depends heavily on connectivity with the rest of the brain. However, how cerebellar volume has changed over the course of primate evolution relative to one of its primary connections, the cerebrum, remains incompletely understood.”

Inputs and outputs include many extra-cerebellar areas, but we focus our interpretation on the cerebral cortex, which is known to be highly interconnected with the ansiform area.

In our Revised manuscript, we make sure to mention that 1.) The cerebellum is widely connected. This means that expansion of the whole cerebellum can capture selection on many different behaviors, which we now state explicitly:

“The cerebellum is not only connected to many areas of the cerebral cortex but also to the subcortex and body. Its macroscale evolution could thus reflect selection on a combination of behaviors including sensorimotor, vestibular, and socio-cognitive abilities and adaptations in social niche, diet, and mode of locomotion.”

We also mention that the cerebral cortex is one of the primary connections more explicitly. This should serve as a reminder that there are more connections that influence cerebellar evolution:

“However, how cerebellar volume has changed over the course of evolution relative to one of its primary connections, the cerebrum, remains incompletely understood.”

We also mention that 2.) Previous studies investigating ansiform function have pointed towards a role in integrative brain functions. We go on to explain* that its evolution may thus hold more specific relation to selection on such integrative brain functions:

* Much of this paragraph has also formed responses to previous questions.

“However, macroscale evolutionary analyses of specific functional modules may reflect somewhat more specific selective forces relevant to primate evolution. Although it is impossible to summarize a brain area’s function in a single term – especially in an evolutionary context – crura I-II appear to primarily be involved in coordinating more distributed processes associated with social cognition, language, and emotion. This idea is furthered by functional imaging in humans, that reveals a strong involvement in the multi-demand and socio-linguistic networks, together suggesting the area’s general involvement in complex integrative computation. For example, crura I-II shows fMRI activations in tasks involving language and theory of mind, but also executive and emotional processing. Resting-state profiles and

meta-analytic connectivity studies show that these functions may mainly come from cerebral connectivity, as the area is connected with cerebral areas also involved in such transmodal, integrative functions.”

and:

“It is not guaranteed that ansiform functions established in humans will be organized in the same way in other primates. However, strong conservation of CCS structural connectivity across primates, alongside high correspondence between structural and functional connectivity, makes it reasonable to assume globally similar functions are supported by this area across primates.”

and:

“The ansiform is a remarkable cerebellar area. It has a uniquely prolonged developmental trajectory. Moreover, its disruption is often connected to brain disorders with cognitive alterations while developmental disruption strongly alters normal socio-cognitive development. In the capuchin (*Cebus apella*), it is interconnected with frontal rather than motor cortical areas. As in all cerebello-cerebral functional modules, a reciprocal closed-loop system connects the ansiform with transmodal cerebral areas through distinct pontine, thalamic, and dentate nucleus zones . Recent CCS tractography work also illustrates high similarity between humans and chimpanzees. Most generally, primate ansiforms connect reciprocally to transmodal cerebral cortical areas. Accordingly, evolutionary changes to the ansiform may reflect the selection of abilities supported by an ansiform-transmodal cerebral cortical module.”

Are you excluding vestibulo-cerebellar functions?

We appreciate the mention of cerebellar vestibular functions. Our main interpretation builds on the extensive ansiform-transmodal cerebral cortex connectivity, and functional activations in the ansiform. Vestibulo-cerebellar functions most definitely drive evolution of the cerebellum in some way (relative to the cerebrum, perhaps), as do many functions. We agree that it is good to explicitly name this and other cerebellar functions, and mention that overall cerebellar scaling should be driven by cerebellar involvement in a great diversity of functions:

“The cerebellum is not only connected to many areas of the cerebral cortex but also to the subcortex and other areas of the body. Its macroscale evolution could thus reflect selection on a combination of behaviors including sensorimotor, vestibular, as well as socio-cognitive abilities and adaptations in social niche, diet, and mode of locomotion.”

The anatomical relationship between cortex and cerebellum is critical but nowhere do you say that the cerebral cortex projects to cerebellum via relays in the pontine nuclei and that cerebellum projects back to cortex via thalamic relays.

We thank the Reviewer for this comment. At the start of the Revised manuscript, we have added a couple of sentences describing minimally the reciprocal connectivity of the cerebello-cerebral system that our work builds on. Additionally, we explicitly state that the functional modules we mention, are defined as separated, reciprocally connected subregions within this system:

“Within the highly connected CCS, motor and prefrontal cortex (PFC) streams are largely separated into distinct projection zones. Such an example of a separated information stream can be described as a distinct *functional module*, supporting relatively distinct brain functions.”

and:

“The ansiform is a remarkable cerebellar area. It has a uniquely prolonged developmental trajectory. Moreover, its disruption is often connected to brain disorders with cognitive alterations while developmental disruption strongly alters normal socio-cognitive development. In the capuchin (*Cebus apella*), it is interconnected with frontal rather than motor cortical areas. As in all cerebello-cerebral functional modules, a reciprocal closed-loop system connects the ansiform with transmodal cerebral areas through distinct pontine, thalamic, and dentate nucleus zones. Recent CCS tractography work also illustrates high similarity between humans and chimpanzees. Most generally, primate ansiforms connect reciprocally to transmodal cerebral cortical areas. Accordingly, evolutionary changes to the ansiform may reflect the selection of abilities supported by an ansiform-transmodal cerebral cortical module.”

Also you do not really make clear that there are regions of the cerebellum that are not interconnected with cortex.

We thank the Reviewer for pointing this out. We agree that it would give a fairer treatment if early on, we mention that overall cerebellar evolution is driven by widespread extra-cerebellar connectivity. Of this connectivity, a substantial part may be cerebral, but many other connections exist (vestibular, subcortical, to the rest of the body):

“In either scenario, cerebellar function depends heavily on connectivity with the rest of the brain. However, how cerebellar volume has changed over the course of evolution relative to one of its primary connections, the cerebrum, remains incompletely understood.”

and:

“A reciprocal CCS connection, supportive of cerebellar learning in motor and non-motor function, was first established in non-human primates (NHPs). Within the highly connected CCS, motor and prefrontal cortex (PFC) streams are largely separated into distinct projection zones. Such an example of a separated information stream can be described as a distinct *functional module*, supporting relatively distinct brain functions.”

and:

“The cerebellum is not only connected to many areas of the cerebral cortex but also to the subcortex and other areas of the body. Its macroscale evolution could thus reflect selection on a combination of behaviors including sensorimotor, vestibular, and socio-cognitive abilities, as well as changes in social niche, diet, and mode of locomotion.”

For the ansiform, specifically, we go on to mention that previous research has implied its strong interrelationship with the cerebral cortex. We also nuance the view, by emphasizing that the

transmodal cortex is its primary input/output. This means that studying its evolution may be somewhat more specific to the function supported by the ansiform-transmodal cortex connection:

“Resting-state profiles and meta-analytic connectivity studies show that these functions may mainly come from cerebral connectivity, as the area is connected with cerebral areas also involved in such transmodal, integrative functions.”

and:

“It is not guaranteed that ansiform functions established in humans will be organized in the same way in other primates. However, strong conservation of CCS structural connectivity across primates, alongside high correspondence between structural and functional connectivity, makes it reasonable to assume globally similar functions are supported by this area across primates.”

It would be helpful to explain what isometric scaling is and means. Please define hypo-allometric. Please define “exceptional scaling.” Please define hyperscaling.

In the Revised Abstract, we simplify the terms, to be more suitable for a broader audience:

“Together, our analyses support proportional cerebellar-cerebral scaling, whereas ansiforms have expanded faster than the cerebellum or cerebrum. We did not find different scaling between strepsirrhines and haplorhines, nor between apes and non-apes.”

In the Revised manuscript main text, we have upon the first mention of these terms made sure to explain that 1) isometric scaling means equal scaling (traits expand in the same proportion), 2) hypo-allometric scaling means a slope of expansion that is lower than isometric scaling (volume of region y increases slower than that of region x), and 3) hyper-allometry the opposite (region y increases faster than region x). We have added plain-language synonyms within brackets upon first mention of the specific jargon:

“We used a large (34 species, 63 specimens; Figure 1) open MRI dataset alongside modern phylogenetic comparative methods to consolidate previous evidence of isometric (volumes of region y and region x expand at the same rate) scaling between cerebellum and cerebrum. We evaluated potential differences in slopes or intercepts of cerebellar scaling in apes and non-apes, as well as strepsirrhines and haplorhines. Additionally, we aimed to assess ansiform hyper-allometry (hyperscaling; volume of y expands faster than x) in an allometric, phylogenetic context. Where possible, we explored intraspecific variability, substantial among brain traits and an inherent challenge for comparative primatology. Our analyses confirmed chiefly isometric scaling between cerebellum and cerebrum. Re-analysis of the Stephan collection data indicated that this relation may be hypo-allometric (hyposcaling; volume of y expands slower than x), a finding supported by several robustness analyses of our data. Conversely, by accounting for allometry and phylogeny, we find that primate-general ansiform hyper-allometry directly explains previous observations of this area’s large relative size in large-brained primates.”

Exceptional scaling should be interpreted as scaling that deviates significantly from the main scaling trend: scaling for a species or group of species that is exceptional relative to the entire sample. Since none of our results argue for exceptional scaling (all primate-general), it is not used in the Revised manuscript. Instead, we explain that our findings support primate-general scaling. For example:

“Conversely, by accounting for allometry and phylogeny, we find that primate-general ansiform hyperallometry directly explains previous observations of this area’s large relative size in large-brained primates.”

What does behavioral emergence mean?

We thank the Reviewer for inquiring about the use of this term. ‘Behavioral emergence’ is used to describe how behavioral abilities can arise from common anatomy at a larger scale. The described anatomical trajectories that humans lie at the end of, can by themselves not explain abilities that may be relatively unique to humans. We try to make this point by showing that the scaling of the cerebellum and ansiform is primate-general. Thus, anatomically, no pattern appears to support uniquely human behaviors by themselves.

However, emergently, human-unique behaviors *may* result from an interplay between high neuron numbers, increased transmodal connectivity and area volume, and likely many other factors. We make more explicit in the Revised manuscript that this is an informed speculation on our end:

“Connected to this, primate-general hyperscaling can reframe brain scaling as a matter of size. Humans may have specialized cognitive abilities and even exceptionally large relative brain size, but evolution has not acted to make humans exempt from the laws of evolution (the *scale naturæ*). Although specific anatomical specializations may in part cause human abilities, large human brains may also lead to some level of ‘behavioral emergence’, through an interaction of allometries that affect whole-brain scaling. Specifically, we speculate that the combination of many primate-general allometries might lead to increasingly abundant neuroanatomical systems that support the emergence of abilities that are more than the sum of their parts. The ansiform-transmodal cortex module clearly contributes to integrative primate behaviors, but its evolution is primate-general. Anatomically, humans may merely represent an extreme of many allometries: their large CCSs having enlarged modules supporting complex and abstract functional processing, increased white-matter connectivity, and a large neuronal pool. The same architecture at a different scale may support relatively unique behaviors.”

The methods and results seem sound but there are major problems in extracting the hypotheses driving the study, the neuroanatomical basis for the study and the implications of the results.

We hope the points and solutions raised above have taken care of these problems in communicating the rationale, findings, and implications of our study.

With respect to the specific points raised here, we hope the additional edits helped further illustrate our rationale by explaining the functional module concept more clearly. Moreover, by using the ansiform area-cerebral cortex reciprocal loop as an example, we further detailed the neuroanatomical basis of the study. Lastly, by simplifying terminology, and detailing how the causes of whole cerebellar and ansiform area evolution may differ, we hope to have further clarified the implications of our results.

Reviewer #3 (Remarks to the Author):

The revised version is substantially improved - in particular, the rewritten introduction is a vast improvement with much greater clarity and focus on the key issues. This is a good paper.

We thank the Reviewer for their appreciation of our Revision and work. We hope to succinctly take care of the remaining points, which we think have primarily resulted from misphrasing on our end.

I will leave the other referees to address the issues they picked up on, such as data quality/reliability.

The other Reviewers appear satisfied with our additional analyses examining data quality.

The two substantive remaining issues from my perspective are as follows:

- responding to my query on the point of including ratios of cerebellum to neocortex size, the authors give, in my opinion, a tortuous set of reasons that do not justify including ratios.

We thank the Reviewer greatly for their insight on this topic. We agree that part of our argumentation was circular and contrived, and that by themselves the allometries can describe necessary scaling information. We have taken out any description of ratios that implied their functional relevance.

We still report observed ratios in our sample in Table 1 and retain the Figures, but now only to show how (previous) comparative studies might misrepresent a functional implication of ratios (see later comment). Throughout the manuscript it is made clear that these ratios result **directly** from allometric scaling.

We believe this is valuable, since it will also help interpret previous studies that report ratios. By reporting how an allometry of ~ 1.3 can lead to high ratios of the ansiform in large-brained primates, we may clear up previous misinterpretation of comparative scaling findings (such as the Balsters 2010 and Hanson 2013 papers might spur). We have also slipped into such misinterpretation of ratios and think that mentioning (and showing) how they are related to allometry is important. In our opinion, it is better to address this problem than to ignore it.

They suggest that the ratios help to interpret the allometric patterns (“We report ratios for illustrative purposes, facilitating better understanding of underlying scaling and hypothesizing about differences between clades”), but it doesn’t make sense to say that ratios help you understand allometry: it is the other way around - the only way we can interpret the ratios is by reference to the allometries, and the latter is all we need to know about.

We agree with the Reviewer that the interpretation should be that allometries cause ratio differences to arise, not the other way around. We have removed any sentence implying functional relevance of the ratios themselves.

Technically, we agree that reporting the allometry should thus be sufficient. However, considering previous misinterpretations, including our own, we believe that there is value in showing and mentioning that ratios result from allometries (see above). We have thus retained throughout the manuscript but go on to explicitly mention how they result directly from allometry throughout the

Revised manuscript (including in the subscripts of the relevant Tables and Figures). This should allow us to better reconcile the implications of previous findings.

In the Methods:

“Cerebellar-to-cerebral and ansiform-to-cerebellum ratios are provided merely to show that volume ratios observed within extant primate’s CCS result directly from allometric scaling. Ratios are strongly confounded by allometry, with hyper-allometries leading to high ratios with increasing size, and vice versa.”

...and:

“Brain traits of species that diverged more recently are more likely to be similar. We detected substantial phylogenetic signal, and correlations between cerebello-cerebral traits illustrated severe collinearity (Figure 3), demonstrating the importance of accounting for phylogeny and allometry.”

In the Results:

We removed most of the discussion on ratios, replacing it with a sentence that explains that they result from allometry:

“We observed that ratios of cerebellum-to-cerebrum (4c) and ansiform-to-cerebellum (4e) may vary greatly across species (See also Table 1). However, these ratios result directly from the allometries described in (Figure 5a,c), and should thus not be interpreted functionally”

We also changed the title of this paragraph to echo the main message:

“Ratios are confounded by allometry”

In the Table and Figure subscripts we describe this problem, as well. Some examples (all changes are marked in yellow in the manuscript):

Table 1: “Species median measurements are provided for cerebellar, cerebral, and ansiform volumes (in mm³). Species ratios between median cerebellar and cerebral volumes, and ansiform and cerebellar volumes are also given (in percentages). Importantly, these ratios are reported merely to illustrate how allometric scaling of cerebello-cerebral system components may lead to wide-ranging ratios reported in the literature.”

Figure 4: “Note that the ratios in (c,e) serve merely to illustrate how ratios may result from the allometries described in **Figure 5a,c.**”

Figure 5: “Lemuriformes and Hominidea were the two clades with most impressive cerebellar-to-cerebral volume ratios (**Figure 4c**) and were thus specifically colored here (Lemuriformes colored in a bold gray; **a-d**) to show how ratios were confounded by the primate-general allometry between areas of the cerebello-cerebral system. Although species belonging to these infraorders displayed slightly higher

cerebellum-to-cerebrum scaling than the primate sample as a whole **(a)**, most of the differences in ratio results directly from allometry.”

and:

“Because of strong positive allometry, species with larger cerebella and cerebra are expected to have larger relative ansiforms, *directly accounting for high ratios reported in (4e)*.”

Figure 8: “Altogether, the analyses again show how allometry *causes* diverse ratios between brain areas *to arise* across species.”

Similar mentions of the role of allometry in causing ratios can be seen throughout the subscripts of our supplemental Figures.

They also mention comparability with previous studies, but the use of ratios in those studies was flawed for the same reason. In my view, including ratios and analyses/figures based on these (i) simply muddies the water, and (ii) encourages the unfortunate habit in the literature of treating them as though they are functionally meaningful. This has created huge problems in the field of brain evolution, where people fail to use allometric correction when making claims about the human brain. Just because Balsters et al, for example, discussed ratios, doesn't justify continuing to do so and perpetuating misunderstandings.

We hope that our explanation of allometry directly causing ratio differences to arise (including the warning to not value the interpretation of ratios - especially without allometric or phylogenetic correction) achieves the opposite effect. We believe it may rather help in reappraising previous, misinterpreted, work. Regarding the failure to use allometric and phylogenetic correction, we dedicated room in the Methods to mention how it is essential due to statistical interdependence of the traits and species:

Methods:

“Extant primate traits are not statistically independent, sharing variable amounts of evolutionary history: brain traits of species that diverged more recently are more likely to be similar. We detected substantial phylogenetic signal, and correlations between cerebello-cerebral traits illustrated severe collinearity (Figure 3), illustrating the importance of accounting for phylogeny and allometry.”

“To account for trait non-independence and reveal allometric scaling between traits, we performed phylogenetic generalized least squares (PGLS) regressions implemented in *nlme*.”

We have made this examination of cerebellar (and especially ansiform) evolution in an allometric and phylogenetic context a primary hook of our manuscript (mentioned in the Abstract and at the end of the Introduction):

Based on concerns from Reviewer #1, we refrained from using the terms ‘phylogeny’ and ‘allometry’ in the abstract directly, to keep the abstract inviting to a broader audience.

Abstract:

“Here, we manually segmented 63 cerebella (34 primate species; 9 infraorders) and 30 ansiforms (13 species; 8 infraorders) to understand how cerebellar and ansiform volumes have evolved over the primate lineage. Together, our analyses support proportional cerebellar-cerebral scaling, whereas ansiforms have expanded faster than the cerebellum or cerebrum. We did not find different scaling between strepsirrhines/haplorhines, nor between apes/non-apes. In sum, our study shows primate-general structural reorganization of the ansiform, relative to the cerebello-cerebral system. This holds significance for specialized brain functions in an evolutionary context.”

Introduction:

“Conversely, by accounting for allometry and phylogeny, we find that primate-general ansiform hyperscaling directly explains previous observations of this area’s large relative size in large-brained primates.”

I recommend removal of the analyses and figures involving ratios unless simply and clearly to make the point that they give a false impression

We opted for the second suggestion, retaining the analyses: they show how ratios may be misleading. We focus chiefly on describing that ratios are heavily confounded by allometry. We do this for our main analysis (Figure 4 and 5), but also for the analysis that aims to explore strepsirrhine/haplorhine scaling differences (Figure S7). In both, we show ratios on the tree to illustrate how ratio differences may result from primate allometries.

Within their respective subscript and in-text, we add the same note: these ratios result from allometry:

(for **Table 1**, **Figure 4**, and **Figure 5**: see above)

Figure S7: “*These normalized volumes show how interacting allometries may lead to proportional changes between cerebellum and brain (a) or cerebrum and brain (b) between strepsirrhines and haplorhines. The ratios are not to be interpreted functionally, unlike the allometries described in the main text, Figure 6.*”

- the authors have responding to my comments about lack of comparability with previous work on hominoid cerebellar enlargement by running phylogenetic ANCOVAs (as In Barton & Venditti 2014). This is fine, but they need to measure their conclusions in relation to the overall weight of evidence.

We thank the Reviewer for their consideration. We agree that we should carefully weigh our evidence with previous evidence. Especially in the Discussion, it is important to note that although our results do not support a grade shift, neither do they give a strong contraindication of the results of Barton & Venditti, 2014. Conversely, visually they argue in the same direction. Within our Results and Discussion, we interpret this shift carefully, since statistically it is not significant or trending towards significance.

Here they do not replicate the finding of a significant grade shift, although the trend is in that direction. Their data set for haplorrhine primates is, however, substantially smaller than that of B&V.

We agree that our sample size may explain the lack of significance, or of a trend towards significance ($p=.20$), for this grade shift. However, our analysis in Figure 6g (pANCOVA) includes a larger subset of species than most of the underlying datasets of Barton and Venditti 2014.

Moreover, the dataset is not much smaller than their main analysis (I counted 37 species on the tree). We think lack of power may be a fair explanation for the inconclusiveness of the combined sub-dataset analyses in BV2014 and our study. We mention in our manuscript that the oft-reported issue of small sample sizes might especially lead to controversy when approaching isometry, since smaller differences in underlying data can alter the overall conclusions:

“A combination of small samples, near-isometric scaling, and relatively small differences between apes and non-apes may cause controversy on cerebello-cerebral scaling.”

“Although sometimes assumed to be of minor importance, within-brain shrinkage was expected to play a larger role. Cerebellar, cerebral, and ansiform shrinkage may differ, perhaps based on white-matter content. Simulating this shrinkage (Figure S8c,d,g-j) revealed that conclusions on cerebello-cerebral scaling may differ depending on how areas have shrunk relative to one another. This may also explain contradictory conclusions on near-isometric cerebellar scaling across datasets.”

Our examinations in different data subsets, as well as simulation analyses, illustrate this problem as well. Whereas the ansiform scaling conclusion is resilient to tissue-specific shrinkage differences, cerebellum-cerebrum scaling can support a range of conclusions in a dataset of this N:

“Our main PGLS supports isometric cerebello-cerebral evolution, tending towards hypo-allometry (Figure 5a; ~ 0.95). To assess the effects of potential confounding factors, we ran several robustness analyses. These indicated that our main results generally hold when PGLS is rerun in data i) without potential outliers identified from the literature, ii) from only museum specimens, and iii) accounting for shrinkage differences between brain areas. Introducing shrinkage differences within-brain showed that cerebello-cerebral scaling may range from rather strong hypo-allometries (~ 0.8) to slightly weaker hyper-allometries (~ 1.1) (Figure S8c, d). Introducing severe artificial shrinkage differences across-brains showed mostly reduced allometries relative to our main PGLS (Figure S8e, f), as did reanalysis without literature outliers (S9a; ~ 0.93) and the Stephan replication (Figure S8b; ~ 0.92). A recent study reported similarly significant hypo-allometric cerebello-cerebral surface area scaling across mammals (~ 0.92). The MNHN-only PGLS was the exception (Figure S8a; ~ 0.98), with an increased slope relative to our main analysis. Although our results might hint at a slight cerebello-cerebral hypo-allometry, it becomes clear from these data how sparsely sampled, low-n, comparative data may lead to contradictory conclusions, especially when considering near-isometric scaling.”

Although separating apes from non-apes in the haplorhine sample (Figure 6g) - alongside apes' general positive-deviation from primate-general allometry (Fig 5a) - argues in the direction of (BV2014), we do not mention the grade shift as a trend due to insufficient statistics to back this up. Instead, we carefully appraise our results in relation to those of Barton and Venditti, recognizing their compelling evidence in combination with equally compelling archaeological evidence. We focus on the analysis of evolutionary rate, which appeared to give even stronger results, and which we did not do:

“Previous studies have reported tight primate cerebello-cerebral scaling, as well as indications that cerebella may have become relatively large and undergo accelerated growth in *homoidea*. Although we found no statistical evidence for relatively large cerebella in apes, restricting analyses to haplorhines (as in Barton & Venditti (2014)) somewhat accentuates a small non-significant grade shift in apes (Figure 6g). Additionally, apes’ positive deviations in PGLS (Figure 5, 8) argued towards accelerated cerebellar growth in apes. A combination of small samples, near-isometric scaling, and relatively small differences between apes and non-apes may cause conflicting conclusions on cerebello-cerebral scaling. However, our study did not consider branch-wise evolutionary rates, which appeared more strongly accelerated in apes. In line with the notion of accelerated cerebellar growth, relatively large cerebella are noted in contemporary humans and apes, but not in recent human ancestors.”

In addition, I think they have included strepsirrhines in the relevant ANCOVA (hominoids versus all other primates) - as discussed in my original review and in their response, strepsirrhines appear to have a large cerebellum relative to neocortex, because they have a small neocortex (compared to haplorhines) not because they have a large cerebellum (as shown by the analyses controlling for body size). So the basis of cerebellar relative to neocortex size is completely different.

We thank the Reviewer for pointing this out! We are happy to further clarify. Whereas Figure 6f includes our entire sample for the ape vs. non-ape pANCOVA, Figure 6g includes just haplorhines as suggested by the Reviewer previously. In this analysis, the 9 strepsirrhine species are taken out, testing ape vs. non-ape scaling with pANCOVA in 25 haplorhine species.

We agree with the Reviewer that strepsirrhine brains, and cerebra, are smaller and that that can drive the higher cerebellar relative size (as revealed in PGLS). We mention how large relative cerebella can come from different underlying scaling between strepsirrhines/haplorhines:

“Relatively large cerebella were also noted in lemurs, although this may result from smaller cerebra in strepsirrhines.”

and:

“Within the brain, both apes and lemurs appear to deviate positively from primate-general cerebello-cerebral scaling in PGLS. Future studies may seek to explore how relatively large cerebella may result from cerebral scaling, as well as investigate behavioral correlates of large cerebella. Both are likely quite different between strepsirrhines and haplorhines.”

In our response, we argued that within-brain, cerebral scaling is virtually identical between strepsirrhines and haplorhines (even visually indistinguishable in PGLS) in our sample (6e). Since our analysis does not deal with brain scaling relative to the body (6b,c, Figure 7), but rather brain reorganization (6d,e), we feel that this cerebrum-to-brain scaling is more relevant to our interpretation than cerebrum-to-body scaling is. Strepsirrhines may indeed have smaller cerebra relative to their bodies (although we could not statistically confirm a grade shift in our data, $p=.65$). It may be more accurate to state that cerebrum-to-body scaling is different between the groups, directly resulting from brain-to-body scaling differences. However, within the brain, our data slightly

suggest that the cerebellum might be relatively bigger in strepsirrhines. We do not overstate or - interpret this result either, due to a lack of significance ($p=.50$):

“Although strepsirrhines are smaller-brained and have smaller cerebra, within-brain their cerebella might be slightly larger than those of haplorhines (although not significantly), whereas cerebra scaled virtually identically relative to brain volume between groups (Figure 6). Although brain-body scaling (Figure 7) may be related to cognitive ability, as per the encephalization quotient hypothesis, we argue that reorganizations of behavior-supporting systems within brains may be more relevant. Brain reorganization may paint a nuanced and complementary picture regarding links to comparative ability alongside (relative) brain size. Within the brain, both apes and lemurs appear to deviate positively from primate-general cerebello-cerebral scaling. Future studies may seek to explore how relatively large cerebella may result from cerebral scaling, as well as investigate behavioral correlates of large cerebella. Both are likely greatly different between strepsirrhines/haplorhines”

Hence, this analysis will militate against finding a significant grade shift between apes and monkeys, and the replication needs to be done excluding strepsirrhines, ie. ANCOVA comparing hominoids and other haplorrhines – they don’t do this analysis.

We thank the Reviewer for raising this valid concern. We are happy to refer to our analysis in (Figure 6g) where we performed ape vs. non-ape pANOCVA in the haplorhine-only dataset (N=25 species, of which N=7 apes).

If they don’t have the statistical power to do this convincingly they should say so.

Although our sample size can be considered fairly large for a comparative study, we agree that not finding a statistically significant grade shift may be related to relatively low power in combination with a relationship that is close to isometry, which we make explicit in the Discussion:

“Although our results might hint towards a slight cerebello-cerebral hypo-allometry, it becomes clear from these data how sparsely sampled, low-n comparative data may lead to contradictory conclusions, especially when considering near-isometric scaling. Our robustness analyses show that providing several complementary views on comparative data may reveal more than small sample p-values.”

It’s important to note that B&V not only demonstrated the grade shift across a larger data set, they also replicated it across 6 out of 8 of the underlying data sets (B&V, supplementary information), and in terms of neuron numbers as well as volumes, so it is pretty well established (and by others since).

We agree that B&V2014 provide compelling evidence and have thus rephrased our interpretation so that it is not taken to oppose the validity of their results. Within our Results, we report a failure to replicate, which - in the Discussion - we go on to explain as potentially related to lack of power. However, we do mention that studies on this topic thus remain slightly indecisive. We explain why studies may remain indecisive, and reiterate how important large datasets and a common framework are:

“A combination of small samples, near-isometric scaling, and relatively small differences between apes and non-apes may cause controversy on cerebello-cerebral scaling.”

“Rerunning PGLS in data excluding potential outlying cerebellar volumes based on the literature (Figure S9) revealed that the main PGLS was robust, but had potentially somewhat overestimated cerebello-cerebral scaling. Discrepancies between these results and volumes reported across the literature show impact of intraspecific variation.”

“Our study compares brain areas within specimens, helping to minimize the influence of variability across brains. Notwithstanding, shrinkage could alter exact scaling formulae and would ideally be accounted for.”

“Although sometimes assumed to be of minor importance, within-brain shrinkage was expected to play a larger role. Cerebellar, cerebral, and ansiform shrinkage may differ, perhaps based on white-matter content. Simulating this shrinkage (Figure S8c,d,g-j) revealed that conclusions on cerebello-cerebral scaling may differ depending on how areas have shrunk relative to one another. This may also explain contradictory conclusions on near-isometric cerebellar scaling across datasets. Ansiform hyperscaling was robust to shrinkage. In the future, standard recording of fresh brain weights, and calculation of time-related shrinkage across brain areas may help combat uncertainty in volumetric estimates and potentially contradictory results.”

“Community-wide sharing accelerates comparative neurosciences

Systematic analysis of larger datasets containing both many species and sizable intraspecific samples might resolve previous controversies. Creating community-wide guidelines for data collection and statistical handling can facilitate integration of data from primate studies. Statistical methods to incorporate intraspecific samples already exist. To further data integration, primate databases can be used to earmark primate brains with unique identifiers, linked to relevant metadata including sex, age, body weight, captivity status, data availability, and inclusion in specific studies. Lastly, museum collections can greatly expand comparative samples.”

Hence, the authors' statement at the end of the introduction “Contrary to our hypothesis based on previous work, hominoids did not exhibit exceptional scaling, showing that ansiform hyperscaling may be primate-general” needs to be carefully contextualised so as not to give the impression that their analyses disprove relative cerebellar enlargement in apes.

We agree that careful contextualisation is important. Concerning the paragraph above, we have now made explicit that we were referring to Balsters et al. in 2010. In the Revised manuscript, we have made clear that this hypothesis was based on non-phylogenetically, non-allometry corrected analysis that overinterpreted ratios (subtly, without implying any conscious wrongdoing by the respective authors):

“Conversely, by accounting for allometry and phylogeny, we find that primate-general ansiform hyperscaling directly explains previous observations of this area’s large relative size in large-brained primates.”

Concerning cerebellar relative to cerebral scaling, we recontextualize the lack of significance as a potential combination of i.) lack of power, and related ii.) data specific idiosyncrasies, or iii). a biological effect (scaling between apes and non-apes appears generally relatively close to each other, and to isometry).

See our above response on cerebello-cerebral scaling conceptualisation.

We also explicitly mention that our study does not consider branch-specific evolutionary rates, which painted a strong picture of accelerated cerebellar evolution in apes in BV2014:

“However, our study did not consider branch-wise evolutionary rates, which appeared more strongly accelerated in apes. In line with the notion of accelerated cerebellar growth, relatively large cerebella are noted in contemporary humans and apes, but not in recent human ancestors.”

Minor points:

I recommend removing “cognitive” from the title. The definition of what qualifies as ‘cognitive’ versus sensory, motor etc is contentious and probably it is not a sensible distinction, particularly when applied to a brain area.

We have removed the word ‘cognitive’ from the title, recognizing it may under- or misrepresent the functional diversity of the ansiform area. We have instead added emphasis on the primate-general components of our Results by adding it to the title:

“PRIMATE-GENERAL EXPANSION OF CEREBELLAR CRURA I-II: A 34-SPECIES PHYLOGENETIC COMPARATIVE ANALYSIS OF THE CEREBELLO-CEREBRAL SYSTEM”

Abstract: “advanced associative abilities” – here, the word ‘advanced’ implies progressive evolution towards a primate-like condition. Evolution is not progressive and the definition of ‘advanced’ is problematic. Primates are not on a ‘more advanced’ branch of evolution. It tends to perpetuate anthropocentric myths about evolution. May I suggest “specialised associative abilities”?

We thank the Reviewer for pointing out this important point. We are strongly opposed to the ancient idea of a *scala naturæ*. Although our usage of advanced was supposed to indicate advanced abilities in general i.e., abilities that are not basic in their nature, we recognize how easily this may be misinterpreted to mean abilities that are more advanced in some than in others.

Reviewer #1 raised valid concerns on the usage of terms that may not be generally known, or correctly understood, by a broader audience. Therefore, we opted to not use the ‘specialized associative abilities’ here. In the Abstract, we keep it highly general:

“The reciprocal connections between the cerebellum and the cerebrum have been suggested to simultaneously play a role in brain size increase and support a broad array of brain functions in primates.”

To establish what type of brain functions we mean, we explain the proposed function of the ansiform (over primate evolution):

“Although it is impossible to summarize a brain area’s function in a single term – especially in an evolutionary context – crura I-II appear to primarily be involved in coordinating more distributed processes associated with social cognition, language, and emotion. This idea is furthered by functional imaging in humans, that reveals a strong involvement in the multi-demand and socio-linguistic networks, together suggesting the area’s general involvement in complex integrative computation. For example, crura I-II

shows fMRI activations in tasks involving language and theory of mind, but also executive and emotional processing. Resting-state profiles and meta-analytic connectivity studies show that these functions may mainly come from cerebral connectivity, as the area is connected with cerebral areas also involved in such transmodal, integrative functions. A cross-species homolog of human crura I-II has been referred to as the ansiform area. The term is accredited to Louis Bolk's work on the mammal cerebellum, where he first defined cerebellar lobular terminology. We adopt the term 'ansiform' here to refer to this area. It is not guaranteed that ansiform functions established in humans will be organized in the same way in other primates. However, strong conservation of CCS structural connectivity across primates, alongside high correspondence between structural and functional connectivity, makes it reasonable to assume globally similar functions are supported by this area across primates."

REVIEWERS' COMMENTS:

Reviewer #1 (Remarks to the Author):

The manuscript has been extensively rewritten to address the concerns of the reviewers. I have no further suggestions.

Reviewer #3 (Remarks to the Author):

The authors have responded carefully to remaining concerns and I have no further comments. Happy to see this nice paper go forward